# A Nearly Optimal and Low-Switching Algorithm for Reinforcement Learning with General Function Approximation

## Abstract

The exploration-exploitation dilemma has been a central challenge in reinforcement learning (RL) with complex model classes. In this paper, we propose a new algorithm, Monotonic Q-Learning with Upper Confidence Bound (MQL-UCB) for RL with general function approximation, where the Bellman operator of the underlying Markov decision process (MDP) is assumed to map any value functions into a function class with a bounded eluder dimension. Our key algorithmic design includes (1) a general deterministic policy-switching strategy that achieves low switching cost, (2) a monotonic value function structure with carefully controlled function class complexity, and (3) a variance-weighted regression scheme that exploits historical trajectories with high data efficiency. MQL-UCB achieves minimax optimal regret of $\widetilde{O}(d\sqrt{HK})$ when $K$ is sufficiently large and near-optimal policy switching cost of $\widetilde{O}(dH)$, with $d$ being the eluder dimension of the function class, $H$ being the planning horizon, and $K$ being the number of episodes. Our work sheds light on designing provably sample-efficient and deployment-efficient Q-learning with nonlinear function approximation.

## 1 Introduction

In reinforcement learning (RL), a learner interacts with an unknown environment and aims to maximize the cumulative reward. As one of the most mainstream paradigms for sequential decision-making, RL found extensive applications in many real-world problems (Kober et al., 2013; Mnih et al., 2015; Lillicrap et al., 2015; Zoph & Le, 2016; Zheng et al., 2018). Theoretically, the RL problem is often formulated as a Markov Decision Process (MDP) (Puterman, 2014). Achieving the optimal regret bound for various MDP settings has been a long-standing fundamental problem in RL research. In tabular MDPs where the state space $\mathcal{S}$ and the action space $\mathcal{A}$ are finite and computationally tractable, the optimal regret bound has been well-established ranging from episodic settings to discounted settings (Azar et al., 2017; Zanette & Brunskill, 2019; Zhang & Ji, 2019; Simchowitz & Jamieson, 2019; Zhang et al., 2020; He et al., 2021b). Nevertheless, these regret guarantees are intolerably large in many real-world applications, where the state space $\mathcal{S}$ and the action space $\mathcal{A}$ are often large and even infinite.

As is commonly applied in applications, *function approximation* schemes have been adopted by theorists to demonstrate the generalization across large state-action spaces, proving the performance guarantees of various RL algorithms for specific function classes. Most close to the tabular MDPs, there were recent works on MDP with linear function approximation under different assumptions such as linear MDPs (Yang & Wang, 2019; Jin et al., 2020), linear mixture MDPs (Modi et al., 2020; Ayoub et al., 2020; Zhou et al., 2021a). Among them, Zhou et al. (2021a) achieved nearly optimal regret bounds for linear mixture MDPs through a model-based approach adopting variance-weighted linear regression. Later, Hu et al. (2022) proposed LSVI-UCB+ algorithm, making an attempt to improve the regret for linear MDP through an over-optimistic value function approach. However, their analysis was later discovered to suffer from a technical issue (Agarwal et al., 2022; He et al., 2022). To fix this issue, Agarwal et al. (2022) introduced similar over-optimistic value functions to construct a monotonic variance estimator and a non-Markovian planning phase, achieving the first statistically optimal regret for linear MDPs. Concurrently, He et al. (2022) proposed LSVI-UCB++, which takes a different approach and employs a rare-switching technique to obtain the optimal regret. In a parallel line of research, there has been a growing body of literaure proposing more general frameworks to unify sample efficient RL algorithms, e.g., Bellman rank (Jiang et al., 2017), Witness rank (Sun et al., 2019), eluder dimension (Russo & Van Roy, 2013), Bellman eluder dimension (Jin

et al., 2021), Bilinear Classes (Du et al., 2021), Decision-Estimation Coefficient (Foster et al., 2021), Admissible Bellman Characterization (Chen et al., 2022) and Decoupling Coefficient (Agarwal & Zhang, 2022a;b). However, when applying these frameworks to linear MDPs, none of them can achieve minimax optimal regret. Notably, Agarwal et al. (2022) proposed a novel algorithm dubbed VO$Q$L for RL with general function approximation, which, to our knowledge, is the only algorithmic framework achieving optimal regret for RL beyond linear function approximation. However, VO$Q$L requires a complicated planning procedure, where the resulting policy does not act greedily with respect to a single optimistic value function. It is natural to ask

> *Whether we can generalize the approach by He et al. (2022) to solve MDPs with general function approximation and simple Markovian planning phase[1] to achieve the optimal regret?*

While the sample efficiency of RL algorithms for MDPs with nonlinear function classes has been comprehensively researched, *deployment efficiency* (Matsushima et al., 2020) is also a major concern in many real-world application scenarios. For example, in recommendation systems (Afsar et al., 2022), it may take several weeks to deploy a new recommendation policy. On the other hand, the system is capable of collecting an enormous amount of data every minute implementing a fixed policy. As a result, it is computationally inefficient to change the executed policy after each data point is collected as is demanded by most of the online RL algorithms in theoretical studies. To resolve this issue, Bai et al. (2019) first introduced the concept of switching cost, defined as the number of policy updates. Following this concept, a series of RL algorithms have been proposed on the theoretical side with low switching cost guarantees (e.g., Zhang et al., 2020; Wang et al., 2021; Qiao et al., 2022; Kong et al., 2021; Velegkas et al., 2022; Li et al., 2023). Among them, only Kong et al. (2021), Velegkas et al. (2022), and Li et al. (2023) considered RL with general function approximation, all of which achieved the switching cost of $O(d^2 H \text{polylog}(K))$, where $d$ is the eluder dimension of the underlying function class, $H$ is the planning horizon, and $K$ is the number of episodes. In contrast, Gao et al. (2021) proved a $\Omega(dH/\log d)$ lower bound of switching cost for any deterministic algorithms in learning linear MDPs. Therefore, the following question remains open:

*Can we design an algorithm with $\widetilde{O}(dH)$ switching cost for MDPs with bounded eluder dimension?*

In this paper, we answer the above two questions simultaneously by proposing a novel algorithm *Monotonic Q-Learning with UCB (MQL-UCB)* with all the aforementioned appealing properties. At the core of our algorithmic design are the following innovative techniques:

- We propose a novel policy-switching strategy based on the cumulative sensitivity of historical data. To the best of our knowledge, this is the first deterministic rare-switching strategy for RL with general function approximation which achieves $\widetilde{O}(dH)$ switching cost. We also prove a nearly matching lower bound for any algorithm with arbitrary policies including both deterministic and stochastic policies (See Theorem C.1 in Appendix C). Previous approaches for low switching cost in RL with general function approximation are sampling-based (Kong et al., 2021; Velegkas et al., 2022; Li et al., 2023) and highly coupled with a sub-sampling technique used for regression, making it less efficient. When restricted to the linear case, sampling-based rare-switching has a worse switching cost of $\widetilde{O}(d^2 H)$ (Section 3.3 in Kong et al. 2021; Theorem 3.3 in Velegkas et al. 2022; Section C in Li et al. 2023) than that in Wang et al. (2021).

- With the novel policy-switching scheme, we illustrate how to reduce the complexity of value function classes while maintaining a series of monotonic value functions, strictly generalizing the LSVI-UCB++ algorithm (He et al., 2022) to general function class with bounded eluder dimension. Based on the structure of the value functions, we further demonstrate how MQL-UCB achieves nearly minimax optimal sample complexity with delicately designed variance estimators. Our work is the first work for RL with general function approximation that achieves the nearly minimax optimal regret when specialized to linear MDPs, while still enjoys simple Markov planning phase.

It is worth noting that very recently, Xiong et al. (2023) also considered low-switching RL with general function approximation, which achieves $\widetilde{O}(dH)$ switching cost. Compared to their approach, our algorithm has a tractable value-iteration-based planning phase and enjoys a minimax optimal regret guarantee at the same time. Qiao et al. (2023) also considered RL with low-switching cost beyond linear function approximation, i.e., MDPs with low inherent Bellman error (Zanette et al.,

---

[1]Markovian planning phase means that the action executed by the algorithm only depends on the current state instead of the prefix trajectory. It is more aligned with the empirical RL approaches since the estimated value function is not well-defined under non-Markovian policy.

Table 1: A comparison of existing algorithms in terms of regret and switching cost for linear MDP and general function class with bounded eluder dimension and Bellman completeness. The results hold for in-homogeneous episodic RL with horizon length $H$, number of episodes $K$ where the total reward obtained in an episode is not larger than 1. For regret, we only present the leading term when $K$ is large enough compared to other variables and hide poly-logarithmic factors in $K$, $d$ or $\dim$, $H$ and the constant. For linear MDPs, $d$ is the dimension of the feature vectors. For general function class, $\dim$ is a shorthand of the eluder dimension of the underlying function class, $\mathcal{N}$ is the covering number of the value function class, and $\mathcal{N}_{\mathcal{S},\mathcal{A}}$ is the covering number of the state-action space.

| Algorithm | Regret | # of Switches | Model Class |
|---|---|---|---|
| LSVI-UCB (Jin et al., 2020) | $d^{3/2}H\sqrt{K}$ | $K$ | |
| LSVI-UCB-RareSwitch (Wang et al., 2021) | $d^{3/2}H\sqrt{K}$ | $\widetilde{O}(dH)$ | Linear MDPs |
| LSVI-UCB++ (He et al., 2022) | $d\sqrt{HK}$ | $\widetilde{O}(dH)$ | |
| $\mathcal{F}$-LSVI (Wang et al., 2020a) | $\dim(\mathcal{F})\sqrt{\log\mathcal{N}\log\mathcal{N}_{\mathcal{S},\mathcal{A}}}\cdot H\sqrt{K}$ | $K$ | |
| GOLF (Jin et al., 2021) | $\sqrt{\dim(\mathcal{F})\log\mathcal{N}}\cdot H\sqrt{K}$ | $K$ | Bounded eluder dimension + Completeness |
| VOQL (Agarwal et al., 2022) | $\sqrt{\dim(\mathcal{F})\log\mathcal{N}\cdot HK}$ | $\widetilde{O}(d^2H)$ | |
| MQL-UCB (Theorem 4.1) | $\sqrt{\dim(\mathcal{F})\log\mathcal{N}\cdot HK}$ | $\widetilde{O}(dH)$ | |

2020b) and generalized linear MDPs Wang et al. (2020b). Their approach can be seen as a slight extension of RL with low-switching cost for linear MDPs, since in both settings, the covariance matrix still exists and they can still use the determinant of the covariance as a criterion for policy switching.

**Notation.** We use lower case letters to denote scalars and use lower and upper case bold face letters to denote vectors and matrices respectively. We denote by $[n]$ the set $\{1,\ldots,n\}$. For a vector $\mathbf{x}\in\mathbb{R}^d$ and a positive semi-definite matrix $\boldsymbol{\Sigma}\in\mathbb{R}^{d\times d}$, we denote by $\|\mathbf{x}\|_2$ the vector's Euclidean norm and define $\|\mathbf{x}\|_{\boldsymbol{\Sigma}}=\sqrt{\mathbf{x}^\top\boldsymbol{\Sigma}\mathbf{x}}$. For two positive sequences $\{a_n\}$ and $\{b_n\}$ with $n=1,2,\ldots$, we write $a_n=O(b_n)$ if there exists an absolute constant $C>0$ such that $a_n\leq Cb_n$ holds for all $n\geq 1$ and write $a_n=\Omega(b_n)$ if there exists an absolute constant $C>0$ such that $a_n\geq Cb_n$ holds for all $n\geq 1$. We use $\widetilde{O}(\cdot)$ to further hide the polylogarithmic factors except log-covering numbers. We use $\mathbb{1}\{\cdot\}$ to denote the indicator function.

## 2 PRELIMINARIES

### 2.1 TIME-INHOMOGENEOUS EPISODIC MDP

We consider a time-inhomogeneous episodic Markov Decision Process (MDP), denoted by a tuple $\mathcal{M}=M(\mathcal{S},\mathcal{A},H,\{\mathbb{P}_h\}_{h=1}^H,\{r_h\}_{h=1}^H)$. Here, $\mathcal{S}$ and $\mathcal{A}$ are the spaces of state and action, respectively, $H$ is the length of each episode, $\mathbb{P}_h:\mathcal{S}\times\mathcal{A}\times\mathcal{S}\to[0,1]$ is the transition probability function at stage $h$ which denotes the probability for state $s$ to transfer to next state $s'$ with current action $a$, and $r_h:\mathcal{S}\times\mathcal{A}\to[0,1]$ is the deterministic reward function at stage $h$. A policy $\pi:=\{\pi_h\}_{h=1}^H$ is a collection of mappings $\pi_h$ from a observed state $s\in\mathcal{S}$ to the simplex of action space $\mathcal{A}$. For any policy $\pi=\{\pi_h\}_{h=1}^H$ and stage $h\in[H]$, we define the value function $V_h^\pi(s)$ and the action-value function $Q_h^\pi(s,a)$ as follows:

$$Q_h^\pi(s,a)=r_h(s,a)+\mathbb{E}\left[\sum_{h'=h+1}^H r_{h'}\big(s_{h'},\pi_{h'}(s_{h'})\big)\bigg|s_h=s,a_h=a\right],\ V_h^\pi(s)=Q_h^\pi(s,\pi_h(s)),$$

where $s_{h'+1}\sim\mathbb{P}_{h'}(\cdot|s_{h'},a_{h'})$. Then, we further define the optimal value function $V_h^*$ and the optimal action-value function $Q_h^*$ as $V_h^*(s)=\max_\pi V_h^\pi(s)$ and $Q_h^*(s,a)=\max_\pi Q_h^\pi(s,a)$. For simplicity, we assume the total reward for each possible trajectory $(s_1,a_1,...,s_H,a_H)$ satisfies $\sum_{h=1}^H r_h(s_h,a_h)\leq 1$. Under this assumption, the value function $V_h^\pi(\cdot)$ and $Q_h^\pi(\cdot,\cdot)$ are bounded in $[0,1]$. For any function $V:\mathcal{S}\to\mathbb{R}$ and stage $h\in[H]$, we define the following first-order Bellman operator $\mathcal{T}_h$ and second-order Bellman operator $\mathcal{T}_h^2$ on function $V$:

$$\mathcal{T}_h V(s_h,a_h)=\mathbb{E}_{s_{h+1}}\big[r_h+V(s_{h+1})|s_h,a_h\big],\mathcal{T}_h^2V(s_h,a_h)=\mathbb{E}_{s_{h+1}}\Big[\big(r_h+V(s_{h+1})\big)^2|s_h,a_h\Big],$$

where $s_{h+1} \sim \mathbb{P}_h(\cdot|s_h, a_h)$ and $r_h = r_h(s_h, a_h)$. For simplicity, we further define $[\mathbb{P}_h V](s, a) = \mathbb{E}_{s' \sim \mathbb{P}_h(\cdot|s,a)} V(s')$ and $[\mathbb{V}_h V](s, a) = \mathcal{T}_h^2 V(s_h, a_h) - \left(\mathcal{T}_h V(s_h, a_h)\right)^2$. Using this notation, for each stage $h \in [H]$, the Bellman equation and Bellman optimality equation take the following forms:

$$Q_h^\pi(s, a) = \mathcal{T}_h V_{h+1}^\pi(s, a), \quad Q^*(s, a) = \mathcal{T}_h V_{h+1}^*(s, a),$$

where $V_{H+1}^\pi(\cdot) = V_{H+1}^*(\cdot) = 0$. At the beginning of each episode $k \in [K]$, the agent selects a policy $\pi^k$ to be executed throughout the episode, and an initial state $s_1$ is arbitrary selected by the environment. For each stage $h \in [H]$, the agent first observes the current state $s_h^k$, chooses an action following the policy $\pi_h^k$, then transits to the next state with $s_{h+1}^k \sim \mathbb{P}_h(\cdot|s_h^k, a_h^k)$ and reward $r_h(s_h, a_h)$. Based on the protocol, we defined the suboptimality gap in episode $k$ as the difference between the value function for selected policy $\pi^k$ and the optimal value function $V_1^*(s_1^k) - V_1^{\pi^k}(s_1^k)$. Based on these definitions, we can define the regret in the first $K$ episodes as follows:

**Definition 2.1.** For any RL algorithm `Alg`, the regret in the first $K$ episodes is denoted by the sum of the suboptimality for episode $k = 1, \ldots, K$,

$$\text{Regret}(K) = \sum_{k=1}^{K} V_1^*(s_1^k) - V_1^{\pi^k}(s_1^k),$$

where $\pi^k$ is the agent's policy at episode $k$.

### 2.2 FUNCTION CLASSES AND COVERING NUMBERS

**Assumption 2.2** (Completeness). Given $\mathcal{F} := \{\mathcal{F}_h\}_{h=1}^{H}$ which is composed of bounded functions $f_h : \mathcal{S} \times \mathcal{A} \to [0, L]$. We assume that for any function $V : \mathcal{S} \to [0, 1]$ there exists $f_1, f_2 \in \mathcal{F}_h$ such that for any $(s, a) \in \mathcal{S} \times \mathcal{A}$,

$$\mathbb{E}_{s' \sim \mathbb{P}_h(\cdot|s,a)} \big[r_h(s, a) + V(s')\big] = f_1(s, a), \text{ and } \mathbb{E}_{s' \sim \mathbb{P}_h(\cdot|s,a)} \Big[\big(r_h(s, a) + V(s')\big)^2\Big] = f_2(s, a).$$

We assume that $L = O(1)$ throughout the paper.

**Remark 2.3.** Completeness is a fundamental assumption in RL with general function approximation, as recognized in Wang et al. (2020b); Jin et al. (2021); Agarwal et al. (2022). Our assumption is the same as that in Agarwal et al. (2022) and is slightly stronger than that in Wang et al. (2020a) and Jin et al. (2021). More specifically, in Wang et al. (2020a), completeness is only required for the first-order Bellman operator. In contrast, we necessitate completeness with respect to the second-order Bellman operator, which becomes imperative during the construction of variance-based weights. Jin et al. (2021) only requires the completeness for the function class $\mathcal{F}_{h+1}$ ($\mathcal{T}_h \mathcal{F}_{h+1} \subseteq \mathcal{F}_h$). However, the GOLF algorithm Jin et al. (2021) requires to solve an intricate optimization problem across the entire episode. In contrast, we employ pointwise exploration bonuses as an alternative strategy, which requires the completeness for function class $\mathcal{V} = \{V : \mathcal{S} \to [0, 1]\}$, i.e., $\mathcal{T}_h \mathcal{V} \subseteq \mathcal{F}_h$. The completeness assumption on the second moment is first introduced by Agarwal et al. (2022), and is crucial for obtaining a tighter regret bound. More specifically, making use of the variance of the value function at the next state is known to be crucial to achieve minimax-optimal regret bound in RL ranging from tabular MDPs (Azar et al., 2017) to linear mixture MDPs (Zhou et al., 2021a) and linear MDPs (He et al., 2022). In RL with general function approximation, the second-moment completeness assumption makes the variance of value functions computationally tractable.

**Definition 2.4** (Generalized Eluder dimension, Agarwal et al. 2022). Let $\lambda \geq 0$, a sequence of state-action pairs $\mathbf{Z} = \{z_i\}_{i \in [T]}$ and a sequence of positive numbers $\boldsymbol{\sigma} = \{\sigma_i\}_{i \in [T]}$. The generalized Eluder dimension of a function class $\mathcal{F} : \mathcal{S} \times \mathcal{A} \to [0, L]$ with respect to $\lambda$ is defined by $\dim_{\alpha, T}(\mathcal{F}) := \sup_{\mathbf{Z}, \boldsymbol{\sigma} : |Z| = T, \boldsymbol{\sigma} \geq \alpha} \dim(\mathcal{F}, \mathbf{Z}, \boldsymbol{\sigma})$

$$\dim(\mathcal{F}, \mathbf{Z}, \boldsymbol{\sigma}) := \sum_{i=1}^{T} \min\left(1, \frac{1}{\sigma_i^2} D_{\mathcal{F}}^2(z_i; z_{[i-1]}, \sigma_{[i-1]})\right),$$

$$D_{\mathcal{F}}^2(z; z_{[t-1]}, \sigma_{[t-1]}) := \sup_{f_1, f_2 \in \mathcal{F}} \frac{(f_1(z) - f_2(z))^2}{\sum_{s \in [t-1]} \frac{1}{\sigma_s^2}(f_1(z_s) - f_2(z_s))^2 + \lambda}.$$

We write $\dim_{\alpha, T}(\mathcal{F}) := H^{-1} \cdot \sum_{h \in [H]} \dim_{\alpha, T}(\mathcal{F}_h)$ for short when $\mathcal{F}$ is a collection of function classes $\mathcal{F} = \{\mathcal{F}_h\}_{h=1}^{H}$ in the context.

**Remark 2.5.** The $D_{\mathcal{F}}^2$ quantity has been introduced in Agarwal et al. (2022) and Ye et al. (2023) to quantify the uncertainty of a state-action pair given a collected dataset with corresponding weights. It was inspired by Gentile et al. (2022) where an unweighted version of uncertainty has been defined for active learning. Prior to that, Wang et al. (2020a) introduced a similar 'sensitivity' measure to determine the sampling probability in their sub-sampling framework. As discussed in Agarwal et al. (2022), when specialized to linear function classes, $D_{\mathcal{F}}^2(z_t; z_{[t-1]}, \sigma_{[t-1]})$ can be written as the elliptical norm $\|z_t\|_{\boldsymbol{\Sigma}_{t-1}^{-1}}^2$, where $\boldsymbol{\Sigma}_{t-1}$ is the weighted covariance matrix of the feature vectors $z_{[t-1]}$.

**Definition 2.6** (Bonus oracle $\bar{D}_{\mathcal{F}}^2$)**.** In this paper, the bonus oracle is denoted by $\bar{D}_{\mathcal{F}}^2$, which computes the estimated uncertainty of a state-action pair $z = (s, a) \in \mathcal{S} \times \mathcal{A}$ with respect to historical data $z_{[t-1]}$ and corresponding weights $\sigma_{[t-1]}$. In detail, we assume that a computable function $\bar{D}_{\mathcal{F}}^2(z; z_{[t-1]}, \sigma_{[t-1]})$ satisfies

$$D_{\mathcal{F}}(z; z_{[t-1]}, \sigma_{[t-1]}) \leq \bar{D}_{\mathcal{F}}(z; z_{[t-1]}, \sigma_{[t-1]}) \leq C \cdot D_{\mathcal{F}}(z; z_{[t-1]}, \sigma_{[t-1]}),$$

where $C$ is a fixed constant.

**Remark 2.7.** Agarwal et al. (2022) also assumed access to such a bonus oracle defined in Definition 2.6, where they assume that the bonus oracle finds a proper bonus from a finite bonus class (Definition 3, Agarwal et al. 2022). Our definition is slightly different in the sense that the bonus class is not assumed to be finite but with a finite covering number. Previous works by Kong et al. (2021) and Wang et al. (2020a) proposed a sub-sampling idea to compute such a bonus function efficiently in general cases, which is also applicable in our framework. In a similar nonlinear RL setting, Ye et al. (2023) assumed that the uncertainly $D_{\mathcal{F}}^2$ can be directly computed, which is a slightly stronger assumption. But essentially, these differences in bonus assumption only lightly affect the algorithm structure and its analysis.

**Definition 2.8** (Covering numbers of function classes)**.** For any $\epsilon > 0$, we define the following covering numbers of the involved function classes:

1. For each $h \in [H]$, there exists an $\epsilon$-cover $\mathcal{C}(\mathcal{F}_h, \epsilon) \subseteq \mathcal{F}_h$ with size $|\mathcal{C}(\mathcal{F}_h, \epsilon)| \leq \mathcal{N}(\mathcal{F}_h, \epsilon)$, such that for any $f \in \mathcal{F}$, there exists $f' \in \mathcal{C}(\mathcal{F}_h, \epsilon)$, such that $\|f - f'\|_\infty \leq \epsilon$. For any $\epsilon > 0$, we define the uniform covering number of $\mathcal{F}$ with respect to $\epsilon$ as $\mathcal{N}_{\mathcal{F}}(\epsilon) := \max_{h \in [H]} \mathcal{N}(\mathcal{F}_h, \epsilon)$.

2. There exists a bonus class $\mathcal{B} : \mathcal{S} \times \mathcal{A} \to \mathbb{R}$ such that for any $t \geq 0$, $z_{[t]} \in (\mathcal{S} \times \mathcal{A})^t$, $\sigma_{[t]} \in \mathbb{R}^t$, the oracle defined in Definition 2.6 $\bar{D}_{\mathcal{F}}(\cdot; z_{[t]}, \sigma_{[t]})$ is in $\mathcal{B}$.

3. For bonus class $\mathcal{B}$, there exists an $\epsilon$-cover $\mathcal{C}(\mathcal{B}, \epsilon) \subseteq \mathcal{B}$ with size $|\mathcal{C}(\mathcal{B}, \epsilon)| \leq \mathcal{N}(\mathcal{B}, \epsilon)$, such that for any $b \in \mathcal{B}$, there exists $b' \in \mathcal{C}(\mathcal{B}, \epsilon)$, such that $\|b - b'\|_\infty \leq \epsilon$.

**Remark 2.9.** In general function approximation, it is common to introduce the additional assumption on the covering number of bonus function classes. For example, in Ye et al. (2023), Agarwal & Zhang (2022a), and Di et al. (2023), the authors either assumed the bonus function class is finite or its covering number is bounded. Even when $\mathcal{F}$ is finite, the bonus function class $D_{\mathcal{F}}$ can still be very large. For example, consider $\mathcal{F} = \{f_1, f_2\}$ where $f_1$ is 1 on every state-action pair, and $f_2 = 2$ on every state-action pair. Then for $\epsilon = 1/4$, the covering number of $D_{\mathcal{F}}$ is at least the number of state-action pairs. This is because the bonus functions resulted from taking each state-action pair as historical data are far from each other in $l_\infty$ measure.

## 3 ALGORITHM

In this section, we will introduce our new algorithm, MQL-UCB. The detailed algorithm is provided in Algorithm 1, and we will proceed to elucidate the essential components of our method in the following subsections.

### 3.1 LOW POLICY-SWITCHING COST

For MQL-UCB algorithm, the value functions $Q_{k,h}, \check{Q}_{k,h}$, along with their corresponding policy $\pi_k$, undergo updates when the agent collects a sufficient number of trajectories within the dataset that could significantly diminish the uncertainty associated with the Bellman operator $\mathcal{T}_h V(\cdot, \cdot)$ through the weighted regression. In the context of linear bandit (Abbasi-Yadkori et al., 2011) or linear MDPs (He et al., 2022), the uncertainty pertaining to the least-square regression is quantified by the covariance matrix $\boldsymbol{\Sigma}_k$. In this scenario, the agent adjusts its policy once the determinant of the covariance matrix doubles, employing a determinant-based criterion. Nevertheless, in the general function approximation setting, such a method is not applicable in the absence of the covariance

matrix which serves as a feature extractor in the linear setting. Circumventing this issue, Kong et al. (2021) proposed a sub-sampling-based method to achieve low-switching properties in nonlinear RL. Their subsampling technique is inspired by Wang et al. (2021), which showed that one only needs to maintain a small subset of historical data to obtain a sufficiently accurate least-square estimator. Such a subset can be generated sequentially according to the *sensitivity* of a new coming data point. However, their approach leads to a switching cost of $\widetilde{O}(d^2 H)$, which does not match the lower bound in linear MDPs proposed by Gao et al. (2021).

To resolve this issue, we proposed a more general deterministic policy-updating framework for nonlinear RL. In detail, we use $\bar{D}^2_{\mathcal{F}_h}(z_{i,h}; z_{[k_{last}-1],h}, \bar{\sigma}_{[k_{last}-1],h})$ to evaluate the information collected at the episode $i$, given the last updating $k_{last}$. Once the collected information goes beyond a threshold $\chi$ from last updating, i,e,

$$\sum_{i \in [k_{last}, k-1]} \frac{1}{\bar{\sigma}^2_{i,h}} \bar{D}^2_{\mathcal{F}_h}(z_{i,h}; z_{[k_{last}-1],h}, \bar{\sigma}_{[k_{last}-1],h}) \geq \chi. \tag{3.1}$$

the agent will perform updates on both the optimistic estimated value function and the pessimistic value function. Utilizing the $D^2_{\mathcal{F}_h}$-based criterion, we will show that the number of policy updates can be bounded by $O(H \cdot \dim_{\alpha, K}(\mathcal{F}))$. This further reduces the complexity of the optimistic value function class and removes additional factors from a uniform convergence argument over the function class. Specifically, we showcase under our rare-switching framework, the $\epsilon$-covering number of the optimistic value function class and the pessimistic value function class at episode $k$ is bounded by

$$\mathcal{N}_\epsilon(k) := [\mathcal{N}_{\mathcal{F}}(\epsilon/2) \cdot \mathcal{N}(\mathcal{B}, \epsilon/2\widehat{\beta}_K)]^{l_k + 1} \tag{3.2}$$

where $\widehat{\beta}_K$ is the maximum confidence radius as shown in Algorithm 1, which will be specified in Lemmas F.3 and F.4.

### 3.2 EXECUTION PHASE

Our algorithm's foundational framework follows the Upper Confidence Bound (UCB) approach. In detail, for each episode $k \in [K]$, we construct an optimistic value function $Q_{k,h}(s, a)$ during the planning phase. Subsequently, in the exploration phase, the agent interacts with the environment, employing a greedy policy based on the current optimistic value function $Q_{k,h}(s, a)$. Once the agent obtains the reward $r^k_h$ and transitions to the next state $s^k_{h+1}$, these outcomes are incorporated into the dataset, contributing to the subsequent planning phase.

### 3.3 WEIGHTED REGRESSION

The estimation of $Q$ function in MQL-UCB extends LSVI-UCB proposed by Jin et al. (2020) to general function classes. While the estimators in LSVI-UCB are computed from the classic least squares regression, we construct the estimated value function $\widehat{f}_{k,h}$ for general function classes by solving the following weighted regression:

$$\widehat{f}_{k,h} = \operatorname*{argmin}_{f_h \in \mathcal{F}_h} \sum_{i \in [k-1]} \frac{1}{\bar{\sigma}^2_{i,h}} (f_h(s^i_h, a^i_h) - r^i_h - V_{k,h+1}(s^i_{h+1}))^2.$$

In the weighted regression, we set the weight $\bar{\sigma}_{k,h}$ as

$$\bar{\sigma}_{k,h} = \max \left\{ \sigma_{k,h}, \alpha, \gamma \cdot \bar{D}^{1/2}_{\mathcal{F}_h}(z; z_{[k-1],h}, \bar{\sigma}_{[k-1],h}) \right\},$$

where $\sigma_{k,h}$ is the estimated variance for the stochastic transition process, $\bar{D}_{\mathcal{F}_h}(z; z_{[k-1],h}, \bar{\sigma}_{[k-1],h})$ denotes the uncertainty of the estimated function $\widehat{f}_{k,h}$ conditioned on the historical observations and

$$\gamma := \sqrt{\log \frac{2HK^2 (2\log(L^2 k/\alpha^4) + 2) \cdot (\log(4L/\alpha^2) + 2) \cdot \mathcal{N}^4_{\mathcal{F}}(\epsilon) \cdot \mathcal{N}^2_\epsilon(K)}{\delta}} \tag{3.3}$$

is used to properly balance the uncertainty across various state-action pairs. It is worth noting that Ye et al. (2023) also introduced the uncertainty-aware variance in the general function approximation with a distinct intention to deal with the adversarial corruption from the attacker. In addition, according to the weighted regression, with high probability, the Bellman operator $\mathcal{T}_h V_{k,h}$ satisfies the following inequality,

$$\lambda + \sum_{i \in [k-1]} \bar{\sigma}^{-2}_{i,h} \left( \widehat{f}_{k,h}(s^i_h, a^i_h) - \mathcal{T}_h V_{k,h+1}(s^i_h, a^i_h) \right)^2 \leq \beta^2_k.$$

According to the definition of Generalized Eluder dimension, the estimation error between the estimated function $\widehat{f}_{k,h}$ and the Bellman operator is upper bounded by:

$$\left|\widehat{f}_{k,h}(s,a) - \mathcal{T}_h V_{k,h+1}(s,a)\right| \leq \widehat{\beta}_k D_{\mathcal{F}_h}(z; z_{[k-1],h}, \bar{\sigma}_{[k-1],h})$$

Therefore, we introduce the exploration bonus $b_{k,h}$ and construct the optimistic value function $Q_{k,h}(s,a)$,i.e., $Q_{k,h}(s,a) \approx \widehat{f}_{k,h}(s,a) + b_{k,h}(s,a)$, where $b_{k,h}(s,a) = \widehat{\beta}_k \cdot \bar{D}_{\mathcal{F}}\left((s,a); z_{[k-1],h}, \sigma_{[k-1],h}\right)$. Inspired by Hu et al. (2022); He et al. (2022); Agarwal et al. (2022), in order to estimate the gap between the optimistic value function $V_{k,h}(s)$ and the optimal value function $V_h^*(s)$, we further construct the pessimistic estimator $\check{f}_{k,h}$ by the following weighted regression

$$\check{f}_{k,h} = \operatorname*{argmin}_{f_h \in \mathcal{F}_h} \sum_{i \in [k-1]} \frac{1}{\check{\sigma}_{i,h}^2}(f_h(s_h^i, a_h^i) - r_h^i - \check{V}_{k,h+1}(s_{h+1}^i))^2,$$

and introduce a negative exploration bonus when generating the pessimistic estimator. $\check{Q}_{k,h}(s,a) \approx \check{f}_{k,h}(s,a) - \check{b}_{k,h}(s,a)$, where $\check{b}_{k,h}(s,a) = \check{\beta}_k \cdot \bar{D}_{\mathcal{F}}((s,a); z_{[k-1],h}, \sigma_{[k-1],h}$. Different from Agarwal et al. (2022), the pessimistic value function $\check{f}_{k,h}$ is computed from a similar weighted-regression scheme as in the case of the optimistic estimator, which enables us to derive a tighter confidence set.

**Algorithm 1** Monotonic Q-Learning with UCB (MQL-UCB)

---

**Require:** Regularization parameter $\lambda$, confidence radius $\{\widetilde{\beta}_k\}_{k \in [K]}$, $\{\widehat{\beta}_k\}_{k \in [K]}$ and $\{\check{\beta}_k\}_{k \in [K]}$.
1: Initialize $k_{\text{last}} = 0$
2: For each stage $h \in [H]$ and state-action $(s,a) \in \mathcal{S} \times \mathcal{A}$, set $Q_{0,h}(s,a) \leftarrow H, \check{Q}_{0,h}(s,a) \leftarrow 0$
3: **for** episodes $k = 1, \ldots, K$ **do**
4:      Received the initial state $s_1^k$.
5:      **for** stage $h = H, \ldots, 1$ **do**
6:         **if** there exists a stage $h' \in [H]$ such that (3.1) holds **then**
7:             $\widehat{f}_{k,h} \leftarrow \operatorname{argmin}_{f_h \in \mathcal{F}_h} \sum_{i \in [k-1]} \frac{1}{\widehat{\sigma}_{i,h}^2}(f_h(s_h^i, a_h^i) - r_h^i - V_{k,h+1}(s_{h+1}^i))^2$
8:             $\check{f}_{k,h} \leftarrow \operatorname{argmin}_{f_h \in \mathcal{F}_h} \sum_{i \in [k-1]} \frac{1}{\check{\sigma}_{i,h}^2}(f_h(s_h^i, a_h^i) - r_h^i - \check{V}_{k,h+1}(s_{h+1}^i))^2$
9:             $\widetilde{f}_{k,h} \leftarrow \operatorname{argmin}_{f_h \in \mathcal{F}_h} \sum_{i \in [k-1]} \left(f_h(s_h^i, a_h^i) - \left(r_h^i + V_{k,h+1}(s_{h+1}^i)\right)^2\right)^2$
10:           $Q_{k,h}(s,a) \leftarrow \min\left\{\widehat{f}_{k,h}(s,a) + b_{k,h}(s,a), Q_{k-1,h}(s,a), 1\right\}$
11:           $\check{Q}_{k,h}(s,a) \leftarrow \max\left\{\check{f}_{k,h}(s,a) - \check{b}_{k,h}(s,a), \check{Q}_{k-1,h}(s,a), 0\right\}$
12:           Set the last updating episode $k_{\text{last}} \leftarrow k$ and number of policies as $l_k \leftarrow l_{k-1} + 1$.
13:         **else**
14:           $Q_{k,h}(s,a) \leftarrow Q_{k-1,h}(s,a), \check{Q}_{k,h}(s,a) \leftarrow \check{Q}_{k-1,h}(s,a)$ and $l_k \leftarrow l_{k-1}$.
15:         **end if**
16:         Set the policy $\pi^k$ as $\pi_h^k(\cdot) \leftarrow \operatorname{argmax}_{a \in \mathcal{A}} Q_{k,h}(\cdot, a)$.
17:         $V_{k,h}(s) \leftarrow \max_a Q_{k,h}(s,a)$
18:         $\check{V}_{k,h}(s) \leftarrow \max_a \check{Q}_{k,h}(s,a)$
19:      **end for**
20:      **for** stage $h = 1, \ldots, H$ **do**
21:         Take action $a_h^k \leftarrow \pi_h^k(s_h^k)$
22:         Set the estimated variance $\sigma_{k,h}$ as in (3.4).
23:         $\bar{\sigma}_{k,h} \leftarrow \max\left\{\sigma_{k,h}, \alpha, \gamma \cdot D_{\mathcal{F}_h}^{1/2}(z; z_{[k-1],h}, \bar{\sigma}_{[k-1],h})\right\}$.
24:         Receive next state $s_{h+1}^k$
25:      **end for**
26: **end for**

---

### 3.4 VARIANCE ESTIMATOR

In this subsection, we provide more details about the variance estimator $\sigma_{k,h}$, which measures the variance of the value function $V_{k,h+1}(s_{h+1}^k)$ caused by the stochastic transition from state-action pair $(s_h^k, a_h^k)$. According to the definition of $\widehat{f}_{k,h}$, the difference between the estimator $\widehat{f}_{k,h}$ and $\mathcal{T}_h V_{k,h+1}$ satisfies

$$\lambda + \sum_{i \in [k-1]} \frac{1}{\bar{\sigma}_{i,h}^2} \left(\widehat{f}_{k,h}(s_h^i, a_h^i) - \mathcal{T}_h V_{k,h+1}(s_h^i, a_h^i)\right)^2$$

$$\leq 2 \sum_{i \in [k-1]} \frac{1}{\bar{\sigma}_{i,h}^2} \left( f(s_h^i, a_h^i) - \hat{f}_k^*(s_h^i, a_h^i) \right) \cdot \hat{\eta}_h^i(V_{k,h+1}),$$

where $\hat{\eta}_h^k(V) = \left( r_h^k + V(s_{h+1}^k) \right) - \mathbb{E}_{s' \sim \mathbb{P}_h(s_h^k, a_h^k)} \left[ r_h(s_h^k, a_h^k, s') + V(s') \right]$ denotes the stochastic transition noise for the value function $V$. However, the generation of the target function $V_{k,h+1}$ relies on previously collected data $z_{[k_{\text{last}}]}$, thus violating the conditional independence property. Consequently, the noise term $\hat{\eta}_h^k(V_{k,h+1})$ may not be unbiased. To address this challenge, it becomes imperative to establish a uniform convergence property over the potential function class, which is first introduced in linear MDPs by (Jin et al., 2020).

Inspired by the previous works (Azar et al., 2017; Hu et al., 2022; Agarwal et al., 2022; He et al., 2022), we decompose the noise of *optimistic* value function $\hat{\eta}_h^k(V_{k,h+1})$ into the noise of *optimal* value function $\hat{\eta}_h^k(V_{h+1}^*)$ and the noise $\hat{\eta}_h^k(V_{k,h+1} - V_{h+1}^*)$ to reduce the extra $\log(N_\epsilon(K))$ dependency in the confidence radius. With the noise decomposition, we evaluate the variances $[\mathbb{V}_h V_{h+1}^*](s, a)$ and $[\mathbb{V}_h(V_{k,h+1} - V_{h+1}^*)](s, a)$ separately.

For the variance of the *optimal* value function $[\mathbb{V}_h V_{h+1}^*](s, a)$, since the optimal value function $V_{h+1}^*$ is independent with the collected data $z_{[k_{\text{last}}]}$, it prevents a uniform convergence-based argument over the function class. However, the optimal value function $V_{h+1}^*$ is unobservable, and it requires several steps to estimate the variance. In summary, we utilize the optimistic function $V_{k,h+1}$ to approximate the optimal value function $V_{h+1}^*$ and calculate the estimated variance $[\bar{\mathbb{V}}_h V_{k,h}]$ as the difference between the second-order moment and the square of the first-order moment of $V_{k,h}$

$$[\bar{\mathbb{V}}_{k,h} V_{k,h+1}] = \bar{\mathcal{T}}_h^2 V_{k,h}(s, a) - \left( \bar{\mathcal{T}}_h V_{k,h+1}(s, a) \right)^2 = \tilde{f}_{k,h} - \hat{f}_{k,h}^2$$

Here, the approximate second-order moment $\tilde{f}_{k,h}$ and the approximate first-order moment $\hat{f}_{k,h}$ is generated by the least-square regression (Lines 7 and 9). In addition, we introduce the exploration bonus $E_{k,h}$ to control the estimation error between the estimated variance and the true variance of $V_{k,h+1}$ and $F_{k,h}$ to control the sub-optimality gap between $V_{k,h+1}$ and $V_{h+1}^*$:

$E_{k,h} = (2L\beta_k + \tilde{\beta}_k) \min \left( 1, \bar{D}_{\mathcal{F}_h}(z; z_{[k-1],h}, \bar{\sigma}_{[k-1],h}) \right),$

$F_{k,h} = \left( \log(\mathcal{N}(\mathcal{F}, \epsilon) \cdot \mathcal{N}_\epsilon(K)) \right) \cdot \min \left( 1, 2\hat{f}_{k,h}(s_h^k, a_h^k) - 2\check{f}_{k,h}(s_h^k, a_h^k) + 4\beta_k \bar{D}_{\mathcal{F}_h}(z; z_{[k-1],h}, \bar{\sigma}_{[k-1],h}) \right),$

where

$$\tilde{\beta}_k = \sqrt{128 \log \frac{\mathcal{N}_\epsilon(k) \cdot \mathcal{N}(\mathcal{F}, \epsilon) \cdot H}{\delta} + 64L\epsilon \cdot k}, \; \beta_k = \sqrt{128 \cdot \log \frac{\mathcal{N}_\epsilon(k) \cdot \mathcal{N}(\mathcal{F}, \epsilon) H}{\delta} + 64L\epsilon \cdot k/\alpha^2}.$$

For the variance of the sub-optimality gap, $[\mathbb{V}_h(V_{k,h+1} - V_{h+1}^*)](s, a)$, based on the structure of optimistic and pessimistic value function, it can be approximate and upper bounded by

$$[\mathbb{V}_h(V_{k,h+1} - V_{h+1}^*)](s_h^k, a_h^k) \leq 2[\mathbb{P}_h(V_{k,h+1} - V_{h+1}^*)](s_h^k, a_h^k)$$
$$\leq 2[\mathbb{P}_h(V_{k,h+1} - \check{V}_{k,h+1})](s_h^k, a_h^k) \approx 2\hat{f}_{k,h}(s_h^k, a_h^k) - 2\check{f}_{k,h}(s_h^k, a_h^k),$$

where the approximate first-order moments $\hat{f}_{k,h}, \check{f}_{k,h}$ are generated by the least-square regression (Lines 7 and 8) and can be dominated by the exploration bonus $F_{k,h}$.

In summary, we construct the estimated variance $\sigma_{k,h}$ as:

$$\sigma_{k,h} = \sqrt{[\bar{\mathbb{V}}_{k,h} V_{k,h+1}](s_h^k, a_h^k) + E_{k,h} + F_{k,h}}. \tag{3.4}$$

## 3.5 MONOTONIC VALUE FUNCTION

As we discussed in the previous subsection, we decompose the value function $V_{k,h}$ and evaluate the variance $[\mathbb{V}_h V_{h+1}^*](s, a)$, $[\mathbb{V}_h(V_{k,h+1} - V_{h+1}^*)](s, a)$ separately. However, for each state-action pair $(s_h^k, a_h^k)$ and any subsequent episode $i > k$, the value function $V_{i,h}$ and corresponding variance $[\mathbb{V}_h(V_{i,h+1} - V_{h+1}^*)](s_h^k, a_h^k)$ may differ from the previous value function $V_{k,h}$ and variance $[\mathbb{V}_h(V_{k,h+1} - V_{h+1}^*)](s_h^k, a_h^k)$. Inspired by He et al. (2022), we ensure that the pessimistic value function $\check{Q}_{k,h}$ maintains a monotonically increasing property during updates, while the optimistic value function $Q_{k,h}$ maintains a monotonically decreasing property. Leveraging these monotonic properties, we can establish an upper bound on the variance as follows:

$$[\mathbb{V}_h(V_{i,h+1} - V_{h+1}^*)](s_h^k, a_h^k) \leq 2[\mathbb{P}_h(V_{i,h+1} - \check{V}_{i,h+1})](s_h^k, a_h^k)$$
$$\leq 2[\mathbb{P}_h(V_{k,h+1} - \check{V}_{k,h+1})](s_h^k, a_h^k) \leq F_{k,h}.$$

In this scenario, the previously employed variance estimator $\sigma_{k,h}$ offers a consistent and uniform upper bound for the variance across all subsequent episodes.

## 4 MAIN RESULTS

In this section, we present our main results on the regret guarantee of MQL-UCB.

**Theorem 4.1.** Suppose Assumption 2.2 holds for function classes $\mathcal{F} := \{\mathcal{F}_h\}_{h=1}^H$ and Definition 2.4 holds with $\lambda = 1$. If we set $\alpha = 1/\sqrt{H}$, $\epsilon = (KLH)^{-1}$, and set $\widehat{\beta}_k = \widecheck{\beta}_k :=$
$\sqrt{O\left(\log \frac{2k^2(2\log(L^2k/\alpha^4)+2)\cdot(\log(4L/\alpha^2)+2)}{\delta/H}\right) \cdot [\log(\mathcal{N}_{\mathcal{F}}(\epsilon)) + 1]} + O(\lambda) + O(\epsilon kL/\alpha^2)$, then with high probability, the regret of MQL-UCB is upper bounded as follows:

$$
\text{Regret}(K) = \widetilde{O}\left(\sqrt{\dim(\mathcal{F})\log\mathcal{N} \cdot HK}\right)
$$
$$
+ \widetilde{O}\left(H^{2.5}\dim^2(\mathcal{F})\sqrt{\log\mathcal{N}}\log(\mathcal{N}\cdot\mathcal{N}_b)\cdot\sqrt{H\log\mathcal{N} + \dim(\mathcal{F})\log(\mathcal{N}\cdot\mathcal{N}_b)}\right),
$$

where we denote the covering number of bonus function class by $\mathcal{N}_b$, the covering number of function class $\mathcal{F}$ by $\mathcal{N}$, and the dimension $\dim_{\alpha,K}(\mathcal{F})$ by $\dim(\mathcal{F})$. At the same time, the switching cost of Algorithm 1 is bounded by $O(\dim_{\alpha,K}(\mathcal{F})\cdot H)$.

When the number of episodes $K$ is sufficiently large, the leading term in our regret bound is $\widetilde{O}\left(\sqrt{\dim(\mathcal{F})\log\mathcal{N}\cdot HK}\right)$. Our result matches the optimal regret achieved by Agarwal et al. (2022). While their proposed algorithm involves the execution of a complicated and non-Markovian policy with an action selection phase based on two series of optimistic value functions and the prefix trajectory, MQL-UCB only requires the knowledge of the current state and a single optimistic state-action value function $Q$ to make a decision over the action space.
As a direct application, we also present the regret guarantee of MQL-UCB for linear MDPs.

**Corollary 4.2.** Under the same conditions of Theorem 4.1, assume that the underlying MDP is a linear MDP such that $\mathcal{F} := \{\mathcal{F}_h\}_{h\in[H]}$ is composed of linear function classes with a known feature mapping over the state-action space $\phi : \mathcal{S} \times \mathcal{A} \to \mathbb{R}^d$. If we set $\lambda = 1$, $\alpha = 1/\sqrt{H}$, then with probability $1 - O(\delta)$, the following cumulative regret guarantee holds for MQL-UCB:

$$
\text{Regret}(K) = \widetilde{O}\left(d\sqrt{HK} + H^{2.5}d^5\sqrt{H + d^2}\right).
$$

**Remark 4.3.** The leading term in our regret bound, as demonstrated in Corollary 4.2, matches the lower bound proved in Zhou et al. (2021a) for linear MDPs. Similar optimal regrets have also been accomplished by He et al. (2022) and Agarwal et al. (2022) for linear MDPs. Since we also apply weighted regression to enhance the precision of our pessimistic value functions, the lower order term (i.e., $H^{2.5}d^5\sqrt{H + d^2}$) in our regret has a better dependency on $H$ than VOQL (Agarwal et al., 2022) and LSVI-UCB++ (He et al., 2022), which may be of independent interest when considering long-horizon MDPs. In addition, the switching cost of Algorithm 1 is bounded by $\widetilde{O}(dH)$, which matches the lower bound in Gao et al. (2021) for deterministic algorithms and our new lower bound in Theorem C.1 for arbitrary algorithms up to logarithmic factors. For more details about our lower bound, please refer to Appendix C

### 4.1 CONNECTION BETWEEN $D_{\mathcal{F}}^2$-UNCERTAINTY AND ELUDER DIMENSION

Previous work by Agarwal et al. (2022) achieved the optimal regret bound $\widetilde{O}\left(\sqrt{\dim(\mathcal{F})\log\mathcal{N}\cdot HK}\right)$, where $\dim(\mathcal{F})$ is defined as the generalized eluder dimension as stated in Definition 2.4. However, the connection between generalized eluder dimension and eluder dimension proposed by Russo & Van Roy (2013) is still under-discussed [2]. Consequently, their results could not be directly compared with the results based on the classic eluder dimension measure (Wang et al., 2020a) or the more general Bellman eluder dimension (Jin et al., 2021).
In this section, we make a first attempt to establish a connection between generalized eluder dimension and eluder dimension in Theorem 4.4.

**Theorem 4.4.** For a function space $\mathcal{G}$, $\alpha > 0$, let $\dim$ be defined in Definition 2.4. WLOG, we assume that for all $g \in \mathcal{G}$ and $z \in \mathcal{Z}$, $|g(z)| \leq 1$. Then the following inequality between

---

[2]Agarwal et al. (2022) (Remark 4) also discussed the relationship between the generalized eluder dimension and eluder dimension. However, there exists a technique flaw in the proof and we will discuss it in Appendix B.

$\dim(\mathcal{F}, \mathbf{Z}, \boldsymbol{\sigma})$ and $\dim_E(\mathcal{G}, 1/\sqrt{T})$ holds for all $\mathbf{Z} := \{z_i\}_{i \in [T]}$ with $z_i \in \mathcal{Z}$ and $\boldsymbol{\sigma} := \{\sigma_i\}_{i \in [T]}$ s.t. $\alpha \leq \sigma_i \leq M \ \forall i \in [T]$:

$$\sum_{i \in [T]} \min\left(1, \frac{1}{\sigma_i^2} D_{\mathcal{G}}^2\left(z_i; z_{[i-1]}, \sigma_{[i-1]}\right)\right) \leq O(\dim_E(\mathcal{F}, 1/\sqrt{T}) \log T \log \lambda T \log(M/\alpha) + \lambda^{-1}).$$

According to the Theorem 4.4, the generalized eluder dimension is upper bounded by eluder dimension up to logarithmic terms. When the number of episodes $K$ is sufficiently large, the leading term in our regret bound in Theorem 4.1 is $\widetilde{O}\left(\sqrt{\dim_E(\mathcal{F}) \log \mathcal{N} \cdot HK}\right)$, where $\dim_E(\mathcal{F})$ is the eluder dimension of the function class $\mathcal{F}$.

## 5 CONCLUSION AND FUTURE WORK

In this paper, we delve into the realm of RL with general function approximation. We proposed the MQL-UCB algorithm with an innovative uncertainty-based rare-switching strategy in general function approximation. Notably, our algorithm only require $\widetilde{O}(dH)$ updating times, which matches with the lower bound established by (Gao et al., 2021) up to logarithmic factors, and obtains a $\widetilde{O}(d\sqrt{HK})$ regret guarantee, which is near-optimal when restricted to the linear cases. Nonetheless, when dealing with function classes extending beyond linearity, the question arises as to whether our algorithm maintains its near-optimality. As part of our future endeavors, we aim to develop a theoretical lower bound that support the efficiency of our framework.

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

## A ADDITIONAL RELATED WORK

### A.1 RL WITH LINEAR FUNCTION APPROXIMATION

In recent years, a substantial body of research has emerged to address the challenges of solving Markov Decision Processes (MDP) with linear function approximation, particularly to handle the vast state and action spaces (Jiang et al., 2017; Dann et al., 2018; Yang & Wang, 2019; Du et al., 2019; Sun et al., 2019; Jin et al., 2020; Wang et al., 2020b; Zanette et al., 2020a; Yang & Wang, 2020; Modi et al., 2020; Ayoub et al., 2020; Zhou et al., 2021a; He et al., 2021a; Zhou & Gu, 2022; He et al., 2022; Zhao et al., 2023). These works can be broadly categorized into two groups based on the linear structures applied to the underlying MDP. One commonly employed linear structure is known as the linear MDP (Jin et al., 2020), where the transition probability function $\mathbb{P}_h$ and reward function $r_h$ are represented as linear functions with respect to a given feature mapping $\phi : \mathcal{S} \times \mathcal{A} \to \mathbb{R}^d$. Under this assumption, the LSVI-UCB algorithm (Jin et al., 2020) has been shown to achieve a regret guarantee of $\widetilde{O}(\sqrt{d^3 H^4 K})$. Subsequently, Zanette et al. (2020a) introduced the RLSVI algorithm, utilizing the Thompson sampling method, to attain a regret bound of $\widetilde{O}(\sqrt{d^4 H^5 K})$. More recently, He et al. (2022) improved the regret guarantee to $\widetilde{O}(\sqrt{d^2 H^3 K})$ with the LSVI-UCB++ algorithm, aligning with the theoretical lower bound in Zhou et al. (2021a) up to logarithmic factors. Another line of research has focused on linear mixture MDPs (Modi et al., 2020; Yang & Wang, 2020; Jia et al., 2020; Ayoub et al., 2020; Zhou et al., 2021a), where the transition probability is expressed as a linear combination of basic models $\mathbb{P}_1, \mathbb{P}_2, .., \mathbb{P}_d$. For linear mixture MDPs, Jia et al. (2020) introduced the UCRL-VTR algorithm, achieving a regret guarantee of $\widetilde{O}(\sqrt{d^2 H^4 K})$. Subsequently, Zhou et al. (2021a) enhanced this result to $\widetilde{O}(\sqrt{d^2 H^3 K})$, reaching a nearly minimax optimal regret bound. Recently, several works focused on time-homogeneous linear mixture MDPs, and removed the dependency on the episode length (horizon-free) (Zhang & Ji, 2019; Zhou et al., 2021b; Zhao et al., 2023).

### A.2 RL WITH GENERAL FUNCTION APPROXIMATION

Recent years have witnessed a flurry of progress on RL with nonlinear function classes. To explore the theoretical limits of RL algorithms, various complexity measures have been developed to characterize the hardness of RL instances such as Bellman rank (Jiang et al., 2017), Witness rank (Sun et al., 2019), eluder dimension (Russo & Van Roy, 2013), Bellman eluder dimension (Jin et al., 2021), Bilinear Classes (Du et al., 2021), Decision-Estimation Coefficient (Foster et al., 2021), Admissible Bellman Characterization (Chen et al., 2022), generalized eluder dimension (Agarwal et al., 2022). Among them, only Agarwal et al. (2022) yields a near-optimal regret guarantee when specialized to linear MDPs. In their paper, they proposed a new framework named generalized eluder dimension to handle the weighted objects in weighted regression, which can be seen as a variant of eluder dimension. In their proposed algorithm VOQL, they adopted over-optimistic and over-pessimistic value functions in order to bound the variance of regression targets, making it possible to apply a weighted regression scheme in the model-free framework.

### A.3 RL WITH LOW SWITCHING COST

Most of the aforementioned approaches necessitate updating both the value function and the corresponding policy in each episode, a practice that proves to be inefficient when dealing with substantial datasets. To overcome this limitation, a widely adopted technique involves dividing the time sequence into several epochs and updating the policy only between different epochs. In the context of the linear bandit problem, Abbasi-Yadkori et al. (2011) introduced the rarely-switching OFUL algorithm, where the agent updates the policy only when the determinant of the covariance matrix doubles. This method enables the algorithm to achieve near-optimal regret of $\widetilde{O}(\sqrt{K})$ while maintaining policy updates to $\widetilde{O}(d \log K)$ times. When the number of arms, denoted as $|\mathcal{D}|$, is finite, Ruan et al. (2021) proposed an algorithm with regret bounded by $\widetilde{O}(\sqrt{dK})$ and a mere $\widetilde{O}(d \log d \log K)$ policy-updating times. In the realm of episodic reinforcement learning, Bai et al. (2019) and Zhang et al. (2021) delved into tabular Markov Decision Processes, introducing algorithms that achieve a regret of $\widetilde{O}(\sqrt{T})$ and $\widetilde{O}(SA \log K)$ updating times. Wang et al. (2021) later extended these results to linear MDPs, unveiling the LSVI-UCB-RareSwitch algorithm. LSVI-UCB-RareSwitch delivers a regret bound of $\widetilde{O}(d^3 H^4 K)$ with a policy switching frequency of $\widetilde{O}(d \log T)$, which matches the lower bound of the switching cost (Gao et al., 2021) up to logarithmic terms. Furthermore, if the policy is updated within fixed-length epochs (Han et al., 2020), the method is termed batch learning model, but this falls beyond the scope of our current work. In addition, with the help of stochastic policies, Zhang et al. (2022) porposed an algorithm with $\widetilde{O}(\sqrt{K})$ regret guarantee and only $\widetilde{O}(H)$ swithcing cost for

learning tabular MDPs. Later, Huang et al. (2022) employed the stochastic policy in learning linear MDPs, which is able to find an $\epsilon$-optimal policy with only $\widetilde{O}(H)$ switching cost.

# B    DISCUSSION ABOUT THE $D_{\mathcal{F}}^2$-UNCERTAINTY AND ELUDER DIMENSION

To start with, we first recall the concept of eluder dimension as follows.

**Definition B.1** (Definition of eluder dimension, Russo & Van Roy 2013)**.** The eluder dimension of a function class $\mathcal{G}$ with domain $\mathcal{Z}$ is defined as follows:

- A point $z \in \mathcal{Z}$ is $\epsilon$-dependent on $z_1, z_2, \cdots, z_k \in \mathcal{Z}$ with respect to $\mathcal{G}$, if for all $g_1, g_2 \in \mathcal{G}$ such that $\sum_{i=1}^{k} [g_1(z_i) - g_2(z_i)]^2 \leq \epsilon$ it holds that $|g_1(z) - g_2(z)| \leq \epsilon$.
- Further $z$ is said to be $\epsilon$-independent of $z_1, z_2, \cdots, z_k$ with respect to $\mathcal{G}$, if $z$ is not dependent on $z_1, z_2, \cdots, z_k$.
- The eluder dimension of $\mathcal{G}$, denoted by $\dim_E(\mathcal{G}, \epsilon)$, is the length of the longest sequence of elements in $\mathcal{Z}$ such that every element is $\epsilon'$-independent of its predecessors for some $\epsilon' \geq \epsilon$.

With this definition, we can prove the Theorem 4.4.

*Proof of Theorem 4.4.* Let $\mathcal{I}_j$ $(1 \leq j \leq \lceil \log_2 M/\alpha \rceil)$ be the index set such that $\mathcal{I}_j = \left\{ t \in [T] | \sigma_t \in [2^{j-1} \cdot \alpha, 2^j \alpha] \right\}$.
Then we focus on the summation over $\mathcal{I}_j$ for each $j$. For simplicity, we denote the subsequence $\{z_i\}_{i \in \mathcal{I}_j}$ by $\{x_i\}_{i \in [|\mathcal{I}_j|]}$. Then we have

$$\sum_{i \in \mathcal{I}_j} \min\left(1, \frac{1}{\sigma_i^2} D_{\mathcal{G}}^2\left(z_i; z_{[i-1]}, \sigma_{[i-1]}\right)\right) \leq \sum_{i \in [|\mathcal{I}_j|]} \min\left(1, 4 D_{\mathcal{G}}^2\left(x_i; x_{[i-1]}, 1_{[i-1]}\right)\right)$$

To bound $\sum_{i \in [|\mathcal{I}_j|]} \min\left(1, 4 D_{\mathcal{G}}^2\left(x_i; x_{[i-1]}, 1_{[i-1]}\right)\right)$,

$$\sum_{i \in [|\mathcal{I}_j|]} \min\left(1, 4 D_{\mathcal{G}}^2\left(x_i; x_{[i-1]}, 1_{[i-1]}\right)\right)$$

$$\leq 4/\lambda + \sum_{i \in [|\mathcal{I}_j|]} 4 \int_{1/\lambda T}^1 \mathbb{1}\left\{D_{\mathcal{G}}^2\left(x_i; x_{[i-1]}, 1_{[i-1]}\right) \geq \rho\right\} \mathrm{d}\rho$$

$$\leq 4/\lambda + 4 \int_{1/\lambda T}^1 \sum_{i \in [|\mathcal{I}_j|]} \mathbb{1}\left\{D_{\mathcal{G}}^2\left(x_i; x_{[i-1]}, 1_{[i-1]}\right) \geq \rho\right\} \mathrm{d}\rho \tag{B.1}$$

Then we proceed by bounding the summation $\sum_{i \in [|\mathcal{I}_j|]} \mathbb{1}\left\{D_{\mathcal{G}}^2\left(x_i; x_{[i-1]}, 1_{[i-1]}\right) \geq \rho\right\}$ for each $\rho \geq 1/(\lambda T)$. For short, let $d := \dim_E(\mathcal{G}, 1/\sqrt{T})$. Essentially, it suffices to provide an upper bound of the cardinality of the subset $\mathcal{J}_j := \left\{i \in \mathcal{I}_j | D_{\mathcal{G}}^2\left(x_i; x_{[i-1]}, 1_{[i-1]}\right) \geq \rho\right\}$.
For each $i \in \mathcal{J}_j$, since $D_{\mathcal{G}}^2\left(x_i; x_{[i-1]}, 1_{[i-1]}\right) \geq \rho$, there exists $g_1, g_2$ in $\mathcal{G}$ such that $\frac{(g_1(x_i) - g_2(x_i))^2}{\sum_{t \in [i-1]}(g_1(x_t) - g_2(x_t))^2 + \lambda} \geq \rho/2$. Here $(g_1(x_i) - g_2(x_i))^2 \geq \lambda \rho \geq 1/T$. Denote such value of $(g_1(x_i) - g_2(x_i))^2$ by $L(x_i)$. Then we consider split $cJ_j$ into $\lceil \log_2 T \rceil$ layers such that in each layer $\mathcal{J}_j^k$ $(k \in [\lceil \log_2 T \rceil])$, we have $1/T \leq \xi \leq L(x_i) \leq 2\xi$ for some $\xi$.
Denote the cardinality of $\mathcal{J}_j^k$ by $A$ and the subsequence $\{x_i\}_{i \in \mathcal{I}_j^k}$ by $\{y_i\}_{i \in [A]}$. For the elements in $\{y\}$, we dynamically maintain $\lfloor A/d \rfloor$ queues of elements. We enumerate through $\{y\}$ in its original order and put the current element $y_i$ into the queue $Q$ such that $y_i$ is $\xi$-independent of the current elements in $Q$. By the Pigeonhole principle and the definition of eluder dimension $d$, we can find an element $y_i$ in $\{y\}$ such that $y_i$ is $\xi$-dependent of all the $\lfloor A/d \rfloor$ queues before $i$.
Then by the definition of $L(y_i)$ and $\xi$, we can choose $g_1, g_2$ such that

$$\frac{(g_1(y_i) - g_2(y_i))^2}{\sum_{t \in [i-1]}(g_1(y_t) - g_2(y_t))^2 + \lambda} \geq \rho/2, \quad 2\xi \geq (g_1(y_t) - g_2(y_t))^2 \geq \xi. \tag{B.2}$$

By the $\xi$-dependencies, we have $\sum_{t \in [i-1]}(g_1(y_t) - g_2(y_t))^2 \geq \lfloor A/d \rfloor \cdot \xi$. At the same time $\sum_{t \in [i-1]}(g_1(y_t) - g_2(y_t))^2 \leq 4\xi/\rho$ due to (B.2).
Thus, we deduce that $A = |\mathcal{J}_j^k| \leq O(d/\rho)$ for all $k \in [\lceil \log_2 T \rceil]$.

Substituting it into (B.1), we have

$$\sum_{i\in[|\mathcal{I}_j|]}\min\left(1,4D_{\mathcal{G}}^2\left(x_i;x_{[i-1]},1_{[i-1]}\right)\right)\leq 4/\lambda+4\int_{1/\lambda T}^{1}O(d/\rho\cdot\log_2 T)\mathrm{d}\rho$$

$$=O(d\log_2 T\cdot\log\lambda T+\lambda^{-1}).$$

Here $j$ is chosen arbitrarily. Hence, the generalized eluder dimension can be further bounded as follows:

$$\sum_{i\in[T]}\min\left(1,\frac{1}{\sigma_i^2}D_{\mathcal{G}}^2\left(z_i;z_{[i-1]},\sigma_{[i-1]}\right)\right)\leq O(\dim_E(\mathcal{F},1/\sqrt{T})\log T\log\lambda T\log(M/\alpha)+\lambda^{-1}).$$

$\square$

In the following part, we will also discuss the issue in Agarwal et al. (2022) about the relationship between the standard eluder dimension $\dim_E$ and the generalized eluder dimension $\dim$. In detail, Agarwal et al. (2022) proposed the concept of Generalized Eluder dimension and made the following claim:

**Lemma B.2** (Remark 4, Agarwal et al. 2022). *If we set the weight $\boldsymbol{\sigma}=1$, then the Generalized Eluder dimension $\dim=\sup_{\mathbf{Z},\boldsymbol{\sigma}:|Z|=T}\dim(\mathcal{F},\mathbf{Z},1)$ satisfied*

$$\dim\leq\dim_E(\mathcal{F},\sqrt{\lambda/T})+1,$$

where $\dim_E$ denotes the standard Eluder dimension proposed in Russo & Van Roy (2013). In the proof of Remark 4, Agarwal et al. (2022) claims that given the standard Eluder dimension $\dim_E(\mathcal{F},\sqrt{\lambda})=n$, there are at most $n$ different (sorted) indices $\{t_1,t_2,..\}$ such that $D_{\mathcal{F}}^2(z_{t_i};z_{t_{i-1}},1)\geq\epsilon^2/\lambda$. However, according to the definition of $D_{\mathcal{F}}^2$-uncertainty, we only have

$$D_{\mathcal{F}}^2(z;z_{[t-1]},1):=\sup_{f_1,f_2\in\mathcal{F}}\frac{(f_1(z)-f_2(z))^2}{\sum_{s\in[t-1]}(f_1(z_s)-f_2(z_s))^2+\lambda}.$$

However, $z_{t_i}$ is $\epsilon$-dependence with $z_1,..,z_{t_i-1}$ is only a sufficient condition for the uncertainty $D_{\mathcal{F}}^2(z;z_{[t-1]},1)$ rather than necessary condition. In cases where both $(f_1(z)-f_2(z))^2\geq\epsilon^2$ and $\sum_{s\in[t-1]}(f_1(z_s)-f_2(z_s))^2\geq\epsilon^2$ hold, the uncertainty $D_{\mathcal{F}}^2(z;z_{[t-1]},1)$ may exceed $\epsilon^2/\lambda$, yet it will not be counted within the longest $\epsilon$-independent sequence for the standard Eluder dimension.

## C  LOWER BOUND

In this section, we prove a new lower bound on the switching cost for RL with linear MDPs. Note that linear MDPs belong to the class of MDPs studied in our paper with bounded generalized eluder dimension. In particular, the generalized eluder dimension of linear MDPs is $\widetilde{O}(d)$.

**Theorem C.1.** For any algorithm **Alg** with expected switching cost less than $dH/(16\log K)$, there exists a hard-to-learn linear MDP, such that the expected regret of **Alg** is at least $\Omega(K)$.

**Remark C.2.** Theorem C.1 suggests that, to achieve a sublinear regret guarantee, an $\widetilde{\Omega}(dH)$ switching cost is unavoidable. This lower bound does not violate the upper bound/lower bound of $\widetilde{O}(H)$ proved in Zhang et al. (2022); Huang et al. (2022), which additionally assume that the initial state $s_1^k$ is either fixed or sampled from a fixed distribution. In contrast, our work and Gao et al. (2021) allow the initial state to be adaptively chosen by an adversarial environment, rendering the learning of linear MDPs more challenging. When comparing our lower bound with the result in Gao et al. (2021), it is worth noting that their focus is solely on deterministic algorithms, and they suggest that an $\widetilde{\Omega}(dH)$ switching cost is necessary. As a comparison, our result holds for any algorithm with arbitrary policies including both deterministic and stochastic policies.

To prove the lower bound, we create a series of hard-to-learn MDPs as follows. Each hard-to-learn MDP comprises $d/4$ distinct sub-MDPs denoted as $\mathcal{M}_1,..,\mathcal{M}_{d/4}$. Each sub-MDP $\mathcal{M}_i$ is characterized by two distinct states, initial state $s_{i,0}$ and absorbing state $s_{i,1}$, and shares the same action set $\mathcal{A}=\{a_0,a_1\}$. Since the state and action spaces are finite, these tabular MDPs can always be represented as linear MDPs with dimension $|\mathcal{S}|\times|\mathcal{A}|=d$.

To generate each sub-MDP $\mathcal{M}_i$, for all stage $h \in [H]$, a special action $a_{i,h}$ is uniformly randomly selected from the action set $\{a_0, a_1\}$. Given the current state $s_{i,0}$, the agent transitions to the state $s_{i,0}$ if it takes the special action $a_{i,h}$. Otherwise, the agent transitions to the absorbing state $s_{i,1}$ and remains in that state in subsequent stages. The agent will receive the reward $1$ if it takes the special action $a_{i,H}$ at the state $s_{i,0}$ during the last stage $H$. Otherwise, the agent always receives reward $0$. In this scenario, for sub-MDP $\mathcal{M}_i$, the optimal policy entails following the special action sequence $(a_{i,1}, a_{i,2}, ..., a_{i,H})$ to achieve a total reward of $1$. In contrast, any other action sequence fails to yield any reward.

Now, we partition the $K$ episodes to $d/4$ different distinct epochs. For each epoch (ranging from episodes $4(i-1)K/d + 1$ to episode $4iK/d$), we initialize the state as $s_{i,0}$ and exclusively focus on the sub-MDP $\mathcal{M}_i$. The regret in each epoch can be lower bounded separately as follows:

**Lemma C.3.** For each epoch $i \in [d/4]$ and any algorithm **Alg** capable of deploying arbitrary policies, if the expected switching cost in epoch $i$ is less than $H/(2 \log K)$, the expected regret of **Alg** in the $i$-th epoch is at least $\Omega(K/d)$.

*Proof of Lemma C.3.* Given that each sub-MDP is independently generated, policy updates before epoch $i$ only offer information for the sub-MDPs $\mathcal{M}_1, ..., \mathcal{M}_{i-1}$ and do not provide any information for the current epoch $i$. In this scenario, there is no distinction between epochs and for simplicity, we only focus on the first epoch, encompassing episodes $1$ to $4K/d$.

Now, let $k_0 = 0$ and we denote $\mathcal{K} = \{k_1, k_2, ...\}$ as the set of episodes where the algorithm **Alg** updates the policy. If **Alg** does not update the policy $i$ times, we set $k_i = 4K/d + 1$. For simplicity, we set $C = 2 \log K$ and for each $i \leq H/C$, we define the events $\mathcal{E}$ as the algorithm **Alg** has not reached the state $s_{1,0}$ at the stage $iC$ before the episode $k_i$.

Conditioned on the events $\mathcal{E}_1, ..., \mathcal{E}_{i-1}$, the algorithm **Alg** does not gather any information about the special action $a_{1,h}$ for stage $h \geq (i-1)C + 1$. In this scenario, the special actions can still be considered as uniformly randomly selected from the action set $a_0, a_1$. For each episode between $k_{i-1}$ and $k_i$, the probability that a policy $\pi$ arrives at state $s_{1,0}$ at stage $iC$ is upper-bounded by:

$$\mathbb{E}_{a_{1,(i-1)C+1}, ..., a_{1,iC}} \big[ \Pi_{h=1}^{iC} \Pr(\pi_h(s_{1,0}) = a_{1,h}) \big]$$

$$\leq \mathbb{E}_{a_{1,(i-1)C+1}, ..., a_{1,iC}} \big[ \Pi_{h=(i-1)C+1}^{iC} \Pr(\pi_h(s_{1,0}) = a_{1,h}) \big] = \frac{1}{2^C},$$

where the first inequality holds due to $\pi(s_{1,0}) = a_{1,h}) \leq 1$ and the second equation holds due to the random generation process of the special actions. Notice that there are at most $K$ episodes between the $k_{i-1}$ and $k_i$ and after applying an union bound for all episodes, we have

$$\Pr(\mathcal{E}_i | \mathcal{E}_1, ..., \mathcal{E}_{i-1}) \leq 1 - \frac{K}{2^C} = 1 - \frac{1}{K}.$$

Furthermore, we have

$$\Pr(\mathcal{E}_{H/C}) \leq \Pr(\mathcal{E}_1 \cap \mathcal{E}_2 \cap \mathcal{E}_3 \cap ... \cap \mathcal{E}_{H/C})$$

$$= \Pi_{i=1}^{H/C} \Pr(\mathcal{E}_i | \mathcal{E}_1, ..., \mathcal{E}_{i-1})$$

$$\leq (1 - \frac{1}{1K})^{H/C} \leq 1 - \frac{H}{CK}. \tag{C.1}$$

Notice that the agent cannot receive any reward unless it has reached the state $s_{i,0}$ at the last stage $H$. Therefore, the expected regret for the algorithm **Alg** for the sub-MDP $\mathcal{M}$ can be bounded by the switching cost $\delta$:

$$\mathbb{E}_{\mathcal{M}}[\text{Regret}(\textbf{Alg})] \geq \mathbb{E}_{\mathcal{M}}[\mathbb{1}(\mathcal{E}_{H/C}) \times (k_{H/C} - 1)]$$

$$\geq \mathbb{E}_{\mathcal{M}}[k_{H/C} - 1] - \frac{H}{C}$$

$$\geq \mathbb{E}_{\mathcal{M}} \big[ \mathbb{1}(\delta < H/C) \cdot K/d \big] - \frac{H}{C}$$

$$\geq 4K/d - \mathbb{E}_{\mathcal{M}}[\delta] \cdot \frac{4KC}{dH} - \frac{H}{C}, \tag{C.2}$$

where the first inequality holds due to the fact that the agent receives no reward before $k_{H/C}$ conditioned on the event $\mathcal{E}_{H/C}$, the second inequality holds due to (C.1) with $k_{H/C} - 1 \leq K$, the

third inequality holds due to the definition of $k_{H/C}$ and the last inequality holds due to $\mathbb{E}[\mathbb{1}(x \geq a)] \leq \mathbb{E}[x]/a$ for any non-negative random variable $x$. According to the result in (C.1), if the expected switching $\mathbb{E}[\delta] \leq H/(2C)$, the expected regret is at least $\Omega(K)$, when $K$ is large enough compared to $H$. $\qquad\square$

With the help of Lemma C.1, we can prove Theorem C.1.

*Proof of Theorem C.1.* For these constructed hard-to-learn MDPs and any given algorithm **Alg**, we denote the expected switching cost for sub-MDP $\mathcal{M}_i$ as $\delta_i$.
According to Lemma C.3, we have

$$\begin{aligned}
\mathbb{E}_{\mathcal{M}_i}[\text{Regret}(\mathbf{Alg})] &\geq \Pr\left(\delta_i < H/(2\log K)\right) \cdot K/d \\
&\geq \left(1 - \mathbb{E}_{\mathcal{M}_i}[\delta_i] \cdot 2\log K/H\right) \cdot K/d,
\end{aligned} \tag{C.3}$$

where the first inequality holds due to lemma C.1 and laast inequality holds due to $\mathbb{E}[\mathbb{1}(x \geq a)] \leq \mathbb{E}[x]/a$ for any non-negative random variable $x$. Taking a summation of (C.3) for all sub-MDps, we have

$$\begin{aligned}
\mathbb{E}_{\mathcal{M}}[\text{Regret}(\mathbf{Alg})] &= \sum_{i=1}^{d/4} \mathbb{E}_{\mathcal{M}_i}[\text{Regret}(\mathbf{Alg})] \\
&\geq \sum_{i=1}^{d/4} \left(1 - \mathbb{E}_{\mathcal{M}_i}[\delta_i] \cdot 2\log K/H\right) \cdot K/d \\
&= (d/4 - \mathbb{E}_{\mathcal{M}}[\delta] \cdot 2\log K/H) \cdot K/d,
\end{aligned}$$

where $\delta$ is the total expected switching cost. Therefore, for any algorithm with total expected switching cost less than $dH/(16\log K)$, the expected is lower bounded by

$$\mathbb{E}_{\mathcal{M}}[\text{Regret}(\mathbf{Alg})] \geq (d/4 - \mathbb{E}_{\mathcal{M}}[\delta] \cdot 2\log K/H) \cdot K/d \geq K/8.$$

Thus, we finish the proof of Theorem C.1. $\qquad\square$

# D  PROOF OF THEOREM 4.1

In this section, we provide the proof of Theorem 4.1.

## D.1  HIGH PROBABILITY EVENTS

In this subsection, we define the following high-probability events:

$$\underline{\mathcal{E}}_{k,h}^{\widetilde{f}} = \left\{ \lambda + \sum_{i \in [k-1]} \left( \widetilde{f}_{k,h}(s_h^i, a_h^i) - \mathcal{T}_h^2 V_{k,h+1}(s_h^i, a_h^i) \right)^2 \leq \widetilde{\beta}_k^2 \right\}, \tag{D.1}$$

$$\underline{\mathcal{E}}_{k,h}^{\widehat{f}} = \left\{ \lambda + \sum_{i \in [k-1]} \frac{1}{\overline{\sigma}_{i,h}^2} \left( \widehat{f}_{k,h}(s_h^i, a_h^i) - \mathcal{T}_h V_{k,h+1}(s_h^i, a_h^i) \right)^2 \leq \beta_k^2 \right\}, \tag{D.2}$$

$$\underline{\mathcal{E}}_{k,h}^{\check{f}} = \left\{ \lambda + \sum_{i \in [k-1]} \frac{1}{\overline{\sigma}_{i,h}^2} \left( \check{f}_{k,h}(s_h^i, a_h^i) - \mathcal{T}_h \check{V}_{k,h+1}(s_h^i, a_h^i) \right)^2 \leq \beta_k^2 \right\}, \tag{D.3}$$

where

$$\widetilde{\beta}_k := \sqrt{128 \log \frac{\mathcal{N}_\epsilon(k) \cdot \mathcal{N}_{\mathcal{F}}(\epsilon) \cdot H}{\delta} + 64L\epsilon \cdot k}, \quad \beta_k := \sqrt{128 \cdot \log \frac{\mathcal{N}_\epsilon(k) \cdot \mathcal{N}_{\mathcal{F}}(\epsilon)H}{\delta} + 64L\epsilon \cdot k/\alpha^2}.$$

$\underline{\mathcal{E}}_{k,h}^{\widetilde{f}}$, $\underline{\mathcal{E}}_{k,h}^{\widehat{f}}$ and $\underline{\mathcal{E}}_{k,h}^{\check{f}}$ are the hoeffding-type concentration results for $\widetilde{f}_{k,h}$, $\widehat{f}_{k,h}$ and $\check{f}_{k,h}$ respectively.

Then we define the following bellman-type concentration events for $\widehat{f}_{k,h}$, $\check{f}_{k,h}$, which implies a tighter confidence set due to carefully designed variance estimators and an inverse-variance weighted regression scheme for the general function classes.

$$\mathcal{E}_h^{\widehat{f}} = \left\{ \lambda + \sum_{i=1}^{k-1} \frac{1}{(\overline{\sigma}_{i,h'})^2} \left( \widehat{f}_{k,h'}(s_{h'}^i, a_{h'}^i) - \mathcal{T}_{h'} V_{k,h'+1}(s_{h'}^i, a_{h'}^i) \right)^2 \leq \widehat{\beta}_k^2, \ \forall h \leq h' \leq H, k \in [K] \right\},$$

$$\mathcal{E}_h^{\check{f}} = \left\{ \lambda + \sum_{i=1}^{k-1} \frac{1}{(\bar{\sigma}_{i,h'})^2} \left( \check{f}_{k,h'}(s_{h'}^i, a_{h'}^i) - \mathcal{T}_{h'} V_{k,h'+1}(s_{h'}^i, a_{h'}^i) \right)^2 \leq \check{\beta}_k^2, \ \forall h \leq h' \leq H, k \in [K] \right\},$$
(D.4)

where

$$\widehat{\beta}_k^2 = \check{\beta}_k^2 := O\left( \log \frac{2k^2 \left( 2\log(L^2 k/\alpha^4) + 2 \right) \cdot \left( \log(4L/\alpha^2) + 2 \right)}{\delta/H} \right) \cdot \left[ \log(\mathcal{N}_{\mathcal{F}}(\epsilon)) + 1 \right]$$
$$+ O(\lambda) + O(\epsilon k L/\alpha^2).$$

We also define the following events which are later applied to prove the concentration of $\widehat{f}_{k,h}$ and $\check{f}_{k,h}$ by induction.

$$\bar{\mathcal{E}}_{k,h}^{\widehat{f}} = \left\{ \lambda + \sum_{i \in [k-1]} \frac{\widehat{\mathbb{1}}_{i,h}}{(\bar{\sigma}_{i,h})^2} \left( \widehat{f}_{k,h}(s_h^i, a_h^i) - \mathcal{T}_h V_{k,h+1}(s_h^i, a_h^i) \right)^2 \leq \widehat{\beta}_k^2 \right\},$$
(D.5)

where $\widehat{\mathbb{1}}_{i,h}$ is the shorthand for the following product of indicator functions

$$\widehat{\mathbb{1}}_{i,h} := \mathbb{1}\left( [\mathbb{V}_h V_{h+1}^*](s_h^i, a_h^i) \leq \bar{\sigma}_{i,h}^2 \right) \cdot \mathbb{1}\left( [\mathcal{T}_h V_{i,h+1} - \mathcal{T} V_{h+1}^*] \leq (\log \mathcal{N}_{\mathcal{F}}(\epsilon) + \log \mathcal{N}_\epsilon(K))^{-1} \bar{\sigma}_{i,h}^2 \right)$$
$$\cdot \mathbb{1}\left( V_{i,h+1}(s) \geq V_{h+1}^*(s) \quad \forall s \in \mathcal{S} \right).$$
(D.6)

$$\bar{\mathcal{E}}_{k,h}^{\check{f}} = \left\{ \lambda + \sum_{i \in [k-1]} \frac{\check{\mathbb{1}}_{i,h}}{(\bar{\sigma}_{i,h})^2} \left( \check{f}_{k,h}(s_h^i, a_h^i) - \mathcal{T}_h \check{V}_{k,h+1}(s_h^i, a_h^i) \right)^2 \leq \check{\beta}_k^2 \right\},$$
(D.7)

where $\check{\mathbb{1}}_{i,h}$ is the shorthand for the following product of indicator functions

$$\check{\mathbb{1}}_{i,h} := \mathbb{1}\left( [\mathbb{V}_h V_{h+1}^*](s_h^i, a_h^i) \leq \bar{\sigma}_{i,h}^2 \right) \cdot \mathbb{1}\left( [\mathcal{T} V_{h+1}^* - \mathcal{T}_h \check{V}_{i,h+1}] \leq (\log \mathcal{N}_{\mathcal{F}}(\epsilon) + \log \mathcal{N}_\epsilon(K))^{-1} \bar{\sigma}_{i,h}^2 \right)$$
$$\cdot \mathbb{1}\left( \check{V}_{i,h+1}(s) \leq V_{h+1}^*(s) \quad \forall s \in \mathcal{S} \right).$$
(D.8)

The following Lemmas suggest that previous events holds with high probability.

**Lemma D.1.** Let $\widetilde{f}_{k,h}$ be defined as in line 9 of Algorithm 1, we have $\underline{\mathcal{E}}^{\widetilde{f}} := \bigcap_{k \geq 1, h \in [H]} \underline{\mathcal{E}}_{k,h}^{\widetilde{f}}$ holds with probability at least $1 - \delta$, where $\underline{\mathcal{E}}_{k,h}^{\widetilde{f}}$ is defined in (D.1).

**Lemma D.2.** Let $\widehat{f}_{k,h}$ be defined as in line 7 of Algorithm 1, we have $\underline{\mathcal{E}}^{\widehat{f}} := \bigcap_{k \geq 1, h \in [H]} \underline{\mathcal{E}}_{k,h}^{\widehat{f}}$ holds with probability at least $1 - \delta$, where $\underline{\mathcal{E}}_{k,h}^{\widehat{f}}$ is defined in (D.2).

**Lemma D.3.** Let $\check{f}_{k,h}$ be defined as in line 8 of Algorithm 1, we have $\underline{\mathcal{E}}^{\check{f}} := \bigcap_{k \geq 1, h \in [H]} \underline{\mathcal{E}}_{k,h}^{\check{f}}$ holds with probability at least $1 - \delta$, where $\underline{\mathcal{E}}_{k,h}^{\check{f}}$ is defined in (D.3).

**Lemma D.4.** Let $\widehat{f}_{k,h}$ be defined as in line 7 of Algorithm 1, we have $\bar{\mathcal{E}}^{\widehat{f}} := \bigcap_{k \geq 1, h \in [H]} \bar{\mathcal{E}}_{k,h}^{\widehat{f}}$ holds with probability at least $1 - 2\delta$, where $\mathcal{E}_{k,h}^{\widehat{f}}$ is defined in (D.5).

**Lemma D.5.** Let $\check{f}_{k,h}$ be defined as in line $\cdot$ of Algorithm 1, we have $\bar{\mathcal{E}}^{\check{f}}$ holds with probability at least $1 - 2\delta$.

### D.2 OPTIMISM AND PESSIMISM

Based on the high probability events, we have the folloing lemmas for the function $\widehat{f}_{k,h}(s, a)$ and $\check{f}_{k,h}(s, a)$

**Lemma D.6.** On the events $\mathcal{E}_h^{\widehat{f}}$ and $\mathcal{E}_h^{\check{f}}$, for eack episode $k \in [K]$, we have

$$\left| \widehat{f}_{k,h}(s, a) - \mathcal{T}_h V_{k,h+1}(s, a) \right| \leq \widehat{\beta}_k D_{\mathcal{F}_h}(z; z_{[k-1],h}, \bar{\sigma}_{[k-1],h})$$
$$\left| \check{f}_{k,h}(s, a) - \mathcal{T}_h \check{V}_{k,h+1}(s, a) \right| \leq \check{\beta}_k D_{\mathcal{F}_h}(z; z_{[k-1],h}, \bar{\sigma}_{[k-1],h}),$$

where $z = (s, a)$ and $z_{[k-1],h} = \{z_{1,h}, z_{2,h}, .., z_{k-1,h}\}$.

**Lemma D.7.** On the events $\mathcal{E}_{h+1}^{\hat{f}}$ and $\mathcal{E}_{h+1}^{\check{f}}$, for each stage $h \leq h' \leq H$ and episode $k \in [K]$, we have $Q_{k,h}(s,a) \geq Q_h^*(s,a) \geq \check{Q}_{k,h}(s,a)$. Furthermore, for the value functions $V_{k,h}(s)$ and $\check{V}_{k,h}(s)$, we have $V_{k,h}(s) \geq V_h^*(s) \geq \check{V}_{k,h}(s)$.

### D.3   MONOTONIC VARIANCE ESTIMATOR

In this subsection, we introduce the following lemmas to construct the monotonic variance estimator.

**Lemma D.8.** On the events $\underline{\mathcal{E}}^{\tilde{f}}$, $\underline{\mathcal{E}}^{\hat{f}}$ and $\underline{\mathcal{E}}^{\check{f}}$, for eack episode $k \in [K]$ and stage $h \in [H]$, we have

$$\left|\hat{f}_{k,h}(s,a) - \mathcal{T}_h V_{k,h+1}(s,a)\right| \leq \beta_k D_{\mathcal{F}_h}(z; z_{[k-1],h}, \bar{\sigma}_{[k-1],h})$$
$$\left|\check{f}_{k,h}(s,a) - \mathcal{T}_h \check{V}_{k,h+1}(s,a)\right| \leq \beta_k D_{\mathcal{F}_h}(z; z_{[k-1],h}, \bar{\sigma}_{[k-1],h})$$
$$\left|\tilde{f}_{k,h}(s,a) - \mathcal{T}_h^2 V_{k,h+1}(s,a)\right| \leq \tilde{\beta}_k D_{\mathcal{F}_h}(z; z_{[k-1],h}, \bar{\sigma}_{[k-1],h})$$

where $z = (s,a)$ and $z_{[k-1],h} = \{z_{1,h}, z_{2,h}, .., z_{k-1,h}\}$.

**Lemma D.9.** On the events $\underline{\mathcal{E}}^{\tilde{f}}$, $\underline{\mathcal{E}}^{\hat{f}}$, $\underline{\mathcal{E}}^{\check{f}}$, $\mathcal{E}_{h+1}^{\hat{f}}$, $\mathcal{E}_{h+1}^{\check{f}}$, for each episode $k \in [K]$, the empirical variance $[\bar{\mathbb{V}}_h V_{k,h+1}](s_h^k, a_h^k)$ satisfies the following inequalities:

$$\left|[\bar{\mathbb{V}}_h V_{k,h+1}](s_h^k, a_h^k) - [\mathbb{V}_h V_{k,h+1}](s_h^k, a_h^k)\right| \leq E_{k,h},$$
$$\left|[\bar{\mathbb{V}}_h V_{k,h+1}](s_h^k, a_h^k) - [\mathbb{V}_h V_{h+1}^*](s_h^k, a_h^k)\right| \leq E_{k,h} + F_{k,h}.$$

**Lemma D.10.** On the events $\underline{\mathcal{E}}^{\hat{f}}$, $\underline{\mathcal{E}}^{\check{f}}$, $\mathcal{E}_{h+1}^{\hat{f}}$, $\mathcal{E}_{h+1}^{\check{f}}$, for each episode $k \in [K]$ and $i > k$, we have

$$(\log \mathcal{N}_{\mathcal{F}}(\epsilon) + \log \mathcal{N}_\epsilon(K)) \cdot \left[\mathbb{V}_h(V_{i,h+1} - V_{h+1}^*)\right](s_h^k, a_h^k) \leq F_{i,h}$$
$$(\log \mathcal{N}_{\mathcal{F}}(\epsilon) + \log \mathcal{N}_\epsilon(K)) \cdot \left[\mathbb{V}_h(V_{h+1}^* - \check{V}_{i,h+1})\right](s_h^k, a_h^k) \leq F_{i,h}$$

Based on previous lemmas, we can construct an optimistic estimator $\sigma_{k,h}$ for the transition variance. Under this situation, the weighted regression have the following guarantee.

**Lemma D.11.** If the events $\underline{\mathcal{E}}^{\tilde{f}}$, $\underline{\mathcal{E}}^{\hat{f}}$, $\underline{\mathcal{E}}^{\check{f}}$, $\bar{\mathcal{E}}^{\hat{f}}$ and $\bar{\mathcal{E}}^{\check{f}}$ holds, then events $\mathcal{E}^{\hat{f}} = \mathcal{E}_1^{\hat{f}}$ and $\mathcal{E}^{\check{f}} = \mathcal{E}_1^{\check{f}}$ holds.

### D.4   PROOF OF REGRET BOUND

In the subsection, we first define the following high probability events to control the stochastic noise from the transition process:

$$\mathcal{E}_1 = \left\{\forall h \in [H], \sum_{k=1}^K \sum_{h'=h}^H [\mathbb{P}_h(V_{k,h+1} - V_{h+1}^{\pi^k})](s_h^k, a_h^k)\right.$$
$$\left. - \sum_{k=1}^K \sum_{h'=h}^H \left(V_{k,h+1}(s_{h+1}^k) - V_{h+1}^{\pi^k}(s_{h+1}^k)\right) \leq \sqrt{2HK \log(H/\delta)}\right\},$$
$$\mathcal{E}_2 = \left\{\forall h \in [H], \sum_{k=1}^K \sum_{h'=h}^H [\mathbb{P}_h(V_{k,h+1} - \check{V}_{k,h+1})](s_h^k, a_h^k)\right.$$
$$\left. - \sum_{k=1}^K \sum_{h'=h}^H \left(V_{k,h+1}(s_{h+1}^k) - \check{V}_{k,h+1}(s_{h+1}^k)\right) \leq \sqrt{2HK \log(H/\delta)}\right\}.$$

According to Azuma–Hoeffding inequality (Lemma G.1), we dicrely have the following results.

**Lemma D.12.** Events $\mathcal{E}_1$ and $\mathcal{E}_2$ holds with probability at least $1 - 2\delta$.

Next, we need the following lemma to control the summation of confidence radius.

**Lemma D.13.** For any parameters $\beta \geq 1$ and stage $h \in [H]$, the summation of confidence radius over episode $k \in [K]$ is upper bounded by

$$\sum_{k=1}^K \min\left(\beta D_{\mathcal{F}_h}(z; z_{[k-1],h}, \bar{\sigma}_{[k-1],h}), 1\right)$$

$$\leq (1 + \beta\gamma^2) \dim_{\alpha,K}(\mathcal{F}_h) + 2\beta\sqrt{\dim_{\alpha,K}(\mathcal{F}_h)}\sqrt{\sum_{k=1}^{K}(\sigma_{k,h}^2 + \alpha^2)},$$

where $z = (s, a)$ and $z_{[k-1],h} = \{z_{1,h}, z_{2,h}, .., z_{k-1,h}\}$.

Then, we can deompose the total regret in the first $K$ episode to the summation of variance $\sum_{k=1}^{K}\sum_{h=1}^{H}\sigma_{k,h}^2$ as the following lemma.

**Lemma D.14.** On the events $\mathcal{E}^{\hat{f}} = \mathcal{E}_1^{\hat{f}}$, $\mathcal{E}^{\check{f}} = \mathcal{E}_1^{\check{f}}$ and $\mathcal{E}_1$, for each stage $h \in [H]$, the regret in the first $K$ episodes can be decomposes and controlled as:

$$\sum_{k=1}^{K}\left(V_{k,h}(s_h^k) - V_h^{\pi^k}(s_h^k)\right) \leq 2CH(1+\chi)(1+\hat{\beta}_k\gamma^2)\dim_{\alpha,K}(\mathcal{F}) + \sqrt{2HK\log(H/\delta)}$$

$$+ 4C(1+\chi)\hat{\beta}_k\sqrt{\dim_{\alpha,K}(\mathcal{F})}\sqrt{H\sum_{k=1}^{K}\sum_{h=1}^{H}(\sigma_{k,h}^2 + \alpha^2)},$$

and for all stage $h \in [H]$, we further have

$$\sum_{k=1}^{K}\sum_{h=1}^{H}\left[\mathbb{P}_h(V_{k,h+1} - V_{h+1}^{\pi^k})\right](s_h^k, a_h^k)$$

$$\leq 2CH^2(1+\chi)(1+\hat{\beta}_k\gamma^2)\dim_{\alpha,K}(\mathcal{F}) + 4CH(1+\chi)\hat{\beta}_k\sqrt{\dim_{\alpha,K}(\mathcal{F}_h)}\sqrt{H\sum_{k=1}^{K}\sum_{h=1}^{H}(\sigma_{k,h}^2 + \alpha^2)}$$

$$+ 2\sqrt{2H^3K\log(H/\delta)}.$$

In addition, the gap betweeen the optimistic value function $V_{k,h}(s)$ and pessimistic value function $\check{V}_{k,h}(s)$ can be upper bounded by the following lemma.

**Lemma D.15.** On the events $\mathcal{E}^{\hat{f}} = \mathcal{E}_1^{\hat{f}}$, $\mathcal{E}^{\check{f}} = \mathcal{E}_1^{\check{f}}$ and $\mathcal{E}_2$, for each stage $h \in [H]$, the regret in the first $K$ episodes can be decomposes and controlled as:

$$\sum_{k=1}^{K}\left(V_{k,h}(s_h^k) - \check{V}_{k,h}(s_h^k))\right) \leq 4CH(1+\chi)(1+\hat{\beta}_k\gamma^2)\dim_{\alpha,K}(\mathcal{F}) + \sqrt{2HK\log(H/\delta)}$$

$$+ 8C(1+\chi)\hat{\beta}_k\sqrt{\dim_{\alpha,K}(\mathcal{F})}\sqrt{H\sum_{k=1}^{K}\sum_{h=1}^{H}(\sigma_{k,h}^2 + \alpha^2)},$$

and for all stage $h \in [H]$, we further have

$$\sum_{k=1}^{K}\sum_{h=1}^{H}\left[\mathbb{P}_h(V_{k,h+1} - \check{V}_{k,h+1})\right](s_h^k, a_h^k)$$

$$\leq 4CH^2(1+\chi)(1+\hat{\beta}_k\gamma^2)\dim_{\alpha,K}(\mathcal{F}_h) + 8CH(1+\chi)\hat{\beta}_k\sqrt{\dim_{\alpha,K}(\mathcal{F})}\sqrt{H\sum_{k=1}^{K}\sum_{h=1}^{H}(\sigma_{k,h}^2 + \alpha^2)}$$

$$+ 2\sqrt{2H^3K\log(H/\delta)}.$$

We define the following high probability event $\mathcal{E}_3$ to control the summation of variance.

$$\mathcal{E}_3 = \left\{\sum_{k=1}^{K}\sum_{h=1}^{H}[\mathbb{V}_h V_{h+1}^{\pi^k}](s_h^k, a_h^k) \leq 3K + 3H\log(1/\delta)\right\}.$$

According to Jin et al. (2018) (Lemma C.5)[3], with probability at least $1 - \delta$. Condition on this event, the summation of variance $\sum_{k=1}^{K}\sum_{h=1}^{H}\sigma_{k,h}^2$ can be upper bounded by the following lemma.

---

[3]Jin et al. (2018) showed that $\sum_{k=1}^{K}\sum_{h=1}^{H}[\mathbb{V}_h V_{h+1}^{\pi^k}](s_h^k, a_h^k) = \widetilde{O}(KH^2 + H^3)$ when $\sum h = 1^H r_h(s_h, a_h) \leq H$. In this work, we assume the total reward satisfied $\sum_{h=1}^{H} r_h(s_h, a_h) \leq 1$, and the summation of variance is upper bounded by $\widetilde{O}(K + H)$

**Lemma D.16.** On the events $\mathcal{E}_1, \mathcal{E}_2, \mathcal{E}_3, \mathcal{E}^{\hat{f}} = \mathcal{E}_1^{\hat{f}}$ and $\mathcal{E}^{\check{f}} = \mathcal{E}_1^{\check{f}}$, the total estimated variance is upper bounded by:

$$\sum_{k=1}^{K} \sum_{h=1}^{H} \sigma_{k,h}^2 \leq (\log \mathcal{N}_{\mathcal{F}}(\epsilon) + \log \mathcal{N}_\epsilon(K)) \times O\big((1+\gamma^2)(\beta_k + H\widehat{\beta}_k + \widetilde{\beta}_k)H \dim_{\alpha,K}(\mathcal{F})\big)$$

$$+ (\log \mathcal{N}_{\mathcal{F}}(\epsilon) + \log \mathcal{N}_\epsilon(K))^2 \times O\big((\beta_k + H\widehat{\beta}_k + \widetilde{\beta}_k)^2 H \dim_{\alpha,K}(\mathcal{F})\big)$$

$$+ O(K + KH\alpha^2).$$

With the help of all previous lemmas, we can prove the Theorem 4.1

*Proof of Theorem 4.1.* The low switching cost result is given by Lemma F.1.

After taking a union bound, the high probability events $\mathcal{E}_1, \mathcal{E}_2, \mathcal{E}_3, \underline{\mathcal{E}}^{\widetilde{f}}, \underline{\mathcal{E}}^{\hat{f}}, \underline{\mathcal{E}}^{\check{f}}, \bar{\mathcal{E}}^{\hat{f}}$ and $\bar{\mathcal{E}}^{\check{f}}$ holds with probability at least $1 - 10\delta$. Conditioned on these events, the regret is upper bounded by

$$\text{Regret}(K)$$

$$= \sum_{k=1}^{K} \left( V_1^*(s_1^k) - V_{k,1}^{\pi^k}(s_1^k) \right)$$

$$\leq \sum_{k=1}^{K} \left( V_{k,1}(s_1^k) - V_{k,1}^{\pi^k}(s_1^k) \right)$$

$$\leq 2CH(1+\chi)(1+\widehat{\beta}_K\gamma^2) \dim_{\alpha,K}(\mathcal{F}) + \sqrt{2HK \log(H/\delta)}$$

$$+ 4C(1+\chi)\widehat{\beta}_K \sqrt{\dim_{\alpha,K}(\mathcal{F})} \sqrt{H \sum_{k=1}^{K} \sum_{h=1}^{H} (\sigma_{k,h}^2 + \alpha^2)}$$

$$\leq O\left( H \dim_{\alpha,K}(\mathcal{F}) \cdot (1 + \widehat{\beta}_K + \gamma^2) \right) + O\left( \widehat{\beta}_K \sqrt{\dim_{\alpha,K}(\mathcal{F})} \cdot \sqrt{HK} \right) + \sqrt{2HK \log(H/\delta)}$$

$$+ O\left( H \dim_{\alpha,K}(\mathcal{F})\widehat{\beta}_K(\beta_K + H\widehat{\beta}_K + \widetilde{\beta}_K) \cdot \log[\mathcal{N}_{\mathcal{F}}(\epsilon) \cdot \mathcal{N}_\epsilon(K)] \right) + O\left( H \dim_{\alpha,K}(\mathcal{F})\widehat{\beta}_K(1+\gamma^2) \right)$$

$$= \widetilde{O}\left( H^3 \dim_{\alpha,K}^2(\mathcal{F}) \cdot \log \mathcal{N}_{\mathcal{F}}\left(\frac{1}{2KHL}\right) \cdot \log\left[ \mathcal{N}_{\mathcal{F}}\left(\frac{1}{2KHL}\right) \cdot \mathcal{N}\left(\mathcal{B}, \frac{1}{2KHL\widehat{\beta}_K}\right) \right] \right)$$

$$+ \widetilde{O}\left( H^{2.5} \dim_{\alpha,K}^{2.5}(\mathcal{F}) \cdot \sqrt{\log \mathcal{N}_{\mathcal{F}}\left(\frac{1}{2KHL}\right)} \cdot \log^{1.5}\left[ \mathcal{N}_{\mathcal{F}}\left(\frac{1}{2KHL}\right) \cdot \mathcal{N}\left(\mathcal{B}, \frac{1}{2KHL\widehat{\beta}_K}\right) \right] \right)$$

$$+ \widetilde{O}\left( \sqrt{\dim_{\alpha,K}(\mathcal{F}) \log \mathcal{N}_{\mathcal{F}}\left(\frac{1}{2KHL}\right) \cdot HK} \right) \tag{D.9}$$

where the first inequality holds due to Lemma D.7, the second inequality holds due to Lemma D.14 and the last inequality holds due to Lemma D.16. Thus, we complete the proof of Theorem 4.1. We can reorganize (D.9) into the following upper bound for regret,

$$\text{Regret}(K) = \widetilde{O}(\sqrt{\dim(\mathcal{F}) \log \mathcal{N} \cdot HK})$$

$$+ \widetilde{O}\left( H^{2.5} \dim^2(\mathcal{F}) \sqrt{\log \mathcal{N}} \log(\mathcal{N} \cdot \mathcal{N}_b) \cdot \sqrt{H \log \mathcal{N} + \dim(\mathcal{F}) \log(\mathcal{N} \cdot \mathcal{N}_b)} \right),$$

where we denote the covering number of bonus function class $\mathcal{N}\left(\mathcal{B}, \frac{1}{2KHL\widehat{\beta}_K}\right)$ by $\mathcal{N}_b$, the covering number of function class $\mathcal{F}$ by $\mathcal{N}$ and the dimension $\dim_{\alpha,K}(\mathcal{F})$ by $\dim(\mathcal{F})$. □

# E PROOF OF LEMMAS IN APPENDIX B

In this section, we provide the detailed proof of lemmas in Appendix B.

## E.1 PROOF OF HIGH PROBABILITY EVENTS

### E.1.1 PROOF OF LEMMA D.1

*Proof of Lemma D.1.* We first prove that for an arbitrarily chosen $h \in [H]$, $\underline{\mathcal{E}}_{k,h}^{\widetilde{f}}$ holds with probability at least $1 - \delta/H$ for all $k$.

For simplicity, in this proof, we denote $\mathcal{T}_h^2 V_{k,h+1}(s_h^i, a_h^i)$ as $\widetilde{f}_k^*(s_h^i, a_h^i)$ where $\widetilde{f}_k^* \in \mathcal{F}_h$ exists due to our assumption. For any function $V : \mathcal{S} \to [0, 1]$, let $\widetilde{\eta}_h^k(V) = \left(r_h^k + V(s_{h+1}^k)\right)^2 - \mathbb{E}_{s' \sim s_h^k, a_h^k}\left[\left(r(s_h^k, a_h^k, s') + V(s')\right)^2\right]$.

By simple calculation, for all $f \in \mathcal{F}_h$, we have

$$\sum_{i \in [k-1]} \left(f(s_h^i, a_h^i) - \widetilde{f}_k^*(s_h^i, a_h^i)\right)^2 + 2 \underbrace{\sum_{i \in [k-1]} \left(f(s_h^i, a_h^i) - \widetilde{f}_k^*(s_h^i, a_h^i)\right) \cdot \widetilde{\eta}_h^i(V_{k,h+1})}_{I(f, \widetilde{f}_k^*, V)}$$

$$= \sum_{i \in [k-1]} \left[\left(r_h^i + V_{k,h+1}(s_h^i)\right)^2 - f(s_h^i, a_h^i)\right]^2 - \sum_{i \in [k-1]} \left[\left(r_h^i + V_{k,h+1}(s_h^i)\right)^2 - \widetilde{f}_k^*(s_h^i, a_h^i)\right]^2.$$

Due to the definition of $\widetilde{f}_{k,h}$, we have

$$\sum_{(i,j) \in [k-1] \times [H]} \left(\widetilde{f}_{k,h}(s_j^i, a_j^i) - \widetilde{f}_k^*(s_j^i, a_j^i)\right)^2 + 2I(\widetilde{f}_{k,h}, \widetilde{f}_k^*, V_{k,h+1}) \le 0. \qquad (\text{E.1})$$

Then we give a high probability bound for $-I(f, \widetilde{f}_k^*, V_{k,h+1})$ through the following calculation. Applying Lemma G.4, for fixed $f$, $\bar{f}$ and $V$, with probability at least $1 - \delta$,

$$-I(f, \bar{f}, V) := - \sum_{i \in [k-1]} \left(f(s_h^i, a_h^i) - \bar{f}(s_h^i, a_h^i)\right) \cdot \widetilde{\eta}_h^i(V)$$

$$\le 8\lambda \sum_{i \in [k-1]} \left(f(s_h^i, a_h^i) - \bar{f}(s_h^i, a_h^i)\right)^2 + \frac{1}{\lambda} \cdot \log \frac{1}{\delta}.$$

By the definition of $V$ in line $\cdot$ of Algorithm 1, $V_{k,h+1}$ lies in the optimistic value function class $\mathcal{V}_k$ defined in (F.4). Applying a union bound for all the value functions $V^c$ in the corresponding $\epsilon$-net $\mathcal{V}^c$, we have

$$-I(f, \bar{f}, V^c) \le \frac{1}{4} \sum_{i \in [k-1]} \left(f(s_h^i, a_h^i) - \bar{f}(s_h^i, a_h^i)\right)^2 + 32 \cdot \log \frac{\mathcal{N}_\epsilon(k)}{\delta}$$

holds for all $k$ with probability at least $1 - \delta$.

For all $V$ such that $\|V - V^c\|_\infty \le \epsilon$, we have $|\eta_h^i(V) - \eta_h^i(V^c)| \le 4\epsilon$. Therefore, with probability $1 - \delta$, the following bound holds for $I(f, \bar{f}, V_{k,h+1})$:

$$-I(f, \bar{f}, V_{k,h+1}) \le \frac{1}{4} \sum_{i \in [k-1]} \left(f(s_h^i, a_h^i) - \bar{f}(s_h^i, a_h^i)\right)^2 + 32 \cdot \log \frac{\mathcal{N}_\epsilon(k)}{\delta} + 4\epsilon \cdot k,$$

To further bound $I(\widetilde{f}_{k,h}, \widetilde{f}_k^*, V_{k,h+1})$ in (E.1), we apply an $\epsilon$-covering argument on $\mathcal{F}_h$ and show that with probability at least $1 - \delta$,

$$-I(\widetilde{f}_{k,h}, \widetilde{f}_k^*, V_{k,h+1}) \le \frac{1}{4} \sum_{i \in [k-1]} \left(\widetilde{f}_{k,h}(s_h^i, a_h^i) - \widetilde{f}_k^*(s_h^i, a_h^i)\right)^2 + 32 \cdot \log \frac{\mathcal{N}_\epsilon(k) \cdot \mathcal{N}_\mathcal{F}(\epsilon)}{\delta}$$

$$+ 16L\epsilon \cdot k \qquad (\text{E.2})$$

for probability at least $1 - \delta$.

Substituting (E.2) into (E.1) and rearranging the terms,

$$\sum_{i \in [k-1]} \left(\widetilde{f}_{k,h}(s_h^i, a_h^i) - \widetilde{f}_k^*(s_h^i, a_h^i)\right)^2 \le 128 \log \frac{\mathcal{N}_\epsilon(k) \cdot \mathcal{N}_\mathcal{F}(\epsilon) \cdot H}{\delta} + 64L\epsilon \cdot k$$

for all $k$ with probability at least $1 - \delta/H$.

Finally, we apply a union bound over all $h \in [H]$ to conclude that $\mathcal{E}^{\widetilde{f}}$ holds with probability at least $1 - \delta$.

$\square$

### E.1.2 PROOF OF LEMMAS D.2 AND D.3

*Proof of Lemma D.2.* Similar to the proof of Lemma D.1, we first prove that for an arbitrarily chosen $h \in [H]$, $\mathcal{E}_{k,h}^{\hat{f}}$ holds with probability at least $1 - \delta/H$ for all $k$.

In this proof, we denote $\mathcal{T}_h V_{k,h+1}(s_h^i, a_h^i)$ as $\hat{f}_k^*(s_h^i, a_h^i)$ where $\hat{f}_k^* \in \mathcal{F}_h$ exists due to our assumption. For any function $V : \mathcal{S} \to [0, 1]$, let $\hat{\eta}_h^k(V) = \left(r_h^k + V(s_{h+1}^k)\right) - \mathbb{E}_{s' \sim s_h^k, a_h^k}\left[r_h(s_h^k, a_h^k, s') + V(s')\right]$. For all $f \in \mathcal{F}_h$, we have

$$\sum_{i \in [k-1]} \frac{1}{(\bar{\sigma}_{i,h})^2}\left(f(s_h^i, a_h^i) - \hat{f}_k^*(s_h^i, a_h^i)\right)^2 + 2 \underbrace{\sum_{i \in [k-1]} \frac{1}{(\bar{\sigma}_{i,h})^2}\left(f(s_h^i, a_h^i) - \hat{f}_k^*(s_h^i, a_h^i)\right) \cdot \hat{\eta}_h^i(V_{k,h+1})}_{I(f, \hat{f}_k^*, V)}$$

$$= \sum_{i \in [k-1]} \frac{1}{(\bar{\sigma}_{i,h})^2}\left[r_h^i + V_{k,h+1}(s_h^i)^2 - f(s_h^i, a_h^i)\right]^2 - \sum_{i \in [k-1]} \frac{1}{(\bar{\sigma}_{i,h})^2}\left[r_h^i + V_{k,h+1}(s_h^i) - \hat{f}_k^*(s_h^i, a_h^i)\right]^2.$$

By the definition of $\hat{f}_{k,h}$, we have

$$\sum_{i \in [k-1]} \frac{1}{(\bar{\sigma}_{i,h})^2}\left(\hat{f}_{k,h}(s_h^i, a_h^i) - \hat{f}_k^*(s_h^i, a_h^i)\right)^2 + 2I(\hat{f}_{k,h}, V_{k,h+1}) \le 0. \tag{E.3}$$

Then we give a high probability bound for $-I(\hat{f}_{k,h}, \hat{f}_k^*, V_{k,h+1})$ through the following calculation. Applying Lemma G.4, for fixed $f$, $\bar{f}$ and $V$, with probability at least $1 - \delta$,

$$-I(f, \bar{f}, V) := -\sum_{i \in [k-1]} \frac{1}{(\bar{\sigma}_{i,h})^2}\left(f(s_h^i, a_h^i) - \bar{f}(s_h^i, a_h^i)\right) \cdot \hat{\eta}_h^i(V)$$

$$\le 8\lambda \frac{1}{\alpha^2} \sum_{i \in [k-1]} \frac{1}{(\bar{\sigma}_{i,h})^2}\left(f(s_h^i, a_h^i) - \bar{f}(s_h^i, a_h^i)\right)^2 + \frac{1}{\lambda} \cdot \log \frac{1}{\delta}.$$

By the definition of $V$ in line $\cdot$ of Algorithm 1, $V_{k,h+1}$ lies in the optimistic value function class $\mathcal{V}_k$ defined in (F.4). Applying a union bound for all the value functions $V^c$ in the corresponding $\epsilon$-net $\mathcal{V}^c$, we have

$$-I(f, \bar{f}, V^c) \le \frac{1}{4} \sum_{i \in [k-1]} \frac{1}{(\bar{\sigma}_{i,h})^2}\left(f(s_h^i, a_h^i) - \bar{f}(s_h^i, a_h^i)\right)^2 + \frac{32}{\alpha^2} \cdot \log \frac{\mathcal{N}_\epsilon(k)}{\delta}$$

holds for all $k$ with probability at least $1 - \delta$.
For all $V$ such that $\|V - V^c\|_\infty \le \epsilon$, we have $|\eta_h^i(V) - \eta_h^i(V^c)| \le 4\epsilon$. Therefore, with probability $1 - \delta$, the following bound holds for $I(f, \bar{f}, V_{k,h+1})$:

$$-I(f, \bar{f}, V_{k,h+1}) \le \frac{1}{4} \sum_{i \in [k-1]} \frac{1}{(\bar{\sigma}_{i,h})^2}\left(f(s_h^i, a_h^i) - \bar{f}(s_h^i, a_h^i)\right)^2 + \frac{32}{\alpha^2} \cdot \log \frac{\mathcal{N}_\epsilon(k)}{\delta} + 4\epsilon \cdot k/\alpha^2,$$

To further bound $I(\hat{f}_{k,h}, \hat{f}_k^*, V_{k,h+1})$ in (E.3), we apply an $\epsilon$-covering argument on $\mathcal{F}_h$ and show that with probability at least $1 - \delta$,

$$-I(\hat{f}_{k,h}, \hat{f}_k^*, V_{k,h+1}) \le \frac{1}{4} \sum_{i \in [k-1]} \frac{1}{(\bar{\sigma}_{i,h})^2}\left(\hat{f}_{k,h}(s_h^i, a_h^i) - \hat{f}_k^*(s_h^i, a_h^i)\right)^2 + 32 \cdot \log \frac{\mathcal{N}_\epsilon(k) \cdot \mathcal{N}_\mathcal{F}(\epsilon)}{\delta}$$
$$+ 16L\epsilon \cdot k/\alpha^2. \tag{E.4}$$

Substituting (E.4) into (E.3) and rearranging the terms, we have

$$\sum_{i \in [k-1]} \frac{1}{(\bar{\sigma}_{i,h})^2}\left(\hat{f}_{k,h}(s_h^i, a_h^i) - \hat{f}_k^*(s_h^i, a_h^i)\right)^2 \le 128 \cdot \log \frac{\mathcal{N}_\epsilon(k) \cdot \mathcal{N}_\mathcal{F}(\epsilon) H}{\delta} + 64L\epsilon \cdot k/\alpha^2$$

for all $k$ with probability at least $1 - \delta/H$. Then we can complete the proof by using a union bound over all $h \in [H]$. $\qquad \square$

*Proof of Lemma D.3.* The proof is almost identical to the proof of Lemma D.2. $\qquad \square$

### E.1.3 PROOF OF LEMMAS D.4 AND D.5

*Proof of Lemma D.4.* Similar to the proof of Lemma D.1, we first prove that for an arbitrary $h \in [H]$, $\bar{\mathcal{E}}_{k,h}^{\hat{f}}$ holds with probability at least $1 - \delta/H$ for all $k$.

In this proof, we denote $\mathcal{T}_h V_{k,h+1}(s_h^i, a_h^i)$ as $\hat{f}_k^*(s_h^i, a_h^i)$ where $\hat{f}_k^* \in \mathcal{F}$ exists due to our assumption. For any function $V : \mathcal{S} \to [0, 1]$, let $\hat{\eta}_h^k(V) = \left(r_h^k + V(s_{h+1}^k)\right) - \mathbb{E}_{s' \sim s_h^k, a_h^k}\left[r_h(s_h^k, a_h^k, s') + V(s')\right]$. For all $f \in \mathcal{F}_h$, we have

$$\sum_{i \in [k-1]} \frac{\widehat{\mathbb{1}}_{i,h}}{(\bar{\sigma}_{i,h})^2}\left(f(s_h^i, a_h^i) - \hat{f}_k^*(s_h^i, a_h^i)\right)^2 + 2\underbrace{\sum_{i \in [k-1]} \frac{\widehat{\mathbb{1}}_{i,h}}{(\bar{\sigma}_{i,h})^2}\left(f(s_h^i, a_h^i) - \hat{f}_k^*(s_h^i, a_h^i)\right) \cdot \hat{\eta}_h^i(V_{k,h+1})}_{I(f, \hat{f}_k^*, V)}$$

$$= \sum_{i \in [k-1]} \frac{\widehat{\mathbb{1}}_{i,h}}{(\bar{\sigma}_{i,h})^2}\left[r_h^i + V_{k,h+1}(s_h^i)^2 - f(s_h^i, a_h^i)\right]^2 - \sum_{i \in [k-1]} \frac{\widehat{\mathbb{1}}_{i,h}}{(\bar{\sigma}_{i,h})^2}\left[r_h^i + V_{k,h+1}(s_h^i) - \hat{f}_k^*(s_h^i, a_h^i)\right]^2.$$

Due to the definition of $\hat{f}_{k,h}$, we have

$$\sum_{i \in [k-1]} \frac{\widehat{\mathbb{1}}_{i,h}}{(\bar{\sigma}_{i,h})^2}\left(\hat{f}_{k,h}(s_h^i, a_h^i) - \hat{f}_k^*(s_h^i, a_h^i)\right)^2 + 2I(\hat{f}_{k,h}, V_{k,h+1}) \le 0. \tag{E.5}$$

Then it suffices to bound the value of $I(f, \bar{f}, V_{k,h+1})$ for all $f, \bar{f} \in \mathcal{F}$.

Unlike the proof for Lemma D.1, we decompose $I(f, \bar{f}, V_{k,h+1})$ into two parts:

$$I(f, \bar{f}, V_{k,h+1}) = \sum_{i \in [k-1]} \frac{\widehat{\mathbb{1}}_{i,h}}{(\bar{\sigma}_{i,h})^2}\left(f(s_h^i, a_h^i) - \bar{f}(s_h^i, a_h^i)\right) \cdot \hat{\eta}_h^i(V_{h+1}^*)$$

$$+ \sum_{i \in [k-1]} \frac{\widehat{\mathbb{1}}_{i,h}}{(\bar{\sigma}_{i,h})^2}\left(f(s_h^i, a_h^i) - \bar{f}(s_h^i, a_h^i)\right) \cdot \hat{\eta}_h^i(V_{k,h+1} - V_{h+1}^*). \tag{E.6}$$

Then we bound the two terms separately.

For the first term, we first check the following conditions before applying Lemma G.2, which is a variant of Freedman inequality. For fixed $f$ and $\bar{f}$, we have

$$\mathbb{E}\left[\frac{\widehat{\mathbb{1}}_{i,h}}{(\bar{\sigma}_{i,h})^2}\left(f(s_h^i, a_h^i) - \bar{f}(s_h^i, a_h^i)\right) \cdot \hat{\eta}_h^i(V_{h+1}^*)\right] = 0,$$

since $s_{h+1}^i$ is sampled from $\mathbb{P}_h(\cdot|s_h^i, a_h^i)$.

Next, we need to derive a bound for the maximum absolute value of each 'weighted' transition noise:

$$\max_{i \in [k-1]} \left|\frac{\widehat{\mathbb{1}}_{i,h}}{(\bar{\sigma}_{i,h})^2}\left(f(s_h^i, a_h^i) - \bar{f}(s_h^i, a_h^i)\right) \cdot \hat{\eta}_h^i(V_{h+1}^*)\right|$$

$$\le 4 \max_{i \in [k-1]} \left|\frac{\widehat{\mathbb{1}}_{i,h}}{(\bar{\sigma}_{i,h})^2}\left(f(s_h^i, a_h^i) - \bar{f}(s_h^i, a_h^i)\right)\right|$$

$$\le 4 \max_{i \in [k-1]} \frac{1}{(\bar{\sigma}_{i,h})^2}\sqrt{D_{\mathcal{F}_h}^2\left(z_{i,h}; z_{[i-1],h}, \bar{\sigma}_{[i-1],h}\right)\left(\sum_{\tau \in [k-1]} \frac{\widehat{\mathbb{1}}_{\tau,h}}{(\bar{\sigma}_{\tau,h})^2}\left(f(s_h^\tau, a_h^\tau) - \bar{f}(s_h^\tau, a_h^\tau)\right)^2 + \lambda\right)}$$

$$\le 4 \cdot \gamma^{-2}\sqrt{\sum_{\tau \in [k-1]} \frac{\widehat{\mathbb{1}}_{\tau,h}}{(\bar{\sigma}_{\tau,h})^2}\left(f(s_h^\tau, a_h^\tau) - \bar{f}(s_h^\tau, a_h^\tau)\right)^2 + \lambda}, \tag{E.7}$$

where the second inequality follows from the definition of $\mathcal{D}_{\mathcal{F}_h}$ in Definition 2.4, the last inequality holds since $\bar{\sigma}_{i,h} \ge \gamma \cdot D_{\mathcal{F}_h}^{1/2}\left(z_{i,h}; z_{[i-1],h}, \bar{\sigma}_{[i-1],h}\right)$ according to line 23 in Algorithm 1. From the definition of $\widehat{\mathbb{1}}_{i,h}$ in (D.6) we directly obtain the following upper bound of the variance:

$$\sum_{i \in [k-1]} \mathbb{E}\left[\frac{\widehat{\mathbb{1}}_{i,h}}{(\bar{\sigma}_{i,h})^4}\left(f(s_h^i, a_h^i) - \bar{f}(s_h^i, a_h^i)\right)^2 \cdot \hat{\eta}_h^i(V_{h+1}^*)^2\right]$$

$$\leq 4 \sum_{i\in[k-1]} \frac{\widehat{\mathbb{1}}_{i,h}}{(\bar{\sigma}_{i,h})^2} \left(f(s_h^i,a_h^i) - \bar{f}(s_h^i,a_h^i)\right)^2 \leq 4L^2 k/\alpha^2 := V.$$

Applying Lemma G.2 with $V = 4L^2 k/\alpha^2$, $M = 2L/\alpha^2$ and $v = m = 1$, for fixed $f$, $\bar{f}$, $k$, with probability at least $1 - \delta/(2k^2 \cdot \mathcal{N}_{\mathcal{F}}(\epsilon) \cdot H)$,

$$-\sum_{i\in[k-1]} \frac{\widehat{\mathbb{1}}_{i,h}}{(\bar{\sigma}_{i,h})^2} \left(f(s_h^i,a_h^i) - \bar{f}(s_h^i,a_h^i)\right) \cdot \widehat{\eta}_h^i(V_{h+1}^*) \tag{E.8}$$

$$\leq \iota \sqrt{2 \left(8 \sum_{i\in[k-1]} \frac{\widehat{\mathbb{1}}_{i,h}}{(\bar{\sigma}_{i,h})^2} \left(f(s_h^i,a_h^i) - \bar{f}(s_h^i,a_h^i)\right)^2 + 1\right)}$$

$$+ \frac{2}{3}\iota^2 \left(8\gamma^{-2} \sqrt{\sum_{\tau\in[k-1]} \frac{\widehat{\mathbb{1}}_{\tau,h}}{(\bar{\sigma}_{\tau,h})^2} \left(f(s_h^\tau,a_h^\tau) - \bar{f}(s_h^\tau,a_h^\tau)\right)^2 + \lambda} + 1\right) \tag{E.9}$$

$$\leq \left(4\iota + \frac{16}{3}\iota^2\gamma^{-2}\right) \sqrt{\sum_{\tau\in[k-1]} \frac{\widehat{\mathbb{1}}_{\tau,h}}{(\bar{\sigma}_{\tau,h})^2} \left(f(s_h^\tau,a_h^\tau) - \bar{f}(s_h^\tau,a_h^\tau)\right)^2 + \lambda} + \sqrt{2}\iota + \frac{2}{3}\iota^2, \tag{E.10}$$

where $\iota := \iota_1(k,h,\delta) = \sqrt{\log \frac{2k^2\left(2\log(\frac{4L^2k}{\alpha^2})+2\right)\cdot\left(\log(\frac{4L}{\alpha^2})+2\right)\cdot\mathcal{N}_{\mathcal{F}}(\epsilon)}{\delta/H}}$.

Using a union bound across all $f, \bar{f} \in \mathcal{C}(\mathcal{F}_h, \epsilon)$ and $k \geq 1$, with probability at least $1 - \delta/H$,

$$-\sum_{i\in[k-1]} \frac{\widehat{\mathbb{1}}_{i,h}}{(\bar{\sigma}_{i,h})^2} \left(f(s_h^i,a_h^i) - \bar{f}(s_h^i,a_h^i)\right) \cdot \widehat{\eta}_h^i(V_{h+1}^*)$$

$$\leq \left(4\iota + \frac{16}{3}\iota^2\gamma^{-2}\right) \sqrt{\sum_{\tau\in[k-1]} \frac{\widehat{\mathbb{1}}_{\tau,h}}{(\bar{\sigma}_{\tau,h})^2} \left(f(s_h^\tau,a_h^\tau) - \bar{f}(s_h^\tau,a_h^\tau)\right)^2 + \lambda} + \sqrt{2}\iota + \frac{2}{3}\iota^2 \tag{E.11}$$

holds for all $f, \bar{f} \in \mathcal{C}(\mathcal{F}_h, \epsilon)$ and $k$. Due to the definition of $\epsilon$-net, we deduce that for $\widehat{f}_{k,h}$ and $\widehat{f}_k^*$,

$$-\sum_{i\in[k-1]} \frac{\widehat{\mathbb{1}}_{i,h}}{(\bar{\sigma}_{i,h})^2} \left(\widehat{f}_{k,h}(s_h^i,a_h^i) - \widehat{f}_k^*(s_h^i,a_h^i)\right) \cdot \widehat{\eta}_h^i(V_{h+1}^*)$$

$$\leq \left(4\iota + \frac{16}{3}\iota^2\gamma^{-2}\right) \sqrt{\sum_{\tau\in[k-1]} \frac{\widehat{\mathbb{1}}_{\tau,h}}{(\bar{\sigma}_{\tau,h})^2} \left(\widehat{f}_{k,h}(s_h^\tau,a_h^\tau) - \widehat{f}_k^*(s_h^\tau,a_h^\tau)\right)^2 + \lambda} + \sqrt{2}\iota + \frac{2}{3}\iota^2$$

$$+ \left(4\iota + \frac{16}{3}\iota^2\gamma^{-2}\right)\sqrt{16k\epsilon L/\alpha^2} + 8\epsilon k/\alpha^2. \tag{E.12}$$

For the second term in (E.6), the following inequality holds for all $V \in \mathcal{V}_{k,h+1}$,

$$\mathbb{E}\left[\frac{1}{(\bar{\sigma}_{i,h})^2} \left(f(s_h^i,a_h^i) - \bar{f}(s_h^i,a_h^i)\right) \cdot \widehat{\eta}_h^i(V - V_{h+1}^*)\right] = 0,$$

$$\max_{i\in[k-1]} \left|\frac{1}{(\bar{\sigma}_{i,h})^2} \left(f(s_h^i,a_h^i) - \bar{f}(s_h^i,a_h^i)\right) \cdot \widehat{\eta}_h^i(V - V_{h+1}^*)\right|$$

$$\leq 4 \max_{i\in[k-1]} \left|\frac{1}{(\bar{\sigma}_{i,h})^2} \left(f(s_h^i,a_h^i) - \bar{f}(s_h^i,a_h^i)\right)\right|$$

$$\leq 4 \max_{i\in[k-1]} \frac{1}{(\bar{\sigma}_{i,h})^2} \sqrt{D_{\mathcal{F}_h}^2\left(z_{i,h}; z_{[i-1],h}, \bar{\sigma}_{[i-1],h}\right) \sum_{\tau\in[k-1]} \frac{1}{(\bar{\sigma}_{\tau,h})^2} \left(f(s_h^\tau,a_h^\tau) - \bar{f}(s_h^\tau,a_h^\tau)\right)^2 + \lambda}$$

$$= 4 \cdot \gamma^{-2} \sqrt{\sum_{\tau\in[k-1]} \frac{1}{(\bar{\sigma}_{\tau,h})^2} \left(f(s_h^\tau,a_h^\tau) - \bar{f}(s_h^\tau,a_h^\tau)\right)^2 + \lambda},$$

where the calculation is similar to that in (E.7).

We denote the sum of variance by $\text{var}(V - V_{h+1}^*)$ as follows for simplicity:

$$\text{var}(V - V_{h+1}^*) := \sum_{i \in [k-1]} \mathbb{E} \left[ \frac{\widehat{\mathbb{1}}_{i,h}}{(\bar{\sigma}_{i,h})^4} \left( f(s_h^i, a_h^i) - \bar{f}(s_h^i, a_h^i) \right)^2 \cdot \widehat{\eta}_h^i (V - V_{h+1}^*)^2 \right] \tag{E.13}$$

$$\leq k \cdot L^2 / \alpha^4. \tag{E.14}$$

For $V_{k,h+1}$, we have

$$\text{var}(V_{k,h+1} - V_{h+1}^*) := \sum_{i \in [k-1]} \mathbb{E} \left[ \frac{\widehat{\mathbb{1}}_{i,h}}{(\bar{\sigma}_{i,h})^4} \left( f(s_h^i, a_h^i) - \bar{f}(s_h^i, a_h^i) \right)^2 \cdot \widehat{\eta}_h^i (V_{k,h+1} - V_{h+1}^*)^2 \right]$$

$$\leq \frac{4}{\log \mathcal{N}_{\mathcal{F}}(\epsilon) + \log \mathcal{N}_{\epsilon}(K)} \sum_{i \in [k-1]} \frac{\widehat{\mathbb{1}}_{i,h}}{(\bar{\sigma}_{i,h})^2} \left( f(s_h^i, a_h^i) - \bar{f}(s_h^i, a_h^i) \right)^2,$$

where the second inequality holds due to the definition of $\widehat{\mathbb{1}}_{i,h}$ in (D.6).

With a similar argument as shown in (E.8)$\sim$(E.12), we have with probability at least $1 - \delta/(2k^2 \cdot \mathcal{N}_{\mathcal{F}}(\epsilon) \cdot \mathcal{N}_{\epsilon}(k-1) \cdot H)$, for a fixed $f$, $\bar{f}$, $k$ and $V$, (applying Lemma G.2, with $V = k \cdot L^2/\alpha^4$ and $M = 2L/\alpha^2$, $v = (\log \mathcal{N}_{\mathcal{F}}(\epsilon) + \log \mathcal{N}_{\epsilon}(K))^{-1/2}$, $m = v^2$. )

$$- \sum_{i \in [k-1]} \frac{\widehat{\mathbb{1}}_{i,h}}{(\bar{\sigma}_{i,h})^2} \left( f(s_h^i, a_h^i) - \bar{f}(s_h^i, a_h^i) \right) \cdot \widehat{\eta}_h^i (V_{k,h+1} - V_{h+1}^*)$$

$$\leq \iota \sqrt{2 \left( 2\text{var}(V - V_{h+1}^*) + (\log \mathcal{N}_{\mathcal{F}}(\epsilon) + \log \mathcal{N}_{\epsilon}(K))^{-1} \right)}$$

$$+ \frac{2}{3} \iota^2 \left( 8\gamma^{-2} \sqrt{\sum_{\tau \in [k-1]} \frac{\widehat{\mathbb{1}}_{\tau,h}}{(\bar{\sigma}_{\tau,h})^2} \left( f(s_h^\tau, a_h^\tau) - \bar{f}(s_h^\tau, a_h^\tau) \right)^2 + \lambda + (\log \mathcal{N}_{\mathcal{F}}(\epsilon) + \log \mathcal{N}_{\epsilon}(K))^{-1}} \right).$$

where

$$\log \frac{2k^2 \left( 2 \log \frac{L^2 k (\log \mathcal{N}_{\mathcal{F}}(\epsilon) \cdot \mathcal{N}_{\epsilon}(K))^{1/2}}{\alpha^4} + 2 \right) \cdot \left( \log(\frac{4L(\log \mathcal{N}_{\mathcal{F}}(\epsilon) \cdot \mathcal{N}_{\epsilon}(K))}{\alpha^2}) + 2 \right) \cdot \mathcal{N}_{\mathcal{F}}(\epsilon) \cdot \mathcal{N}_{\epsilon}(k)}{\delta/H}$$

$$\leq \log \frac{2k^2 \left( 2 \log \frac{L^2 k}{\alpha^4} + 2 \right) \cdot \left( \log(\frac{4L}{\alpha^2}) + 2 \right) \cdot \mathcal{N}_{\mathcal{F}}^4(\epsilon) \cdot \mathcal{N}_{\epsilon}^2(K)}{\delta/H} := \iota_2^2(k, h, \delta)$$

Using a union bound over all $(f, \bar{f}, V) \in \mathcal{C}(\mathcal{F}_h, \epsilon) \times \mathcal{C}(\mathcal{F}_h, \epsilon) \times \mathcal{V}_{k,h+1}^c$ and all $k \geq 1$, we have the inequality above holds for all such $f, \bar{f}, V, k$ with probability at least $1 - \delta/H$.

There exists a $V_{k,h+1}^c$ in the $\epsilon$-net such that $\|V_{k,h+1} - V_{k,h+1}^c\|_\infty \leq \epsilon$. Then we have

$$- \sum_{i \in [k-1]} \frac{\widehat{\mathbb{1}}_{i,h}}{(\bar{\sigma}_{i,h})^2} \left( \widehat{f}_{k,h}(s_h^i, a_h^i) - \widehat{f}_k^*(s_h^i, a_h^i) \right) \cdot \widehat{\eta}_h^i (V_{k,h+1} - V_{h+1}^*)$$

$$\leq O \left( \frac{\iota_2(k, h, \delta)}{\sqrt{\log \mathcal{N}_{\mathcal{F}}(\epsilon) + \log \mathcal{N}_{\epsilon}(K)}} + \frac{\iota_2(k, h, \delta)^2}{\gamma^{-2}} \right)$$

$$\cdot \sqrt{\sum_{\tau \in [k-1]} \frac{\widehat{\mathbb{1}}_{\tau,h}}{(\bar{\sigma}_{\tau,h})^2} \left( \widehat{f}_{k,h}(s_h^\tau, a_h^\tau) - \widehat{f}_k^*(s_h^\tau, a_h^\tau) \right)^2 + \lambda}$$

$$+ O(\epsilon k L/\alpha)^2) + O \left( \frac{\iota_2^2(k, h, \delta)}{\log \mathcal{N}_{\mathcal{F}}(\epsilon) + \log \mathcal{N}_{\epsilon}(K)} \right), \tag{E.15}$$

for all $k$ with at least probability $1 - \delta/H$.

Substituting (E.15) and (E.12) into (E.5), we can conclude that

$$\sum_{i \in [k-1]} \frac{\widehat{\mathbb{1}}_{i,h}}{(\bar{\sigma}_{i,h})^2} \left( \widehat{f}_{k,h}(s_h^i, a_h^i) - \widehat{f}_k^*(s_h^i, a_h^i) \right)^2$$

$$\leq O\left(\left(\frac{\iota_2(k,h,\delta)}{\sqrt{\log \mathcal{N}_{\mathcal{F}}(\epsilon) + \log \mathcal{N}_{\epsilon}(K)}} + \iota_2(k,h,\delta)^2 \cdot \gamma^{-2}\right)^2\right) + O\left(\left(\iota_1(k,h,\delta) + \iota_1(k,h,\delta)^2/\gamma^2\right)^2\right)$$

From the definition of $\gamma$ in (3.3), we can rewrite the upper bound of the squared loss $\sum_{i\in[k-1]} \frac{\widehat{\mathbb{1}}_{i,h}}{(\bar{\sigma}_{i,h})^2}\left(\widehat{f}_{k,h}(s_h^i, a_h^i) - \widehat{f}_k^*(s_h^i, a_h^i)\right)^2$ as follows:

$$\lambda + \sum_{i\in[k-1]} \frac{\widehat{\mathbb{1}}_{i,h}}{(\bar{\sigma}_{i,h})^2}\left(\widehat{f}_{k,h}(s_h^i, a_h^i) - \widehat{f}_k^*(s_h^i, a_h^i)\right)^2$$

$$\leq O\left(\log \frac{2k^2\left(2\log\frac{L^2 k}{\alpha^4} + 2\right)\cdot\left(\log(\frac{4L}{\alpha^2}) + 2\right)}{\delta/H}\right) \cdot [\log(\mathcal{N}_{\mathcal{F}}(\epsilon)) + 1] + O(\lambda) + O(\epsilon k L/\alpha^2).$$

$\square$

*Proof of Lemma D.5.* The proof is almost identical to the proof of Lemma D.4. $\square$

### E.2 PROOF OF OPTIMISM AND PESSIMISM
### E.2.1 PROOF OF LEMMA D.6

*Proof of Lemma D.6.* Acoording to the definition of $D_{\mathcal{F}}^2$ function, we have

$$\left(\widehat{f}_{k,h}(s,a) - \mathcal{T}_h V_{k,h+1}(s,a)\right)^2$$

$$\leq D_{\mathcal{F}_h}^2(z; z_{[k-1],h}, \bar{\sigma}_{[k-1],h}) \times \left(\lambda + \sum_{i=1}^{k-1} \frac{1}{(\bar{\sigma}_{i,h})^2}\left(\widehat{f}_{k,h}(s_h^i, a_h^i) - \mathcal{T}_h V_{k,h+1}(s_h^i, a_h^i)\right)^2\right)$$

$$\leq \widehat{\beta}_k^2 \times D_{\mathcal{F}_h}^2(z; z_{[k-1],h}, \bar{\sigma}_{[k-1],h}),$$

where the first inequality holds due the definition of $D_{\mathcal{F}}^2$ function with the Assumption 2.2 and the second inequality holds due to the events $\mathcal{E}_h^{\widehat{f}}$. Thus, we have

$$\left|\widehat{f}_{k,h}(s,a) - \mathcal{T}_h V_{k,h+1}(s,a)\right| \leq \widehat{\beta}_k D_{\mathcal{F}_h}(z; z_{[k-1],h}, \bar{\sigma}_{[k-1],h}).$$

With a similar argument, for the pessimitic value function $\widecheck{f}_{k,h}$, we have

$$\left(\widecheck{f}_{k,h}(s,a) - \mathcal{T}_h \widecheck{V}_{k,h}(s,a)\right)^2$$

$$\leq D_{\mathcal{F}_h}^2(z; z_{[k-1],h}, \bar{\sigma}_{[k-1],h}) \times \left(\lambda + \sum_{i=1}^{k-1} \frac{1}{(\bar{\sigma}_{i,h})^2}\left(\widecheck{f}_{k,h}(s_h^i, a_h^i) - \mathcal{T}_h \widecheck{V}_{k,h+1}(s_h^i, a_h^i)\right)^2\right)$$

$$\leq \widecheck{\beta}_k^2 \times D_{\mathcal{F}_h}^2(z; z_{[k-1],h}, \bar{\sigma}_{[k-1],h}),$$

where the first inequality holds due the definition of $D_{\mathcal{F}}^2$ function with the Assumption 2.2 and the second inequality holds due to the events $\mathcal{E}_h^{\widecheck{f}}$. In addition, we have

$$\left|\widecheck{f}_{k,h}(s,a) - \mathcal{T}_h \widecheck{V}_{k,h+1}(s,a)\right| \leq \widehat{\beta}_k D_{\mathcal{F}_h}(z; z_{[k-1],h}, \bar{\sigma}_{[k-1],h}).$$

Thus, we complete the proof of Lemma D.6. $\square$

### E.2.2 PROOF OF LEMMA D.7

*Proof of Lemma D.7.* We use induction to prove the optimistic and pessimistic property. First, we study the basic case with the last stage $H + 1$. In this situation, $Q_{k,H+1}(s,a) = Q_h^*(s,a) = \widecheck{Q}_{k,h}(s,a) = 0$ and $V_{k,h}(s) = V_h^*(s) = \widecheck{V}_{k,h}(s) = 0$ holds for all state $s \in \mathcal{S}$ and action $a \in \mathcal{A}$. Therefore, Lemma D.7 holds for the basic case (stage $H + 1$).

Second, if Lemma D.7 holds for stage $h + 1$, then we focus on the stage $h$. Notice that the event $\widetilde{\mathcal{E}}_h$ directly imlpies the event $\widetilde{\mathcal{E}}_{h+1}$. Therefore, according to the induction assumption, the following inequality holds for all state $s \in \mathcal{S}$ and episode $k \in [K]$.

$$V_{k,h+1}(s) \geq V_{h+1}^*(s) \geq \widecheck{V}_{k,h}(s). \tag{E.16}$$

Thus, for all episode $k \in [K]$ and state-action pair $(s,a) \in \mathcal{S} \times \mathcal{A}$, we have

$$
\begin{aligned}
&\widehat{f}_{k,h}(s,a) + b_{k,h}(s,a) - Q_h^*(s,a) \\
&\geq \mathcal{T}_h V_{k,h+1}(s,a) - \widehat{\beta}_k \cdot D_{\mathcal{F}_h}(z; z_{[k-1],h}, \bar{\sigma}_{[k-1],h}) + b_{k,h}(s,a) - Q_h^*(s,a) \\
&\geq \mathcal{T}_h V_{k,h+1}(s,a) - Q_h^*(s,a) \\
&= \mathbb{P}_h V_{k,h+1}(s,a) - \mathbb{P}_h V_h^*(s,a) \\
&\geq 0,
\end{aligned}
\tag{E.17}
$$

where the first inequality holds due to Lemma D.6, the second inequality holds due to the definition of the exploration bonus $b_{k,h}$ and the last inequality holds due to the (E.16). Therefore, the optimal value function $Q_h^*(s,a)$ is upper bounded by

$$
Q_h^*(s,a) \leq \min\left\{ \min_{1 \leq i \leq k} \widehat{f}_{i,h}(s,a) + b_{i,h}(s,a), 1 \right\} \leq Q_{k,h}(s,a),
\tag{E.18}
$$

where the first inequlity holds due to (E.17) with the fact that $Q_h^*(s,a) \leq 1$ and the second inequality holds due to the update rule of value function $Q_{k,h}$.

With a similar argument, for the pessimistic estimator $\breve{f}_{k,h}$, we have

$$
\begin{aligned}
&\breve{f}_{k,h}(s,a) - b_{k,h}(s,a) - Q_h^*(s,a) \\
&\leq \mathcal{T}_h \breve{V}_{k,h+1}(s,a) + \breve{\beta}_k \cdot D_{\mathcal{F}_h}(z; z_{[k-1],h}, \bar{\sigma}_{[k-1],h}) - b_{k,h}(s,a) - Q_h^*(s,a) \\
&\leq \mathcal{T}_h \breve{V}_{k,h+1}(s,a) - Q_h^*(s,a) \\
&= \mathbb{P}_h \breve{V}_{k,h+1}(s,a) - \mathbb{P}_h V_h^*(s,a) \\
&\leq 0,
\end{aligned}
\tag{E.19}
$$

where the first inequality holds due to Lemma D.6, the second inequality holds due to the definition of the exploration bonus $b_{k,h}$ and the last inequality holds due to the (E.19). Therefore, the optimal value function $Q_h^*(s,a)$ is lower bounded by

$$
Q_h^*(s,a) \geq \max\left\{ \max_{1 \leq i \leq k} \breve{f}_{i,h}(s,a) - b_{i,h}(s,a), 0 \right\} \geq \breve{Q}_{k,h}(s,a),
\tag{E.20}
$$

where the first inequality holds due to (E.19) with the fact that $Q_h^*(s,a) \geq 0$ and the second inequality holds due to the update rule of value function $\breve{Q}_{k,h}$.

Furthermore, for the value functions $V_{k,h}$ and $\breve{V}_{k,h}$, we have

$$
V_{k,h}(s) = \max_a Q_{k,h}(s,a) \geq \max_a Q_h^*(s,a) = V_h^*(s),
$$

$$
\breve{V}_{k,h}(s) = \max_a \breve{Q}_{k,h}(s,a) \leq \max_a Q_h^*(s,a) = V_h^*(s),
$$

where the first inequality holds due to (E.18) and the second inequality holds due to (E.20). Thus, by induction, we complete the proof of Lemma D.7. □

### E.3 Proof of Monotonic Variance Estimator
### E.3.1 Proof of Lemma D.8

*Proof of Lemma D.8.* Acoording to the definition of $D_{\mathcal{F}}^2$ function, we have

$$
\begin{aligned}
&\left(\widehat{f}_{k,h}(s,a) - \mathcal{T}_h V_{k,h+1}(s,a)\right)^2 \\
&\leq D_{\mathcal{F}_h}^2(z; z_{[k-1],h}, \bar{\sigma}_{[k-1],h}) \times \left(\lambda + \sum_{i=1}^{k-1} \frac{1}{(\bar{\sigma}_{i,h})^2} \left(\widehat{f}_{k,h}(s_h^i, a_h^i) - \mathcal{T}_h V_{k,h+1}(s_h^i, a_h^i)\right)^2\right) \\
&\leq \beta_k^2 \times D_{\mathcal{F}_h}^2(z; z_{[k-1],h}, \bar{\sigma}_{[k-1],h}),
\end{aligned}
$$

where the first inequality holds due the definition of $D_{\mathcal{F}}^2$ function with the Assumption 2.2 and the second inequality holds due to the events $\underline{\mathcal{E}}^{\widehat{f}}$. Thus, we have

$$
\left|\widehat{f}_{k,h}(s,a) - \mathcal{T}_h V_{k,h+1}(s,a)\right| \leq \beta_k D_{\mathcal{F}_h}(z; z_{[k-1],h}, \bar{\sigma}_{[k-1],h}).
$$

For the pessimitic value function $\check{f}_{k,h}$, we have

$$\left(\check{f}_{k,h}(s,a) - \mathcal{T}_h \check{V}_{k,h}(s,a)\right)^2$$

$$\leq D^2_{\mathcal{F}_h}(z; z_{[k-1],h}, \bar{\sigma}_{[k-1],h}) \times \left(\lambda + \sum_{i=1}^{k-1} \frac{1}{(\bar{\sigma}_{i,h})^2}\left(\check{f}_{k,h}(s_h^i, a_h^i) - \mathcal{T}_h \check{V}_{k,h+1}(s_h^i, a_h^i)\right)^2\right)$$

$$\leq \beta_k^2 \times D^2_{\mathcal{F}_h}(z; z_{[k-1],h}, \bar{\sigma}_{[k-1],h}),$$

where the first inequality holds due the definition of $D^2_{\mathcal{F}}$ function with the Assumption 2.2 and the second inequality holds due to the events $\underline{\mathcal{E}}^{\check{f}}$. In addition, we have

$$\left|\check{f}_{k,h}(s,a) - \mathcal{T}_h \check{V}_{k,h+1}(s,a)\right| \leq \beta_k D_{\mathcal{F}_h}(z; z_{[k-1],h}, \bar{\sigma}_{[k-1],h}).$$

With a similar argument, for the second-order estimator $\widetilde{f}_{k,h}$, we have

$$\left(\widetilde{f}_{k,h}(s,a) - \mathcal{T}_h^2 V_{k,h}(s,a)\right)^2$$

$$\leq D^2_{\mathcal{F}_h}(z; z_{[k-1],h}, \bar{\sigma}_{[k-1],h}) \times \left(\lambda + \sum_{i=1}^{k-1} \frac{1}{(\bar{\sigma}_{i,h})^2}\left(\widetilde{f}_{k,h}(s_h^i, a_h^i) - \mathcal{T}_h^2 V_{k,h+1}(s_h^i, a_h^i)\right)^2\right)$$

$$\leq \widetilde{\beta}_k^2 \times D^2_{\mathcal{F}_h}(z; z_{[k-1],h}, \bar{\sigma}_{[k-1],h}),$$

where the first inequality holds due the definition of $D^2_{\mathcal{F}}$ function with the Assumption 2.2 and the second inequality holds due to the events $\underline{\mathcal{E}}^{\widetilde{f}}$. Therefore, we have

$$\left|\widetilde{f}_{k,h}(s,a) - \mathcal{T}_h^2 V_{k,h}(s,a)\right| \leq \widetilde{\beta}_k D_{\mathcal{F}_h}(z; z_{[k-1],h}, \bar{\sigma}_{[k-1],h}).$$

Now, we complete the proof of Lemma D.8. □

### E.3.2 PROOF OF LEMMA D.9

*Proof of Lemma D.9.* First, according to Lemma D.8, we have

$$\left|[\bar{\mathbb{V}}_h V_{k,h+1}](s_h^k, a_h^k) - [\mathbb{V}_h V_{k,h+1}](s_h^k, a_h^k)\right|$$

$$= \left|\widetilde{f}_{k,h} - \widehat{f}_{k,h}^2 - [\mathbb{P}_h V_{k,h+1}^2](s_h^k, a_h^k) + \left([\mathbb{P}_h V_{k,h+1}](s_h^k, a_h^k)\right)^2\right|$$

$$\leq \left|\widehat{f}_{k,h}^2 - \left([\mathbb{P}_h V_{k,h+1}](s_h^k, a_h^k)\right)^2\right| + \left|\widetilde{f}_{k,h} - [\mathbb{P}_h V_{k,h+1}^2](s_h^k, a_h^k)\right|$$

$$\leq 2L\left|\widehat{f}_{k,h} - \left([\mathbb{P}_h V_{k,h+1}](s_h^k, a_h^k)\right)\right| + \left|\widetilde{f}_{k,h} - [\mathbb{P}_h V_{k,h+1}^2](s_h^k, a_h^k)\right|$$

$$\leq (2L\beta_k + \widetilde{\beta}_k) D_{\mathcal{F}_h}(z; z_{[k-1],h}, \bar{\sigma}_{[k-1],h})$$

$$= E_{k,h}, \tag{E.21}$$

where the first inequality holds due to $|a+b| \leq |a| + |b|$, the second inequality holds due to $|a^2 - b^2| = |a-b| \cdot |a+b| \leq |a-b| \cdot 2\max(|a|, |b|)$ and the last inequalirtnholds due to Lemma D.8.

For the difference between variances $[\mathbb{V}_h V_{k,h+1}](s_h^k, a_h^k)$ and $[\mathbb{V}_h V_{h+1}^*](s_h^k, a_h^k)$, it can be upper bounded by

$$\left|[\mathbb{V}_h V_{k,h+1}](s_h^k, a_h^k) - [\mathbb{V}_h V_{h+1}^*](s_h^k, a_h^k)\right|$$

$$= \left|[\mathbb{P}_h V_{k,h+1}^2](s_h^k, a_h^k) - \left([\mathbb{P}_h V_{k,h+1}](s_h^k, a_h^k)\right)^2 - [\mathbb{P}_h (V_{h+1}^*)^2](s_h^k, a_h^k) + \left([\mathbb{P}_h V_{h+1}^*](s_h^k, a_h^k)\right)^2\right|$$

$$\leq \left|[\mathbb{P}_h V_{k,h+1}^2](s_h^k, a_h^k) - [\mathbb{P}_h (V_{h+1}^*)^2](s_h^k, a_h^k)\right| + \left|\left([\mathbb{P}_h V_{k,h+1}](s_h^k, a_h^k)\right)^2 - \left([\mathbb{P}_h V_{h+1}^*](s_h^k, a_h^k)\right)^2\right|$$

$$\leq 4\left([\mathbb{P}_h V_{k,h+1}](s_h^k, a_h^k) - [\mathbb{P}_h V_{h+1}^*](s_h^k, a_h^k)\right)$$

$$\leq \left([\mathbb{P}_h V_{k,h+1}](s_h^k, a_h^k) - [\mathbb{P}_h \check{V}_{k,h+1}](s_h^k, a_h^k)\right)$$

$$\leq \widehat{f}_{k,h}(s_h^k, a_h^k) - \check{f}_{k,h}(s_h^k, a_h^k) + 2\beta_k D_{\mathcal{F}_h}(z; z_{[k-1],h}, \bar{\sigma}_{[k-1],h}) \tag{E.22}$$

where the first inequality holds due to $|a+b| \leq |a| + |b|$, the second inequality holds due to $1 \geq V_{k,h+1}(\cdot) \geq V_{h+1}^*(\cdot) \geq 0$ (Lemma D.7), the third inequality holds due to $V_{h+1}^*(\cdot) \geq \check{V}_{k,h+1}(\cdot)$

(Lemma D.7) and the last inequality holds due to Lemma D.8. Combining the results in (E.21) and (E.22) with the fact that $0 \leq V_{k,h+1}(\cdot), V_{h+1}^*(\cdot) \leq 1$, we have

$$
\begin{aligned}
&\left| [\bar{\mathbb{V}}_h V_{k,h+1}](s_h^k, a_h^k) - [\mathbb{V}_h V_{h+1}^*](s_h^k, a_h^k) \right| \\
&\leq \left| [\bar{\mathbb{V}}_h V_{k,h+1}](s_h^k, a_h^k) - [\mathbb{V}_h V_{k,h+1}](s_h^k, a_h^k) \right| + \left| [\mathbb{V}_h V_{k,h+1}](s_h^k, a_h^k) - [\mathbb{V}_h V_{h+1}^*](s_h^k, a_h^k) \right| \\
&\leq E_{k,h} + F_{k,h}.
\end{aligned}
$$

Thus, we complete the proof of Lemma D.9. $\qquad\square$

### E.3.3 Proof of Lemma D.10

*Proof of Lemma D.10.* On the events $\underline{\mathcal{E}}^{\widehat{f}}$ and $\mathcal{E}_{h+1}^{\widehat{f}}$, we have

$$
\begin{aligned}
&\left[ \mathbb{V}_h (V_{i,h+1} - V_{h+1}^*) \right](s_h^k, a_h^k) \\
&= [\mathbb{P}_h (V_{i,h+1} - V_{h+1}^*)^2](s_h^k, a_h^k) - \left( [\mathbb{P}_h (V_{i,h+1} - V_{h+1}^*)](s_h^k, a_h^k) \right)^2 \\
&\leq [\mathbb{P}_h (V_{i,h+1} - V_{h+1}^*)^2](s_h^k, a_h^k) \\
&\leq 2 [\mathbb{P}_h (V_{i,h+1} - V_{h+1}^*)](s_h^k, a_h^k) \\
&\leq 2 \left( [\mathbb{P}_h V_{i,h+1}](s_h^k, a_h^k) - [\mathbb{P}_h \check{V}_{k,h+1}](s_h^k, a_h^k) \right) \\
&\leq 2 \left( [\mathbb{P}_h V_{k,h+1}](s_h^k, a_h^k) - [\mathbb{P}_h \check{V}_{k,h+1}](s_h^k, a_h^k) \right),
\end{aligned}
\tag{E.23}
$$

where the first equation holds since the reward function is deterministic, the second inequality holds due to Lemma D.7 with the fact that $0 \leq V_{i,h+1}(s), V_{h+1}^*(s) \leq 1$ and the third inequality holds due to Lemma D.7 and the last inequality holds due to the fact that $V_{k,h+1} \geq V_{i,h+1}$.

With a similar argument, on the events $\underline{\mathcal{E}}^{\check{f}}$ and $\mathcal{E}_{h+1}^{\check{f}}$, we have

$$
\begin{aligned}
&\left[ \mathbb{V}_h (V_{h+1}^* - \check{V}_{i,h+1}) \right](s_h^k, a_h^k) \\
&= [\mathbb{P}_h (V_{h+1}^* - \check{V}_{i,h+1})^2](s_h^k, a_h^k) - \left( [\mathbb{P}_h (V_{h+1}^* - \check{V}_{i,h+1})](s_h^k, a_h^k) \right)^2 \\
&\leq [\mathbb{P}_h (V_{h+1}^* - \check{V}_{i,h+1})^2](s_h^k, a_h^k) \\
&\leq 2 [\mathbb{P}_h (V_{h+1}^* - \check{V}_{i,h+1})](s_h^k, a_h^k) \\
&\leq 2 \left( [\mathbb{P}_h V_{k,h+1}](s_h^k, a_h^k) - [\mathbb{P}_h \check{V}_{i,h+1}](s_h^k, a_h^k) \right) \\
&\leq 2 \left( [\mathbb{P}_h V_{k,h+1}](s_h^k, a_h^k) - [\mathbb{P}_h \check{V}_{k,h+1}](s_h^k, a_h^k) \right),
\end{aligned}
\tag{E.24}
$$

where the first inequality holds since the reward function is deterministic, the second and third inequality holds due to Lemma D.7 with the fact that $0 \leq \check{V}_{i,h+1}(s), V_{h+1}^*(s) \leq H$, the last inequality the fact $\check{V}_{k,h+1}(s) \leq \check{V}_{i,h+1}(s)$. For both variances $\left[ \mathbb{V}_h (V_{i,h+1} - V_{h+1}^*) \right](s_h^k, a_h^k)$ and $\left[ \mathbb{V}_h (V_{h+1}^* - \check{V}_{i,h+1}) \right](s_h^k, a_h^k)$, they are upper bounded by

$$
\begin{aligned}
&2 \left( [\mathbb{P}_h V_{k,h+1}](s_h^k, a_h^k) - [\mathbb{P}_h \check{V}_{k,h+1}](s_h^k, a_h^k) \right) \\
&= 2 \mathcal{T}_h V_{k,h+1}(s_h^k, a_h^k) - 2 \mathcal{T}_h \check{V}_{k,h+1}(s_h^k, a_h^k) \\
&\leq 2 \widehat{f}_{k,h}(s_h^k, a_h^k) - 2 \check{f}_{k,h}(s_h^k, a_h^k) + 4 \beta_k D_{\mathcal{F}_h}(z; z_{[k-1],h}, \bar{\sigma}_{[k-1],h}),
\end{aligned}
\tag{E.25}
$$

where the first inequality holds due to Lemma D.8 and the second inequality holds due to the definition of $F_{k,h}$. Substituting the result in (E.25) into (E.23), (E.24) and combining the fact that $\left[ \mathbb{V}_h (V_{i,h+1} - V_{h+1}^*) \right](s_h^k, a_h^k), \left[ \mathbb{V}_h (V_{h+1}^* - \check{V}_{i,h+1}) \right](s_h^k, a_h^k) \leq 1$, we finish the proof of Lemma D.10. $\qquad\square$

### E.4 Proof of Lemmas in Section D.4
### E.4.1 Proof of Lemma D.11

*Proof of Lemma D.11.* We use induction to shows that the conclusion in Lemma D.7 and events $\mathcal{E}_h^{\widehat{f}}, \mathcal{E}_H^{\check{f}}$ holds for each stage $h \in [H]$. First, for the basic situation (stage $H + 1$), $Q_{k,H+1}(s,a) = Q_h^*(s,a) = \check{Q}_{k,h}(s,a) = 0$ and $V_{k,h}(s) = V_h^*(s) = \check{V}_{k,h}(s) = 0$ holds for all state $s \in \mathcal{S}$ and action $a \in \mathcal{A}$. Therefore, Lemma D.7 holds for the basic case (stage $H + 1$)

Second, if Lemma D.7 holds for stage $h + 1$, then we focuse on the stage $h$. According to Lemmas D.10 and Lemma D.9, we have the following inequalitys:

$$\sigma_{i,h}^2 = [\bar{\mathbb{V}}_h V_{i,h+1}](s_h^i, a_h^i) + E_{i,h} + D_{i,h} \geq [\bar{\mathbb{V}}_h V_{h+1}^*](s_h^i, a_h^i),$$

$$\sigma_{i,h}^2 \geq F_{i,h} \geq (\log \mathcal{N}_{\mathcal{F}}(\epsilon) + \log \mathcal{N}_\epsilon(K)) \cdot \left[\mathbb{V}_h(V_{k,h+1} - V_{h+1}^*)\right](s_h^i, a_h^i),$$

$$\sigma_{i,h}^2 \geq F_{i,h} \geq (\log \mathcal{N}_{\mathcal{F}}(\epsilon) + \log \mathcal{N}_\epsilon(K)) \cdot \left[\mathbb{V}_h(V_{h+1}^* - \check{V}_{k,h+1})\right](s_h^i, a_h^i),$$

where the first inequality holds due to Lemma D.9, the second and third inequality holds due to Lemma D.10. Thus, the indicator function in events $\bar{\mathcal{E}}_h^{\hat{f}}$ and $\bar{\mathcal{E}}_h^{\check{f}}$ holds, which implies events $\mathcal{E}_h^{\hat{f}}$, $\mathcal{E}_H^{\check{f}}$ holds. Furthermore, when events $\mathcal{E}_h^{\hat{f}}$, $\mathcal{E}_H^{\check{f}}$ holds, then Lemma D.7 holds for stage $h$. Thus, we complete the proof of Lemma D.11 by induction. $\qquad \square$

### E.4.2 PROOF OF LEMMA D.13

*Proof of Lemma D.13.* For each stage $h$, we divide the episodes $\{1, 2, .., K\}$ to the following sets:

$$\mathcal{I}_1 = \{k \in [K] : D_{\mathcal{F}_h}(z; z_{[k-1],h}, \bar{\sigma}_{[k-1],h}) / \bar{\sigma}_{k,h} \geq 1\},$$

$$\mathcal{I}_2 = \{k \in [K] : D_{\mathcal{F}_h}(z; z_{[k-1],h}, \bar{\sigma}_{[k-1],h}) / \bar{\sigma}_{k,h} < 1, \bar{\sigma}_{k,h} = \sigma_{k,h}\},$$

$$\mathcal{I}_3 = \{k \in [K] : D_{\mathcal{F}_h}(z; z_{[k-1],h}, \bar{\sigma}_{[k-1],h}) / \bar{\sigma}_{k,h} < 1, \bar{\sigma}_{k,h} = \alpha\},$$

$$\mathcal{I}_4 = \{k \in [K] : D_{\mathcal{F}_h}(z; z_{[k-1],h}, \bar{\sigma}_{[k-1],h}) / \bar{\sigma}_{k,h} < 1, \bar{\sigma}_{k,h} = \gamma \times D_{\mathcal{F}_h}^{1/2}(z; z_{[k-1],h}, \bar{\sigma}_{[k-1],h})\}.$$

The number of episode in set $\mathcal{I}_1$ is upper bounded by

$$|\mathcal{I}_1| = \sum_{k \in \mathcal{I}_1} \min \left( D_{\mathcal{F}_h}^2(z; z_{[k-1],h}, \bar{\sigma}_{[k-1],h}) / \bar{\sigma}_{k,h}^2, 1 \right) \leq \dim_{\alpha, K}(\mathcal{F}_h),$$

where the equation holds due to $D_{\mathcal{F}_h}^2(z; z_{[k-1],h}, \bar{\sigma}_{[k-1],h}) / \bar{\sigma}_{k,h}^2 \geq 1$ and the inequality holds due to the definition of Generalized Eluder dimension. Thus, for set $\mathcal{I}_1$, the summation of confidence radius is upper bounded by

$$\sum_{k \in \mathcal{I}_1} \min \left( \beta D_{\mathcal{F}_h}(z; z_{[k-1],h}, \bar{\sigma}_{[k-1],h}), 1 \right) \leq |\mathcal{I}_1| \leq \dim_{\alpha, K}(\mathcal{F}_h). \tag{E.26}$$

For set $\mathcal{I}_2$, the summation of confidence radius is upper bounded by

$$\sum_{k \in \mathcal{I}_2} \min \left( \beta D_{\mathcal{F}_h}(z; z_{[k-1],h}, \bar{\sigma}_{[k-1],h}), 1 \right)$$

$$\leq \sum_{k \in \mathcal{I}_2} \beta D_{\mathcal{F}_h}(z; z_{[k-1],h}, \bar{\sigma}_{[k-1],h})$$

$$\leq \beta \sqrt{\sum_{k \in \mathcal{I}_2} \sigma_{k,h}^2} \cdot \sqrt{\sum_{k \in \mathcal{I}_2} D_{\mathcal{F}_h}^2(z; z_{[k-1],h}, \bar{\sigma}_{[k-1],h}) / \bar{\sigma}_{k,h}^2}$$

$$\leq \beta \sqrt{\dim_{\alpha, K}(\mathcal{F}_h)} \sqrt{\sum_{k \in \mathcal{I}_2} \sigma_{k,h}^2}, \tag{E.27}$$

where the second inequality holds due to Cauchy-Schwartz inequality with $\sigma_{k,h} = \bar{\sigma}_{k,h}$ and the last inequality holds due to the definition of Generalized Eluder dimension with the fact that $D_{\mathcal{F}_h}(z; z_{[k-1],h}, \bar{\sigma}_{[k-1],h}) / \bar{\sigma}_{k,h} < 1$.

With a similar argument, the summation of confidence radius over set $\mathcal{I}_3$ is upper bounded by

$$\sum_{k \in \mathcal{I}_3} \min \left( \beta D_{\mathcal{F}_h}(z; z_{[k-1],h}, \bar{\sigma}_{[k-1],h}), 1 \right)$$

$$\leq \sum_{k \in \mathcal{I}_3} \beta D_{\mathcal{F}_h}(z; z_{[k-1],h}, \bar{\sigma}_{[k-1],h})$$

$$\leq \beta \sqrt{\sum_{k \in \mathcal{I}_3} \alpha^2} \cdot \sqrt{\sum_{k \in \mathcal{I}_3} D_{\mathcal{F}_h}^2(z; z_{[k-1],h}, \bar{\sigma}_{[k-1],h}) / \bar{\sigma}_{k,h}^2}$$

$$\leq \beta \sqrt{\dim_{\alpha,K}(\mathcal{F}_h)} \sqrt{\sum_{k \in \mathcal{I}_3} \alpha^2}, \tag{E.28}$$

where the second inequality holds due to Cauchy-Schwartz inequality with $\bar{\sigma}_{k,h} = \alpha$ and the last inequality holds due to the definition of Generalized Eluder dimension with the fact that $D_{\mathcal{F}_h}(z; z_{[k-1],h}, \bar{\sigma}_{[k-1],h})/\bar{\sigma}_{k,h} < 1$.

Finally, the summation of confidence raduis over set $\mathcal{I}_4$ is upper bounded by With a similar argument, the summation of confidence radius over set $\mathcal{I}_3$ is upper bounded by

$$\sum_{k \in \mathcal{I}_4} \min\left(\beta D_{\mathcal{F}_h}(z; z_{[k-1],h}, \bar{\sigma}_{[k-1],h}), 1\right)$$

$$\leq \sum_{k \in \mathcal{I}_4} \beta D_{\mathcal{F}_h}(z; z_{[k-1],h}, \bar{\sigma}_{[k-1],h})$$

$$= \sum_{k \in \mathcal{I}_4} \beta \gamma^2 D_{\mathcal{F}_h}^2(z; z_{[k-1],h}, \bar{\sigma}_{[k-1],h})/\bar{\sigma}_{k,h}^2$$

$$\leq \beta \gamma^2 \dim_{\alpha,K}(\mathcal{F}_h), \tag{E.29}$$

where the first equation holds due to $\bar{\sigma}_{k,h} = \gamma \times D_{\mathcal{F}_h}^{1/2}(z; z_{[k-1],h}, \bar{\sigma}_{[k-1],h})$ and the last inequality holds due to the definition of Generalized Eluder dimension with $D_{\mathcal{F}_h}(z; z_{[k-1],h}, \bar{\sigma}_{[k-1],h})/\bar{\sigma}_{k,h} < 1$.

Combining the results in (E.26), (E.27), (E.28) and (E.29), we have

$$\sum_{k=1}^{K} \min\left(\beta D_{\mathcal{F}_h}(z; z_{[k-1],h}, \bar{\sigma}_{[k-1],h}), 1\right)$$

$$\leq (1 + \beta\gamma^2) \dim_{\alpha,K}(\mathcal{F}_h) + 2\beta \sqrt{\dim_{\alpha,K}(\mathcal{F}_h)} \sqrt{\sum_{k=1}^{K} (\sigma_{k,h}^2 + \alpha^2)}.$$

Thus, we complete the proof of Lemma D.13. $\qquad\square$

### E.4.3 PROOF OF LEMMA D.14

*Proof of Lemma D.14.* First, for each stage $h \in [H]$ and episode $k \in [K]$, the gap between $V_{k,h}(s_h^k)$ and $V_h^{\pi^k}(s_h^k)$ can be decomposed as:

$$V_{k,h}(s_h^k) - V_h^{\pi^k}(s_h^k)$$

$$= Q_{k,h}(s_h^k, a_h^k) - Q_h^{\pi^k}(s_h^k, a_h^k)$$

$$\leq \min\left(\widehat{f}_{k_{\text{last}},h}(s,a) + b_{k_{\text{last}},h}(s,a), 1\right) - \mathcal{T}_h V_{k,h+1}(s_h^k, a_h^k)$$

$$\quad + \mathcal{T}_h V_{k,h+1}(s_h^k, a_h^k) - \mathcal{T}_h V_{h+1}^{\pi^k}(s_h^k, a_h^k)$$

$$\leq \left[\mathbb{P}_h(V_{k,h+1} - V_{h+1}^{\pi^k})\right](s_h^k, a_h^k) + \min\left(\widehat{\beta}_{k_{\text{last}}} D_{\mathcal{F}_h}(z; z_{[k_{\text{last}}-1],h}, \bar{\sigma}_{[k_{\text{last}}-1],h}), 1\right) + \min\left(b_{k_{\text{last}},h}(s_h^k, a_h^k), 1\right)$$

$$\leq \left[\mathbb{P}_h(V_{k,h+1} - V_{h+1}^{\pi^k})\right](s_h^k, a_h^k) + 2C \cdot \min\left(\widehat{\beta}_{k_{\text{last}}} D_{\mathcal{F}_h}(z; z_{[k_{\text{last}}-1],h}, \bar{\sigma}_{[k_{\text{last}}-1],h}), 1\right)$$

$$\leq \left[\mathbb{P}_h(V_{k,h+1} - V_{h+1}^{\pi^k})\right](s_h^k, a_h^k) + 2C(1+\chi) \cdot \min\left(\widehat{\beta}_k D_{\mathcal{F}_h}(z; z_{[k-1],h}, \bar{\sigma}_{[k-1],h}), 1\right)$$

$$= V_{k,h+1}(s_{h+1}^k) - V_{h+1}^{\pi^k}(s_{h+1}^k) + \left[\mathbb{P}_h(V_{k,h+1} - V_{h+1}^{\pi^k})\right](s_h^k, a_h^k) - \left(V_{k,h+1}(s_{h+1}^k) - V_{h+1}^{\pi^k}(s_{h+1}^k)\right)$$

$$\quad + 2C(1+\chi) \cdot \min\left(\widehat{\beta}_k D_{\mathcal{F}_h}(z; z_{[k-1],h}, \bar{\sigma}_{[k-1],h}), 1\right), \tag{E.30}$$

where the first inequality holds due to the definition of value function $Q_{k,h}(s_h^k, a_h^k)$, the second inequality holds due to Lemma D.6, the third inequality holds due to $b_{k_{\text{last}},h}(s_h^k, a_h^k) \leq C \cdot D_{\mathcal{F}_h}(z; z_{[k_{\text{last}}-1],h}, \bar{\sigma}_{[k_{\text{last}}-1],h})$ and the last inequality holds due to Lemma F.2. Taking a summation of (E.30) over all episode $k \in [K]$ and stage $h' \geq h$, we have

$$\sum_{k=1}^{K} \left(V_{k,h}(s_h^k) - V_h^{\pi^k}(s_h^k)\right)$$

$$\leq \sum_{k=1}^{K} \sum_{h'=h}^{H} \left( \left[ \mathbb{P}_h (V_{k,h+1} - V_{h+1}^{\pi^k}) \right](s_h^k, a_h^k) - \left( V_{k,h+1}(s_{h+1}^k) - V_{h+1}^{\pi^k}(s_{h+1}^k) \right) \right)$$

$$+ \sum_{k=1}^{K} \sum_{h'=h}^{H} 2C(1+\chi) \cdot \min \left( \widehat{\beta}_k D_{\mathcal{F}_h}(z; z_{[k-1],h}, \bar{\sigma}_{[k-1],h}), 1 \right)$$

$$\leq \sum_{k=1}^{K} \sum_{h'=h}^{H} 2C(1+\chi) \cdot \min \left( \widehat{\beta}_k D_{\mathcal{F}_h}(z; z_{[k-1],h}, \bar{\sigma}_{[k-1],h}), 1 \right) + \sqrt{2HK \log(H/\delta)}$$

$$\leq \sum_{h'=h}^{H} 2C(1+\chi)(1+\widehat{\beta}_k \gamma^2) \dim_{\alpha,K}(\mathcal{F}_{h'}) + \sum_{h'=h}^{H} 4C(1+\chi)\widehat{\beta}_k \sqrt{\dim_{\alpha,K}(\mathcal{F}_{h'})} \sqrt{\sum_{k=1}^{K} (\sigma_{k,h'}^2 + \alpha^2)}$$

$$+ \sqrt{2HK \log(H/\delta)}$$

$$\leq 2CH(1+\chi)(1+\widehat{\beta}_k \gamma^2) \dim_{\alpha,K}(\mathcal{F}_h) + 4C(1+\chi)\widehat{\beta}_k \sqrt{\sum_{h'=h}^{H} \dim_{\alpha,K}(\mathcal{F}_{h'})} \sqrt{\sum_{k=1}^{K} \sum_{h'=h}^{H} (\sigma_{k,h'}^2 + \alpha^2)}$$

$$+ \sqrt{2HK \log(H/\delta)}$$

$$\leq 2CH(1+\chi)(1+\widehat{\beta}_k \gamma^2) \dim_{\alpha,K}(\mathcal{F}_h) + 4C(1+\chi)\widehat{\beta}_k \sqrt{\dim_{\alpha,K}(\mathcal{F})} \sqrt{H \sum_{k=1}^{K} \sum_{h=1}^{H} (\sigma_{k,h}^2 + \alpha^2)}$$

$$+ \sqrt{2HK \log(H/\delta)}, \tag{E.31}$$

where the first inequality holds due to (E.30), the second inequality holds due to event $\mathcal{E}_1$, the third inequality holds due to Lemma D.13, the fourth inequality holds due to Cauchy-Schwartz inequality and the last inequality holds due to $\sum_{h'=h}^{H} \dim_{\alpha,K}(\mathcal{F}_{h'}) \leq \sum_{h'=1}^{H} \dim_{\alpha,K}(\mathcal{F}_{h'}) = H \dim_{\alpha,K}(\mathcal{F})$. Furthermore, taking a summation of (E.31), we have

$$\sum_{k=1}^{K} \sum_{h=1}^{H} \left[ \mathbb{P}_h (V_{k,h+1} - V_{h+1}^{\pi^k}) \right](s_h^k, a_h^k)$$

$$= \sum_{k=1}^{K} \sum_{h=1}^{H} \left( V_{k,h+1}(s_{h+1}^k) - V_{h+1}^{\pi^k}(s_{h+1}^k) \right)$$

$$+ \sum_{k=1}^{K} \sum_{h=1}^{H} \left( \left[ \mathbb{P}_h (V_{k,h+1} - V_{h+1}^{\pi^k}) \right](s_h^k, a_h^k) - \left( V_{k,h+1}(s_{h+1}^k) - V_{h+1}^{\pi^k}(s_{h+1}^k) \right) \right)$$

$$\leq \sum_{k=1}^{K} \sum_{h=1}^{H} \left( V_{k,h+1}(s_{h+1}^k) - V_{h+1}^{\pi^k}(s_{h+1}^k) \right) + \sqrt{2HK \log(H/\delta)}$$

$$\leq 2CH^2(1+\chi)(1+\widehat{\beta}_k \gamma^2) \dim_{\alpha,K}(\mathcal{F}) + 4CH(1+\chi)\widehat{\beta}_k \sqrt{\dim_{\alpha,K}(\mathcal{F}_h)} \sqrt{H \sum_{k=1}^{K} \sum_{h=1}^{H} (\sigma_{k,h}^2 + \alpha^2)}$$

$$+ 2\sqrt{2H^3 K \log(H/\delta)},$$

where the first inequality holds due to event $\mathcal{E}_1$ and the second inequality holds due to (E.31). Thus, we complete the proof of Lemma D.14. □

### E.4.4  PROOF OF LEMMA D.15

*Proof of Lemma D.15.* Similar to the proof of Lemma D.14, for each stage $h \in [H]$ and episode $k \in [K]$, the gap between $V_{k,h}(s_h^k)$ and $\check{V}_{k,h}(s_h^k)$ can be decomposed as:

$$V_{k,h}(s_h^k) - \check{V}_{k,h}(s_h^k)$$

$$\leq Q_{k,h}(s_h^k, a_h^k) - \check{Q}_{k,h}(s_h^k, a_h^k)$$

$$\leq \min \left( \widehat{f}_{k_{\text{last}},h}(s,a) + b_{k_{\text{last}},h}(s,a), 1 \right) - \mathcal{T}_h V_{k,h+1}(s_h^k, a_h^k)$$

$$
\begin{aligned}
&- \max\left(\check{f}_{k_{\mathrm{last}},h}(s,a) - b_{k_{\mathrm{last}},h}(s,a), 0\right) + \mathcal{T}_h \check{V}_{k,h+1}(s_h^k, a_h^k) \\
&\quad + \mathcal{T}_h V_{k,h+1}(s_h^k, a_h^k) - \mathcal{T}_h \check{V}_{k,h+1}(s_h^k, a_h^k) \\
&\leq \left[\mathbb{P}_h(V_{k,h+1} - \check{V}_{k,h+1})\right](s_h^k, a_h^k) + 2 \cdot \min\left(\widehat{\beta}_{k_{\mathrm{last}}} D_{\mathcal{F}_h}(z; z_{[k_{\mathrm{last}}-1],h}, \bar{\sigma}_{[k_{\mathrm{last}}-1],h}), 1\right) \\
&\quad + 2 \cdot \min\left(b_{k_{\mathrm{last}},h}(s_h^k, a_h^k), 1\right) \\
&\leq \left[\mathbb{P}_h(V_{k,h+1} - \check{V}_{k,h+1})\right](s_h^k, a_h^k) + 4C \cdot \min\left(\widehat{\beta}_{k_{\mathrm{last}}} D_{\mathcal{F}_h}(z; z_{[k_{\mathrm{last}}-1],h}, \bar{\sigma}_{[k_{\mathrm{last}}-1],h}), 1\right) \\
&\leq \left[\mathbb{P}_h(V_{k,h+1} - \check{V}_{k,h+1})\right](s_h^k, a_h^k) + 4C(1+\chi) \cdot \min\left(\widehat{\beta}_k D_{\mathcal{F}_h}(z; z_{[k-1],h}, \bar{\sigma}_{[k-1],h}), 1\right) \\
&= V_{k,h+1}(s_{h+1}^k) - \check{V}_{k,h+1}(s_{h+1}^k) + \left[\mathbb{P}_h(V_{k,h+1} - \check{V}_{k,h+1})\right](s_h^k, a_h^k) - \left(V_{k,h+1}(s_{h+1}^k) - \check{V}_{k,h+1}(s_{h+1}^k)\right) \\
&\quad + 4C(1+\chi) \cdot \min\left(\widehat{\beta}_k D_{\mathcal{F}_h}(z; z_{[k-1],h}, \bar{\sigma}_{[k-1],h}), 1\right),
\end{aligned}
\tag{E.32}
$$

where the first and second inequalities holds due to the definition of $\check{V}_{k,h}(s_h^k)$ and $V_{k,h}(s_h^k)$, the third inequality holds due to Lemma D.6 with $\widehat{\beta}_{k_{\mathrm{last}}} = \check{\beta}_{k_{\mathrm{last}}}$, the fourth inequality holds due to $b_{k_{\mathrm{last}},h}(s_h^k, a_h^k) \leq C \cdot D_{\mathcal{F}_h}(z; z_{[k_{\mathrm{last}}-1],h}, \bar{\sigma}_{[k_{\mathrm{last}}-1],h})$ and the last inequality holds due to Lemma F.2. Taking a summation of (E.32) over all episode $k \in [K]$ and stage $h' \geq h$, we have

$$
\begin{aligned}
&\sum_{k=1}^{K} \left(V_{k,h}(s_h^k) - \check{V}_{k,h}(s_h^k)\right) \\
&\leq \sum_{k=1}^{K} \sum_{h'=h}^{H} \left(\left[\mathbb{P}_h(V_{k,h+1} - \check{V}_{k,h+1})\right](s_h^k, a_h^k) - \left(V_{k,h+1}(s_{h+1}^k) - \check{V}_{k,h+1}(s_{h+1}^k)\right)\right) \\
&\quad + \sum_{k=1}^{K} \sum_{h'=h}^{H} 4C(1+\chi) \cdot \min\left(\widehat{\beta}_k D_{\mathcal{F}_h}(z; z_{[k-1],h}, \bar{\sigma}_{[k-1],h}), 1\right) \\
&\leq \sum_{k=1}^{K} \sum_{h'=h}^{H} 4C(1+\chi) \cdot \min\left(\widehat{\beta}_k D_{\mathcal{F}_h}(z; z_{[k-1],h}, \bar{\sigma}_{[k-1],h}), 1\right) + \sqrt{2HK \log(H/\delta)} \\
&\leq \sum_{h'=h}^{H} 4C(1+\chi)(1 + \widehat{\beta}_k \gamma^2) \dim_{\alpha,K}(\mathcal{F}_{h'}) + \sum_{h'=h}^{H} 8C(1+\chi)\widehat{\beta}_k \sqrt{\dim_{\alpha,K}(\mathcal{F}_{h'})} \sqrt{\sum_{k=1}^{K}(\sigma_{k,h'}^2 + \alpha^2)} \\
&\quad + \sqrt{2HK \log(H/\delta)} \\
&\leq 4CH(1+\chi)(1 + \widehat{\beta}_k \gamma^2) \dim_{\alpha,K}(\mathcal{F}_h) + 8C(1+\chi)\widehat{\beta}_k \sqrt{\sum_{h'=h}^{H} \dim_{\alpha,K}(\mathcal{F}_{h'})} \sqrt{\sum_{k=1}^{K} \sum_{h'=h}^{H}(\sigma_{k,h'}^2 + \alpha^2)} \\
&\quad + \sqrt{2HK \log(H/\delta)} \\
&\leq 4CH(1+\chi)(1 + \widehat{\beta}_k \gamma^2) \dim_{\alpha,K}(\mathcal{F}_h) + 8C(1+\chi)\widehat{\beta}_k \sqrt{\dim_{\alpha,K}(\mathcal{F})} \sqrt{H \sum_{k=1}^{K} \sum_{h=1}^{H}(\sigma_{k,h}^2 + \alpha^2)} \\
&\quad + \sqrt{2HK \log(H/\delta)},
\end{aligned}
\tag{E.33}
$$

where the first inequality holds due to (E.32), the second inequality holds due to event $\mathcal{E}_2$, the third inequality holds due to Lemma D.13, the fourth inequality holds due to Cauchy-Schwartz inequality and the last inequality holds due to $\sum_{h'=h}^{H} \dim_{\alpha,K}(\mathcal{F}_{h'}) \leq \sum_{h'=1}^{H} \dim_{\alpha,K}(\mathcal{F}_{h'}) = H \dim_{\alpha,K}(\mathcal{F})$. Furthermore, taking a summation of (E.33), we have

$$
\begin{aligned}
&\sum_{k=1}^{K} \sum_{h=1}^{H} \left[\mathbb{P}_h(V_{k,h+1} - \check{V}_{k,h+1})\right](s_h^k, a_h^k) \\
&= \sum_{k=1}^{K} \sum_{h=1}^{H} \left(V_{k,h+1}(s_{h+1}^k) - \check{V}_{k,h+1}(s_{h+1}^k)\right)
\end{aligned}
$$

$$+ \sum_{k=1}^{K} \sum_{h=1}^{H} \left( \left[ \mathbb{P}_h (V_{k,h+1} - \check{V}_{k,h+1}) \right] (s_h^k, a_h^k) - \left( V_{k,h+1}(s_{h+1}^k) - \check{V}_{k,h+1}(s_{h+1}^k) \right) \right)$$

$$\leq \sum_{k=1}^{K} \sum_{h=1}^{H} \left( V_{k,h+1}(s_{h+1}^k) - \check{V}_{k,h+1}(s_{h+1}^k) \right) + \sqrt{2HK \log(H/\delta)}$$

$$\leq 2CH^2(1+\chi)(1 + \widehat{\beta}_k \gamma^2) \dim_{\alpha,K}(\mathcal{F}_h) + 4CH(1+\chi)\widehat{\beta}_k \sqrt{\dim_{\alpha,K}(\mathcal{F})} \sqrt{H \sum_{k=1}^{K} \sum_{h=1}^{H} (\sigma_{k,h}^2 + \alpha^2)}$$

$$+ 3\sqrt{2H^3 K \log(H/\delta)},$$

where the first inequality holds due to event $\mathcal{E}_2$ and the last inequality holds due to (E.33). Thus, we complete the proof of Lemma D.15. $\qquad\square$

### E.4.5 PROOF OF LEMMA D.16

*Proof of Lemma D.16.* According to the definition of estimated variance $\sigma_{k,h}$, the summation of variance can be decomposed as following:

$$\sum_{k=1}^{K} \sum_{h=1}^{H} \sigma_{k,h}^2 = \sum_{k=1}^{K} \sum_{h=1}^{H} [\bar{\mathbb{V}}_{k,h} V_{k,h+1}](s_h^k, a_h^k) + E_{k,h} + F_{k,h}$$

$$= \underbrace{\sum_{k=1}^{K} \sum_{h=1}^{H} \left( [\bar{\mathbb{V}}_{k,h} V_{k,h+1}](s_h^k, a_h^k) - [\mathbb{V}_h V_{k,h+1}](s_h^k, a_h^k) \right)}_{I_1} + \underbrace{\sum_{k=1}^{K} \sum_{h=1}^{H} E_{k,h}}_{I_2} + \underbrace{\sum_{k=1}^{K} \sum_{h=1}^{H} F_{k,h}}_{I_3}$$

$$+ \underbrace{\sum_{k=1}^{K} \sum_{h=1}^{H} \left( [\mathbb{V}_h V_{k,h+1}](s_h^k, a_h^k) - [\mathbb{V}_h V_{h+1}^{\pi^k}](s_h^k, a_h^k) \right)}_{I_4} + \underbrace{\sum_{k=1}^{K} \sum_{h=1}^{H} [\mathbb{V}_h V_{h+1}^{\pi^k}](s_h^k, a_h^k)}_{I_5}.$$

$$(\text{E.34})$$

For the term $I_1$, it can be upper bounded by

$$I_1 = \sum_{k=1}^{K} \sum_{h=1}^{H} \left( [\bar{\mathbb{V}}_{k,h} V_{k,h+1}](s_h^k, a_h^k) - [\mathbb{V}_h V_{k,h+1}](s_h^k, a_h^k) \right) \leq \sum_{k=1}^{K} \sum_{h=1}^{H} E_{k,h}, \qquad (\text{E.35})$$

where the inequality holds due to Lemma D.9.

For the second term $I_2 = \sum_{k=1}^{K} \sum_{h=1}^{H} E_{k,h}$, we have

$$\sum_{k=1}^{K} \sum_{h=1}^{H} E_{k,h} = \sum_{k=1}^{K} \sum_{h=1}^{H} (2L\beta_k + \widetilde{\beta}_k) \min \left( 1, D_{\mathcal{F}_h}(z; z_{[k-1],h}, \bar{\sigma}_{[k-1],h}) \right)$$

$$\leq \sum_{h=1}^{H} (2L\beta_k + \widetilde{\beta}_k) \times (1 + \gamma^2) \dim_{\alpha,K}(\mathcal{F}_h)$$

$$+ \sum_{h=1}^{H} (2L\beta_k + \widetilde{\beta}_k) \times 2\sqrt{\dim_{\alpha,K}(\mathcal{F}_h)} \sqrt{\sum_{k=1}^{K} (\sigma_{k,h}^2 + \alpha^2)}$$

$$\leq (2L\beta_k + \widetilde{\beta}_k) H \times (1 + \gamma^2) \dim_{\alpha,K}(\mathcal{F})$$

$$+ (2L\beta_k + \widetilde{\beta}_k) \times 2\sqrt{\sum_{h=1}^{H} \dim_{\alpha,K}(\mathcal{F}_h)} \sqrt{\sum_{k=1}^{K} \sum_{h=1}^{H} (\sigma_{k,h}^2 + \alpha^2)}$$

$$= (2L\beta_k + \widetilde{\beta}_k) H \times (1 + \gamma^2) \dim_{\alpha,K}(\mathcal{F})$$

$$+ (2L\beta_k + \widetilde{\beta}_k) \times 2\sqrt{\dim_{\alpha,K}(\mathcal{F})} \sqrt{H \sum_{k=1}^{K} \sum_{h=1}^{H} (\sigma_{k,h}^2 + \alpha^2)}, \qquad (\text{E.36})$$

where the inequality holds due to Lemma D.13 and the second inequality holds due to Cauchy-Schwartz inequality.

For the term $I_3$, we have

$$
\begin{aligned}
I_3 &= \sum_{k=1}^{K}\sum_{h=1}^{H} F_{k,h} \\
&= (\log \mathcal{N}_{\mathcal{F}}(\epsilon) + \log \mathcal{N}_{\epsilon}(K)) \\
&\quad \times \sum_{k=1}^{K}\sum_{h=1}^{H} \min\left(1, 2\widehat{f}_{k,h}(s_h^k, a_h^k) - 2\widecheck{f}_{k,h}(s_h^k, a_h^k) + 4\beta_k D_{\mathcal{F}_h}(z; z_{[k-1],h}, \bar{\sigma}_{[k-1],h})\right) \\
&\leq (\log \mathcal{N}_{\mathcal{F}}(\epsilon) + \log \mathcal{N}_{\epsilon}(K)) \\
&\quad \times \sum_{k=1}^{K}\sum_{h=1}^{H} \min\left(1, 2\mathcal{T}_h V_{k,h+1}(s_h^k, a_h^k) - 2\mathcal{T}_h \widecheck{V}_{k,h+1}(s_h^k, a_h^k)(s_h^k, a_h^k) + 8\beta_k D_{\mathcal{F}_h}(z; z_{[k-1],h}, \bar{\sigma}_{[k-1],h})\right) \\
&\leq (\log \mathcal{N}_{\mathcal{F}}(\epsilon) + \log \mathcal{N}_{\epsilon}(K)) \times \sum_{k=1}^{K}\sum_{h=1}^{H}\left[\mathbb{P}_h(V_{k,h+1} - \widecheck{V}_{k,h+1})\right](s_h^k, a_h^k) \\
&\quad + (\log \mathcal{N}_{\mathcal{F}}(\epsilon) + \log \mathcal{N}_{\epsilon}(K)) \times \sum_{k=1}^{K}\sum_{h=1}^{H} \min\left(1, 8\beta_k D_{\mathcal{F}_h}(z; z_{[k-1],h}, \bar{\sigma}_{[k-1],h})\right) \\
&\leq (\log \mathcal{N}_{\mathcal{F}}(\epsilon) + \log \mathcal{N}_{\epsilon}(K)) \times \sum_{k=1}^{K}\sum_{h=1}^{H}\left[\mathbb{P}_h(V_{k,h+1} - \widecheck{V}_{k,h+1})\right](s_h^k, a_h^k) \\
&\quad + (\log \mathcal{N}_{\mathcal{F}}(\epsilon) + \log \mathcal{N}_{\epsilon}(K)) \times (1 + 8\beta_k \gamma^2) H \dim_{\alpha,K}(\mathcal{F}) \\
&\quad + ((\log \mathcal{N}_{\mathcal{F}}(\epsilon) + \log \mathcal{N}_{\epsilon}(K)) \times 16\beta_k \sqrt{\dim_{\alpha,K}(\mathcal{F})}\sqrt{H\sum_{k=1}^{K}\sum_{h=1}^{H}(\sigma_{k,h}^2 + \alpha^2)} \\
&\leq (\log \mathcal{N}_{\mathcal{F}}(\epsilon) + \log \mathcal{N}_{\epsilon}(K)) \times \left(4CH^2(1+\chi)(1+\widehat{\beta}_k\gamma^2)\dim_{\alpha,K}(\mathcal{F}_h) + 2\sqrt{2H^3 K \log(H/\delta)}\right) \\
&\quad + (\log \mathcal{N}_{\mathcal{F}}(\epsilon) + \log \mathcal{N}_{\epsilon}(K)) \times (1 + 8\beta_k \gamma^2) H \dim_{\alpha,K}(\mathcal{F}) \\
&\quad + ((\log \mathcal{N}_{\mathcal{F}}(\epsilon) + \log \mathcal{N}_{\epsilon}(K)) \times 16\beta_k \sqrt{\dim_{\alpha,K}(\mathcal{F})}\sqrt{H\sum_{k=1}^{K}\sum_{h=1}^{H}(\sigma_{k,h}^2 + \alpha^2)}, \quad\quad \text{(E.37)}
\end{aligned}
$$

where the first inequality holds due to Lemma D.8, the second inequality holds due to $V_{k,h+1}(\cdot) \geq V_{h+1}^*(\cdot) \geq \widecheck{V}_{k,h+1}(\cdot)$, the third inequality holds due to Lemma D.13 with Cauchy-Schwartz inequality and the last inequality holds due to D.15.

For the term $I_4$, we have

$$
\begin{aligned}
I_4 &= \sum_{k=1}^{K}\sum_{h=1}^{H}\left([\mathbb{V}_h V_{k,h+1}](s_h^k, a_h^k) - [\mathbb{V}_h V_{h+1}^{\pi^k}](s_h^k, a_h^k)\right) \\
&= \sum_{k=1}^{K}\sum_{h=1}^{H}\left([\mathbb{P}_h V_{k,h+1}^2](s_h^k, a_h^k) - \left([\mathbb{P}_h V_{k,h+1}](s_h^k, a_h^k)\right)^2 - [\mathbb{P}_h(V_{h+1}^{\pi^k})^2](s_h^k, a_h^k) + \left([\mathbb{P}_h V_{h+1}^{\pi^k}](s_h^k, a_h^k)\right)^2\right) \\
&\leq \sum_{k=1}^{K}\sum_{h=1}^{H}\left([\mathbb{P}_h V_{k,h+1}^2](s_h^k, a_h^k) - [\mathbb{P}_h(V_{h+1}^{\pi^k})^2](s_h^k, a_h^k)\right) \\
&\leq 2\sum_{k=1}^{K}\sum_{h=1}^{H}\left([\mathbb{P}_h V_{k,h+1}](s_h^k, a_h^k) - [\mathbb{P}_h V_{h+1}^{\pi^k}](s_h^k, a_h^k)\right) \\
&\leq 8CH^2(1+\chi)(1+\widehat{\beta}_k\gamma^2)\dim_{\alpha,K}(\mathcal{F}) + 16CH(1+\chi)\widehat{\beta}_k\sqrt{\dim_{\alpha,K}(\mathcal{F})}\sqrt{H\sum_{k=1}^{K}\sum_{h=1}^{H}(\sigma_{k,h}^2 + \alpha^2)}
\end{aligned}
$$

$$+ 4\sqrt{2H^3 K \log(H/\delta)}, \tag{E.38}$$

where the first inequality holds due to $V_{k,h+1}(\cdot) \geq V_{h+1}^*(\cdot) \geq V_{h+1}^{\pi^k}(\cdot)$, the second inequality holds due to $0 \leq V_{h+1}^*(\cdot), V_{h+1}^{\pi^k}(\cdot) \leq 1$ and the last iniequality holds due to Lemma D.14.
For the term $I_5$, according to the definition of event $\mathcal{E}_3$, we have

$$I_5 = \sum_{k=1}^{K} \sum_{h=1}^{H} [\mathbb{V}_h V_{h+1}^{\pi^k}](s_h^k, a_h^k) \leq 3K + 3H \log(1/\delta). \tag{E.39}$$

Substituting the results in (E.35), (E.36), (E.37), (E.38) and (E.39) into (E.34), we have

$$\sum_{k=1}^{K} \sum_{h=1}^{H} \sigma_{k,h}^2$$
$$= I_1 + I_2 + I_3 + I_4 + I_5$$
$$\leq (4L\beta_k + 2\widetilde{\beta}_k) H \times (1 + \gamma^2) \dim_{\alpha,K}(\mathcal{F})$$

$$+ (4L\beta_k + 2\widetilde{\beta}_k) \times 2\sqrt{\dim_{\alpha,K}(\mathcal{F})} \sqrt{H \sum_{k=1}^{K} \sum_{h=1}^{H} (\sigma_{k,h}^2 + \alpha^2)}$$

$$+ 8CH^2(1+\chi)(1+\widehat{\beta}_k\gamma^2) \dim_{\alpha,K}(\mathcal{F}) + 16CH(1+\chi)\widehat{\beta}_k \sqrt{\dim_{\alpha,K}(\mathcal{F})} \sqrt{H \sum_{k=1}^{K} \sum_{h=1}^{H} (\sigma_{k,h}^2 + \alpha^2)}$$

$$+ 4\sqrt{2H^3 K \log(H/\delta)} + 3K + 3H \log(1/\delta)$$
$$+ (\log \mathcal{N}_{\mathcal{F}}(\epsilon) + \log \mathcal{N}_{\epsilon}(K)) \times \left( 4CH^2(1+\chi)(1+\widehat{\beta}_k\gamma^2) \dim_{\alpha,K}(\mathcal{F}_h) + 2\sqrt{2H^3 K \log(H/\delta)} \right)$$

$$+ (\log \mathcal{N}_{\mathcal{F}}(\epsilon) + \log \mathcal{N}_{\epsilon}(K)) \times 8CH(1+\chi)\widehat{\beta}_k \sqrt{\dim_{\alpha,K}(\mathcal{F})} \sqrt{H \sum_{k=1}^{K} \sum_{h=1}^{H} (\sigma_{k,h}^2 + \alpha^2)}$$

$$+ (\log \mathcal{N}_{\mathcal{F}}(\epsilon) + \log \mathcal{N}_{\epsilon}(K)) \times (1 + 8\beta_k\gamma^2) H \dim_{\alpha,K}(\mathcal{F})$$

$$+ (\log \mathcal{N}_{\mathcal{F}}(\epsilon) + \log \mathcal{N}_{\epsilon}(K)) \times 16\beta_k \sqrt{\dim_{\alpha,K}(\mathcal{F})} \sqrt{H \sum_{k=1}^{K} \sum_{h=1}^{H} (\sigma_{k,h}^2 + \alpha^2)}$$

$$\leq 3K + 3H \log(1/\delta) + (\log \mathcal{N}_{\mathcal{F}}(\epsilon) + \log \mathcal{N}_{\epsilon}(K)) \times O\left( (1+\gamma^2)(\beta_k + H\widehat{\beta}_k + \widetilde{\beta}_k) H \dim_{\alpha,K}(\mathcal{F}) \right)$$

$$+ (\log \mathcal{N}_{\mathcal{F}}(\epsilon) + \log \mathcal{N}_{\epsilon}(K)) \times O\left( (\beta_k + H\widehat{\beta}_k + \widetilde{\beta}_k) \sqrt{\dim_{\alpha,K}(\mathcal{F})} \sqrt{H \sum_{k=1}^{K} \sum_{h=1}^{H} (\sigma_{k,h}^2 + \alpha^2)} \right)$$

$$+ (\log \mathcal{N}_{\mathcal{F}}(\epsilon) + \log \mathcal{N}_{\epsilon}(K))^2 \times O\left( H^3 \log(H/\delta) \right) + O(K),$$

where the last inequality holds due to $2ab \leq a^2 + b^2$. Notice that for each variable $x$, $x \leq a\sqrt{x} + b$ implies $x \leq a^2 + 2b$. With this fact, we have

$$\sum_{k=1}^{K} \sum_{h=1}^{H} \sigma_{k,h}^2 \leq (\log \mathcal{N}_{\mathcal{F}}(\epsilon) + \log \mathcal{N}_{\epsilon}(K)) \times O\left( (1+\gamma^2)(\beta_k + H\widehat{\beta}_k + \widetilde{\beta}_k) H \dim_{\alpha,K}(\mathcal{F}) \right)$$
$$+ (\log \mathcal{N}_{\mathcal{F}}(\epsilon) + \log \mathcal{N}_{\epsilon}(K))^2 \times O\left( (\beta_k + H\widehat{\beta}_k + \widetilde{\beta}_k)^2 H \dim_{\alpha,K}(\mathcal{F}) \right)$$
$$+ O(K + KH\alpha^2).$$

Thus, we complete the proof of Lemma D.16. $\qquad\qquad\square$

## F COVERING NUMBER ARGUMENT

### F.1 RARE SWITCHING

Based on the policy-updating criterion, the following lemma provides a upper bound of the switching cost.

**Lemma F.1.** The number of episodes when the algorithm updates the value function is at most $O\left(\dim_{\alpha,K}(\mathcal{F}) \cdot H\right)$.

*Proof.* According to line 9, the policy is updated at episode $k$ only when there exists a stage $h \in [H]$ such that

$$\sum_{i \in [k_{last}, k-1]} \frac{1}{\bar{\sigma}_{i,h}^2} D_{\mathcal{F}_h}^2(z_{i,h}; z_{[k_{last}-1],h}, \bar{\sigma}_{[k_{last}-1],h}) \geq \chi/C.$$

and

$$\sum_{i \in [k_{last}, k-2]} \frac{1}{\bar{\sigma}_{i,h}^2} D_{\mathcal{F}}^2(z_{i,h}; z_{[k_{last}-1],h}, \bar{\sigma}_{[k_{last}-1],h}) < \chi. \tag{F.1}$$

Then the following inequality holds,

$$\sup_{f_1, f_2 \in \mathcal{F}_h} \frac{\sum_{i \in [1, k-2]} \frac{1}{\bar{\sigma}_{i,h}^2}(f_1(z_{i,h}) - f_2(z_{i,h}))^2 + \lambda}{\sum_{i \in [1, k_{last}-1]} \frac{1}{\bar{\sigma}_{i,h}^2}(f_1(z_{i,h}) - f_2(z_{i,h}))^2 + \lambda} \tag{F.2}$$

$$= 1 + \sup_{f_1, f_2 \in \mathcal{F}_h} \frac{\sum_{i \in [k_{last}, k-2]} \frac{1}{\bar{\sigma}_{i,h}^2}(f_1(z_{i,h}) - f_2(z_{i,h}))^2}{\sum_{i \in [1, k_{last}-1]} \frac{1}{\bar{\sigma}_{i,h}^2}(f_1(z_{i,h}) - f_2(z_{i,h}))^2 + \lambda}$$

$$\leq 1 + \sum_{i \in [k_{last}, k-2]} \frac{1}{\bar{\sigma}_{i,h}^2} D_{\mathcal{F}_h}^2(z_{i,h}; z_{[k_{last}-1],h}, \bar{\sigma}_{[k_{last}-1],h})$$

$$\leq 1 + \chi, \tag{F.3}$$

where the first inequality holds due to the definition of $D_{\mathcal{F}_h}$ (Definition 2.4), the second inequality follows from (F.1).

(F.3) further gives a lower bound for the summation

$$\sum_{i \in [k_{last}, k-1]} \frac{1}{\bar{\sigma}_{i,h}^2} D_{\mathcal{F}}^2(z_{i,h}; z_{[i-1],h}, \bar{\sigma}_{[i-1],h})$$

$$\geq \frac{1}{1+\chi} \sum_{i \in [k_{last}, k-1]} \frac{1}{\bar{\sigma}_{i,h}^2} D_{\mathcal{F}}^2(z_{i,h}; z_{[k_{last}-1],h}, \bar{\sigma}_{[k_{last}-1],h})$$

$$\geq \frac{\chi/C}{1+\chi}.$$

Note that $\frac{\chi/C}{1+\chi} \leq 1$, we also have

$$\sum_{i \in [k_{last}, k-1]} \min \left\{ 1, \frac{1}{\bar{\sigma}_{i,h}^2} D_{\mathcal{F}}^2(z_{i,h}; z_{[i-1],h}, \bar{\sigma}_{[i-1],h}) \right\} \geq \frac{\chi/C}{1+\chi}.$$

Then we have an upper bound and lower bound for the following summation:

$$l_K \cdot \frac{\chi/C}{1+\chi} \leq \sum_{k=1}^{K} \sum_{h=1}^{H} \min \left\{ 1, \frac{1}{\bar{\sigma}_{k,h}^2} D_{\mathcal{F}}^2(z_{k,h}; z_{[k-1],h}, \bar{\sigma}_{[k-1],h}) \right\} \leq \dim_{\alpha,K}(\mathcal{F}) \cdot H$$

Therefore, the number of policy switching $l_K$ is of order $O(\dim_{\alpha,K}(\mathcal{F}) \cdot H)$.

$\square$

**Lemma F.2** (Stability of uncertainty under rare switching strategy)**.** If the policy is not updated at episode $k$, the uncertainty of all state-action pair $z = (s, a) \in \mathcal{S} \times \mathcal{A}$ and stage $h \in [H]$ satisfies the following stability property:

$$D_{\mathcal{F}_h}^2(z; z_{[k-1],h}, \bar{\sigma}_{[k-1],h}) \geq \frac{1}{1+\chi} D_{\mathcal{F}_h}^2(z; z_{[k_{last}-1],h}, \bar{\sigma}_{[k_{last}-1],h}).$$

*Proof.* Due to the definition of $k_{last}$ in Algorithm 1, we have

$$\sum_{i \in [k_{last}, k-1]} \frac{1}{\bar{\sigma}_{i,h}^2} D_{\mathcal{F}}^2(z_{i,h}; z_{[k_{last}-1],h}, \bar{\sigma}_{[k_{last}-1],h}) < \chi.$$

As is shown in (F.3), here we also have

$$\sup_{f_1, f_2 \in \mathcal{F}_h} \frac{\sum_{i \in [1, k-1]} \frac{1}{\bar{\sigma}_{i,h}^2}(f_1(z_{i,h}) - f_2(z_{i,h}))^2 + \lambda}{\sum_{i \in [1, k_{last}-1]} \frac{1}{\bar{\sigma}_{i,h}^2}(f_1(z_{i,h}) - f_2(z_{i,h}))^2 + \lambda} \leq 1 + \chi.$$

From the definition of $D_{\mathcal{F}_h}$,

$$D_{\mathcal{F}_h}^2(z; z_{[k-1],h}, \bar{\sigma}_{[k-1],h}) \geq \frac{1}{1+\chi} D_{\mathcal{F}_h}^2(z; z_{[k_{last}-1],h}, \bar{\sigma}_{[k_{last}-1],h}).$$

The proof is then completed due to the arbitrariness of $h$. □

## F.2 VALUE FUNCTION CLASS AND ITS COVERING NUMBER

The optimistic value functions at episode $k$ and stage $h \in [H]$ in our construction belong to the following function class:

$$\mathcal{V}_{k,h} = \left\{ V \middle| \max_{a \in \mathcal{A}} \min_{1 \leq i \leq l_k+1} \min\left(1, f_i(\cdot, a) + \beta \cdot b(\cdot, a)\right) \right\}, \tag{F.4}$$

where $l_k$ is the number of updated policies as defined in Algorithm 1, $f_i \in \mathcal{F}_h$ and $b \in \mathcal{B}$.
Similarly, we also define the following pessimistic value function classes for all $k \geq 1$:

$$\check{\mathcal{V}}_{k,h} = \left\{ V \middle| \max_{a \in \mathcal{A}} \max_{1 \leq i \leq l_k+1} \max\left(0, f_i(\cdot, a) - \beta \cdot b(\cdot, a)\right) \right\}, \tag{F.5}$$

**Lemma F.3** ($\epsilon$-covering number of optimistic value function classes). *For optimistic value function class $\mathcal{V}_{k,h}$ defined in (F.4), we define the distance between two value functions $V_1$ and $V_2$ as $\|V_1 - V_2\|_\infty := \max_{s \in \mathcal{S}} |V_1(s) - V_2(s)|$. Then the $\epsilon$-covering number with respect to the distance function can be upper bounded by*

$$\mathcal{N}_\epsilon(k) := [\mathcal{N}_{\mathcal{F}}(\epsilon/2) \cdot \mathcal{N}(\mathcal{B}, \epsilon/2\beta)]^{l_k+1} \tag{F.6}$$

*Proof.* By the definition of $\mathcal{N}(\mathcal{F}, \epsilon)$, there exists an $\epsilon/2$-net of $\mathcal{F}$, denoted by $\mathcal{C}(\mathcal{F}, \epsilon/2)$, such that for any $f \in \mathcal{F}$, we can find $f' \in \mathcal{C}(\mathcal{F}, \epsilon/2)$ such that $\|f - f'\|_\infty \leq \epsilon/2$. Also, there exists an $\epsilon/2\beta$-net of $\mathcal{B}$, $\mathcal{C}(\mathcal{B}, \epsilon/2\beta)$.
Then we consider the following subset of $\mathcal{V}_k$,

$$\mathcal{V}^c = \left\{ V \middle| \max_{a \in \mathcal{A}} \min_{1 \leq i \leq l_k+1} \min\left(1, f_i(\cdot, a) + \beta \cdot b_i(\cdot, a)\right), f_i \in \mathcal{C}(\mathcal{F}_h, \epsilon/2), b_i \in \mathcal{C}(\mathcal{B}, \epsilon/2\beta) \right\}.$$

Consider an arbitrary $V \in \mathcal{V}$ where $V = \max_{a \in \mathcal{A}} \min_{1 \leq i \leq l_k+1} \min(1, f_i(\cdot, a) + \beta \cdot b_i(\cdot, a))$. For each $f_i$, there exists $f_i^c \in \mathcal{C}(\mathcal{F}, \epsilon/2)$ such that $\|f_i - f_i^c\|_\infty \leq \epsilon/2$. There also exists $b^c \in \mathcal{C}(\mathcal{B}, \epsilon/2\beta)$ such that $\|b - b^c\|_\infty \leq \epsilon/2\beta$. Let $V^c = \max_{a \in \mathcal{A}} \min_{1 \leq i \leq l_k+1} \min(1, f_i^c(\cdot, a) + \beta \cdot b^c(\cdot, a)) \in \mathcal{V}^c$. It is then straightforward to check that $\|V - V^c\|_\infty \leq \epsilon/2 + \beta \cdot \epsilon/2\beta = \epsilon$.
By direct calculation, we have $|\mathcal{V}^c| = [\mathcal{N}(\mathcal{F}_h, \epsilon/2) \cdot \mathcal{N}(\mathcal{B}, \epsilon/2\beta)]^{l_k+1}$.
□

**Lemma F.4** ($\epsilon$-covering number of pessimistic value function classes). *For pessimistic value function class $\check{\mathcal{V}}_{k,h}$ defined in (F.5), we define the distance between two value functions $V_1$ and $V_2$ as $\|V_1 - V_2\|_\infty := \max_{s \in \mathcal{S}} |V_1(s) - V_2(s)|$. Then the $\epsilon$-covering number of $\check{\mathcal{V}}_k$ with respect to the distance function can be upper bounded by $\mathcal{N}_\epsilon(k)$ defined in (F.6).*

*Proof.* The proof is nearly the same as that of Lemma F.3. □

## G  AUXILIARY LEMMAS

**Lemma G.1** (Azuma–Hoeffding inequality). Let $\{x_i\}_{i=1}^n$ be a martingale difference sequence with respect to a filtration $\{\mathcal{G}_i\}$ satisfying $|x_i| \le M$ for some constant $M$, $x_i$ is $\mathcal{G}_{i+1}$-measurable, $\mathbb{E}[x_i|\mathcal{G}_i] = 0$. Then for any $0 < \delta < 1$, with probability at least $1 - \delta$, we have

$$\sum_{i=1}^n x_i \le M\sqrt{2n\log(1/\delta)}.$$

**Lemma G.2.** Let $M > 0, V > v > 0$ be constants, and $\{x_i\}_{i \in [t]}$ be stochastic process adapted to a filtration $\{\mathcal{H}_i\}_{i \in [t]}$. Suppose $\mathbb{E}[x_i|\mathcal{H}_{i-1}] = 0$, $|x_i| \le M$ and $\sum_{i \in [t]} \mathbb{E}[x_i^2|\mathcal{H}_{i-1}] \le V^2$ almost surely. Then for any $\delta, \epsilon > 0$, let $\iota = \sqrt{\log \frac{(2\log(V/v)+2)\cdot(\log(M/m)+2)}{\delta}}$ we have

$$\mathbb{P}\left(\sum_{i \in [t]} x_i > \iota\sqrt{2\left(2\sum_{i \in [t]} \mathbb{E}[x_i^2|\mathcal{H}_{i-1}] + v^2\right) + \frac{2}{3}\iota^2\left(2\max_{i \in [t]}|x_i| + m\right)}\right) \le \delta.$$

**Lemma G.3** (Lemma 7, Russo & Van Roy 2014). Consider random variables $(Z_n | n \in \mathbb{N})$ adapted to the filtration $(\mathcal{H}_n : n = 0, 1, ...)$. Assume $\mathbb{E}[\exp\{\lambda Z_i\}]$ is finite for all $\lambda$. Define the conditional mean $\mu_i = \mathbb{E}[Z_i | \mathcal{H}_{i-1}]$. We define the conditional cumulant generating function of the centered random variable $[Z_i - \mu_i]$ by $\psi_i(\lambda) = \log \mathbb{E}[\exp(\lambda[Z_i - \mu_i]) | \mathcal{H}_{i-1}]$. For all $x \ge 0$ and $\lambda \ge 0$,

$$\mathbb{P}\left(\sum_1^n \lambda Z_i \le x + \sum_1^n [\lambda\mu_i + \psi_i(\lambda)] \quad \forall n \in \mathbb{N}\right) \ge 1 - e^{-x}.$$

**Lemma G.4** (Self-normalized bound for scalar-valued martingales). Consider random variables $(v_n | n \in \mathbb{N})$ adapted to the filtration $(\mathcal{H}_n : n = 0, 1, ...)$. Let $\{\eta_i\}_{i=1}^\infty$ be a sequence of real-valued random variables which is $\mathcal{H}_{i+1}$-measurable and is conditionally $\sigma$-sub-Gaussian. Then for an arbitrarily chosen $\lambda > 0$, for any $\delta > 0$, with probability at least $1 - \delta$, it holds that

$$\sum_{i=1}^n \epsilon_i v_i \le \frac{\lambda\sigma^2}{2} \cdot \sum_{i=1}^n v_i^2 + \log(1/\delta)/\lambda \quad \forall n \in \mathbb{N}.$$

