# OpenReview forum: "A Nearly Optimal and Low-Switching Algorithm for Reinforcement Learning with General Function Approximation"
_ICLR.cc/2024/Conference — Submitted to ICLR 2024_

### Official Review · Reviewer_rQVV · 2023-10-19

**Soundness:** 3 good
**Presentation:** 2 fair
**Contribution:** 3 good
**Rating:** 5
**Confidence:** 3

**Summary:**

In this paper, the authors propose an algorithm named MQL-UCB for Reinforcement Learning with general function approximation. The MDP setting satisfies that the Bellman operator could map any value functions into a function class with a bounded Eluder dimension. The algorithm is shown to achieve near optimal regret and switching cost simultaneously.

**Strengths:**

1. The setting of reinforcement learning with low switching cost and general function approximation is interesting and important.

2. The paper is solid, the proof looks correct to me.

3. Both the regret bound and the switching cost bound is nearly minimax optimal.

**Weaknesses:**

1. My main concern is about technical novelty. It seems that the only different component compared to LSVI-UCB++ is the policy switching framework. It would be better if the authors could highlight the technical novelty, especially the part that is different from [1].

[1] Jiafan He, Heyang Zhao, Dongruo Zhou, and Quanquan Gu. Nearly minimax optimal reinforcement learning for linear markov decision processes.

2. It is not very clear how general the function approximation is. It would be better if the authors could discuss more about the relationship between this setting and some other settings, like the linear Bellman complete MDP and the Eluder Condition class. Under these settings, [2] and [3] design algorithms with low switching cost and sub-linear regret bound.

[2] Dan Qiao, Ming Yin, and Yu-Xiang Wang. Logarithmic switching cost in reinforcement learning beyond linear mdps.

[3] Nuoya Xiong, Zhaoran Wang and Zhuoran Yang. A General Framework for Sequential Decision-Making under Adaptivity Constraints.

3. The presentation is not very clear. As a paper focused on theory, a proof sketch is expected. In addition, in line 6 of Algorithm 1, the switching condition is replaced by '...', which is very confusing (although it is discussed in the main text).

**Questions:**

Please see the weakness section.

---

> ### Author Response · Authors · 2023-11-22
> **Response to Reviewer rQVV**
>
> Thank you  for your encouraging feedback!
>
> **Q1** My main concern is about technical novelty. It seems that the only different component compared to LSVI-UCB++ is the policy-switching framework. It would be better if the authors could highlight the technical novelty, especially the part that is different from [1].
>
> **A1** Our main technical novelty lies in the new and more efficient policy-switching scheme for RL with general function approximation, where the policy is updated when the cumulative sensitivity score exceeds a predetermined threshold (as discussed in Section 3.1). Unlike the linear function approximation setting, where the rare-switching technique has been extensively studied and based on the determinant of the covariance [1], it is relatively underexplored for the function class with bounded generalized eluder dimension. We propose a low-switching scheme based on the $D^2$ quantity originally proposed by [2, 3]. Compared with [4], our algorithm has a tractable value-iteration-based planning phase and does not need to recompute the cumulative loss at each episode to check the switching criterion.  [5] also considered RL with low-switching cost beyond linear function approximation, i.e., MDPs with low inherent Bellman error and generalized linear MDPs, both of which belong to the class of MDPs studied in our paper. Their approach can be seen as a slight extension of RL with low-switching cost for linear MDPs, since in both settings, the covariance matrix still exists and they can still use the determinant of the covariance as a criterion for policy switching. Furthermore, we have proved that the low switching cost of our algorithm is nearly optimal (See Appendix C for the new lower bound and its proof.)
>
> With this new rare-switching scheme, it becomes possible to maintain a monotonic series of optimistic and pessimistic value functions while guaranteeing that the covering number of the value function class is still tractable. Previously, it was shown in [1] that ensuring monotonic value functions is crucial to obtaining minimax optimal regret bound for linear MDPs. However, it is underexplored whether similar idea can be applied to nonlinear function approximation. As discussed in Section 1, prior to our approach, the only minimax regret bound beyond linear function approximation was achieved by [3], but with a complex non-Markovian planning phase. Our work is the first work for RL with general function approximation that achieves the nearly minimax optimal regret when specialized to linear MDPs, while still enjoys simple Markov planning phase.
>
> We have also highlighted the above technical novelty in Section 1 and Section 3 in the revision.
>
> ---
>
> **Q2** It is not very clear how general the function approximation is. It would be better if the authors could discuss more about the relationship between this setting and some other settings, like the linear Bellman complete MDP and the Eluder Condition class. Under these settings, [4] and [5] design algorithms with low switching cost and sub-linear regret bound.
>
> **A2** [5] considers a class of MDPs with generalized linear function, which belongs to the class of MDPs with bounded eluder dimension [6]. According to the discussion in Section 4.1, the class of MDPs studied in our paper contains the class of MDPs with bounded eluder dimension.
>
> As for [4], it is known that both their setting and ours capture MDPs with bounded eluder dimension as special cases. However, it is unclear which one is more general and we leave it for future study.
>
> We have also added a discussion on the two related works in Section 1 in our revision.
>
> ---
>
> **Q3** The presentation is not very clear. As a paper focused on theory, a proof sketch is expected. In addition, in line 6 of Algorithm 1, the switching condition is replaced by '...', which is very confusing (although it is discussed in the main text).
>
> **A3** Thank you for your suggestion. Due to the space limit of the main text, the high-level proof of the main theorems are deferred to Appendix D. We will add a proof sketch in the main text provided we are granted additional pages in the final version.
>
> ---
>
> [1] He et al., Nearly minimax optimal reinforcement learning for linear markov decision processes.
>
> [2] Ye et al., Corruption-robust algorithms with uncertainty weighting for nonlinear contextual bandits and markov decision processes.
>
> [3] Agarwal et al., Voql: Towards optimal regret in model-free rl with nonlinear function approximation.
>
> [4] Xiong et al., A General Framework for Sequential Decision-Making under Adaptivity Constraints.
>
> [5] Qiao et al., Logarithmic switching cost in reinforcement learning beyond linear mdps.
>
> [6] Russo et al., Eluder dimension and the sample complexity of optimistic exploration

---

### Official Review · Reviewer_BQ2A · 2023-10-19

**Soundness:** 3 good
**Presentation:** 2 fair
**Contribution:** 2 fair
**Rating:** 6
**Confidence:** 4

**Summary:**

This paper considers RL setting with general function approximations, and targets at achieving near-optimal regret with only a small number of policy switching.
The authors consider the same Generalized Eluder dimension in (Agarwal et al., 2022) as the complexity measure for function classes, and contribute an algorithm, which only performs policy updates once the collected information goes beyond a threshold from last updating.
The authors also show that their algorithm can achieve sublinear regret (near-optimal when restricted to the linear class) with low policy switching.

**Strengths:**

The problem studied in this paper is well-motivated.
The algorithm proposed in this paper indeed achieves some improvement comparing with previous works.
The authors also include several informative discussions, e.g. connection between $D^2_{\mathcal{F}}$-uncertainty and Eluder dimension.

**Weaknesses:**

1. I have doubts on whether the claim that "MQL-UCB achieves near-optimal policy switching" is true.

   It seems to me this paper only considered the case where the algorithm is only allowed to deploy deterministic policy at each iteration, and the $\Omega(dH)$ policy switching lower bound the authors compared with also only holds for such setting. Intuitively, stochastic policies should have better exploartion ability than deterministic policies and should be preferred more in deployment efficienct learning setting. There are also several closely related works ([1] and [2]) have reported that when the algorithm can deploy stochastic or even non-Markovian policies at each iteration, the lower bound of deployment complexity would be $\Omega(H)$ and can be matched.

   However, such clarification is missing and those closely related works are also not mentioned in the paper. I would suggest the authors clarify about this issue somewhere in the introduction and include the discussion with those related works in the paper.

2. It is not very clear to me what the main novelty in technique analysis and innovations in algorithm design in this paper. There are lots of places "inspired by ...(some previous work)" in this paper, and it's not clear what are the novel parts and what are ingredients from previous work.

3. The paper writing can be improved.
* In line 6 of Algorithm 1, seems something is missing after "such that".
* It would be better to discuss the main results (section 4) before section 3

=========== Post Rebuttal ===========

Thanks for the response by authors. The new lower bound looks interesting, and my main concern is addressed, so I decide to raise my score. (and maybe please highlight a little bit that the "...initial state is arbitrarily selected..." in Sec. 2.1 (maybe use \textbf), I think it is an important difference, and it is hard to notice that if one is familiar with basic RL setting and don't carefully read the preliminary part.)

[1] Huang et. al., Towards Deployment-Efficient Reinforcement Learning: Lower Bound and Optimality

[2] Zhang et. al., Near-optimal regret bounds for multi-batch reinforcement learning

**Questions:**

What are the technical novelty and innovation in algorithm designs? Could you highlight it more clear?

---

> ### Author Response · Authors · 2023-11-22
> **Response to Reviewer BQ2A**
>
> Thank you for your insightful comments!
>
> **Q1** I have doubts on whether the claim that "MQL-UCB achieves near-optimal policy switching" is true. It seems to me this paper only considered the case where the algorithm is only allowed to deploy deterministic policy at each iteration, and the $\Omega(dH)$ policy switching lower bound the authors compared with also only holds for such setting. There are also several closely related works ([1] and [2]) have reported that when the algorithm can deploy stochastic or even non-Markovian policies at each iteration, the lower bound of deployment complexity would be $\Omega(H)$ and can be matched.
>
> **A1** Thank you for pointing out these two related works! To thoroughly show the optimality regarding the switching cost, we have added Appendix C in the revision, where a lower bound of order $\tilde \Omega(dH)$ is provided for arbitrary policies including random policies. The reason why we can achieve a different lower bound is that in our setting, we consider reset-free MDPs where the initial state $s_1$ at each episode may be generated according to the historical episodes (or chosen by the environment adaptively) instead of being chosen from a fixed distribution. Our lower bound is new in literature because the previous works such as [1], [2] only considered the environments which will reset the MDP to the same initial state (or initialized from the same state distribution) at the beginning of each episode.
>
> We have also added Remark C.3 to further clarify the difference between these two settings.
>
> ---
>
> **Q2** What are the technical novelty and innovation in algorithm designs? Could you highlight it more clearly?
>
> **A2** Our main technical novelty lies in the new and more efficient policy-switching scheme for RL with general function approximation, where the policy is updated when the cumulative sensitivity score exceeds a predetermined threshold (as discussed in Section 3.1). Unlike the linear function approximation setting, where the rare-switching technique has been extensively studied and based on the determinant of the covariance [3], it is relatively underexplored for the function class with bounded generalized eluder dimension. We propose a low-switching scheme based on the $D^2$ quantity originally proposed by [4, 5]. Compared with [6], our algorithm has a tractable value-iteration-based planning phase and does not need to recompute the cumulative loss at each episode to check the switching criterion.  [7] also considered RL with low-switching cost beyond linear function approximation, i.e., MDPs with low inherent Bellman error and generalized linear MDPs, both of which belong to the class of MDPs studied in our paper. Their approach can be seen as a slight extension of RL with low-switching cost for linear MDPs, since in both settings, the covariance matrix still exists and they can still use the determinant of the covariance as a criterion for policy switching. Furthermore, we have proved that the low switching cost of our algorithm is nearly optimal (See Appendix C for the new lower bound and its proof.)
>
> With this new rare-switching scheme, it becomes possible to maintain a monotonic series of optimistic and pessimistic value functions while guaranteeing that the covering number of the value function class is still tractable. Previously, it was shown in [5] that ensuring monotonic value functions is crucial to obtaining minimax optimal regret bound for linear MDPs. However, it is underexplored whether similar idea can be applied to nonlinear function approximation. As discussed in Section 1, prior to our approach, the only minimax regret bound beyond linear function approximation was achieved by [4], but with a complex non-Markovian planning phase. Our work is the first work for RL with general function approximation that achieves the nearly minimax optimal regret when specialized to linear MDPs, while still enjoys simple Markov planning phase.
>
> We have also highlighted the above technical novelty in Section 1 and Section 3 in the revision.
>
> [1] Huang et. al., Towards Deployment-Efficient Reinforcement Learning: Lower Bound and Optimality
>
> [2] Zhang et. al., Near-optimal regret bounds for multi-batch reinforcement learning
>
> [3] He et al. Nearly minimax optimal reinforcement learning for linear markov decision processes.
>
> [4] Ye et al., Corruption-robust algorithms with uncertainty weighting for nonlinear contextual bandits and markov decision processes.
>
> [5] Agarwal et al., Voql: Towards optimal regret in model-free rl with nonlinear function approximation.
>
> [6] Xiong et al., A General Framework for Sequential Decision-Making under Adaptivity Constraints.
>
> [7] Qiao et al., Logarithmic switching cost in reinforcement learning beyond linear mdps.

---

### Official Review · Reviewer_kAyi · 2023-10-31

**Soundness:** 2 fair
**Presentation:** 2 fair
**Contribution:** 2 fair
**Rating:** 5
**Confidence:** 3

**Summary:**

This paper proposes a new algorithm, Monotonic Q-Learning with Upper Confidence Bound, to solve reinforcement learning with general function approximation.
Specifically, the proposed algorithm is based on a new policy-switching strategy that only requires $\tilde{O}(dH)$ switches, and achieves $O(\sqrt{dim(\mathcal{F}) \log \mathcal{N} HK})$ regret.
The regret is minimax optimal in the special case of linear MDPs.

**Strengths:**

- This paper generalizes the approach by He et al (2022) to solve general function approximation. The authors extend the technique in He et al (2022) by changing the covariate matrix criterion with the newly proposed $D_{\mathcal{F}_h}^2$ criterion.
- Compared to existing papers studying similar problems, this paper is able to get near-optimal regret with fewer number of switches, and uses a simpler planning strategy.

**Weaknesses:**

- A few assumptions look really strong. For example, assuming completeness holds for any function $V$ as well as for second moments is not common in my opinion.
- The optimality of regret is only for linear MDPs, and the authors prove no lower bound for general function approximation.
- Overall, the paper could be better organized, and adding some more intuitive explanations will be helpful for readers.
- There are quite a few typos in the paper. For example, in page 3, $O$ is missing for regrets in table 1. In page 6, a parenthesis is missing in the definition of $b_{k,h}$, and there is an excessive He et al (2022). In page 8, "may differs" should be "may differ".

**Questions:**

- Please clarify points raised in "weaknesses".
- Can you elaborate more on how to find the bonus oracle $\bar{D}^2_{\mathcal{F}}$?
- Why should we care about the monotonic properties of the value functions?
- Given Theorem 4.4, can you rephrase the regret bound in terms of eluder dimension?

---

> ### Author Response · Authors · 2023-11-22
> **Response to Reviewer kAyi (1 / 2)**
>
> Thank you for your insightful comments!
>
> **Q1** A few assumptions look really strong. For example, assuming completeness holds for any function as well as for second moments is not common in my opinion.
>
> **A1** The completeness for the second moment is indeed not common in existing works on RL with general function approximation. It is first introduced by [1], and is crucial for obtaining a tighter regret bound. More specifically, making use of the variance of the value function at the next state is known to be crucial to achieve minimax-optimal regret bound in RL settings ranging from tabular MDPs [2] to linear mixture MDPs [3] and linear MDPs [4]. In MDPs with general function approximation, the second-moment completeness assumption makes the variance of value functions computationally tractable. It is still an open problem whether a nearly optimal regret can be achieved in MDPs with general function approximation without this assumption.
>
> We have also added a discussion on this point in Remark 2.3 in the revision.
>
> ---
>
> **Q2** The optimality of regret is only for linear MDPs, and the authors prove no lower bound for general function approximation.
>
> **A2** This is a great question. We would also like to clarify that in the literature of RL with general function approximation, sometimes people claim the result as an **optimal** regret bound as long as the regret is optimal when it is specialized to the linear function approximation case, e.g., [1].
>
> Currently, we don’t know how to construct a hard-to-learn instance for general function approximation in our framework. Among existing studies of RL with general function approximation, there is a lack of lower bounds that are dependent on the complexity of the function class, except DEC [5]. However, the corresponding algorithm for DEC requires solving a non-convex min-max optimization problem over the distributions of the policy and model class, which is computationally intractable. The pursuit of methodologies to obtain a lower bound for general function approximation under Bellman completeness assumption is an interesting future direction.

---

> ### Author Response · Authors · 2023-11-22
> **Response to Reviewer kAyi (2 / 2)**
>
> **Q3** Overall, the paper could be better organized, and adding some more intuitive explanations will be helpful for readers.
>
> **A3** Thank you for your suggestion. We have added more detailed explanations about our algorithmic design in Section 3.
>
> ---
>
> **Q4** There are quite a few typos in the paper. For example, in page 3, $O$ is missing for regrets in table 1. In page 6, a parenthesis is missing in the definition of $b_{k, h}$, and there is an excessive He et al (2022). In page 8, "may differs" should be "may differ".
>
> **A4** Thank you for pointing out these typos. We have corrected them in the revision.
>
> ---
>
> **Q5** Can you elaborate more on how to find the bonus oracle?
>
> **A5** In the linear case, the bonus is the elliptical norm of the feature vectors of the state-action pair, as discussed in Remark 2.5. Correspondingly, in the generalized linear setting, the bonus oracle is set to $K / \kappa$ times the elliptical norm.
> In the general case, according to the discussion provided in [6], to obtain the bonus function satisfying the required properties, we need to consider the constrained optimization problem $\\max \\bigg\\{ |f_1(z_h)-f_2(z_h)|, f_1,f_2 \\in \\mathcal{F}_h: \\sum\_{k \\in [K] } \\frac{(f\_1(z\_h\^k)- f\_2(z\_h\^k))\^2}{\\hat \\sigma\_{k,h}\^2} \\le (\\hat \\beta\_{k, h})\^2 \\bigg\\}$, and apply a binary search method to find the optimal solution to the optimization problem.
>
> ---
>
> **Q6** Why should we care about the monotonic properties of the value functions?
>
> **A6** The monotoic properties of the value function is needed for the construction of variance estimator.  For each state-action pair $(s_h^k,a_h^k)$ and any subsequent episode $i>k$, the value function $V_{i,h}$ and the corresponding variance $\\big\[\\mathbb{V}\_h (V\_{i, h+1} - V\_{h+1}^* )\\big\](s\_h\^k,a\_h\^k)$ may differ from the previous value function $V_{k,h}$ and variance $\\big\[\\mathbb{V}\_h (V\_{k,h+1}-V\_{h+1}^*)\\big\](s\_h^k,a\_h^k)$, but we still want to use the same weight $1 / \bar \sigma_{k, h}^2$ for future episode $i>k$ so that the variance $\\big\[\\mathbb{V}\_h (V\_{i,h+1}-V\_{h+1}^*)\big\](s\_h^k,a\_h^k)$ can be uniformly bounded for all $i > k$. Note that $\\big\[\\mathbb{V}\_h (V\_{i,h+1}-V\_{h+1}^*)\\big\](s\_h^k,a\_h^k) = O(\\mathbb{E}\_{s'\sim s\_h^k, a\_h^k}[V\_{i, h + 1} - V\_{h + 1}^*(s')])$. Thus, if we maintain a series of monotomic value functions, we can further control their variance terms by $O(\mathbb{E}\_{s'\sim s\_h^k, a\_h^k}[V\_{k, h + 1} - \check V\_{k, h + 1}(s')])$.
>
> ---
>
> **Q7** Given Theorem 4.4, can you rephrase the regret bound in terms of eluder dimension?
>
> **A7** Yes, according to Section 4.1, the generalized eluder dimension is upper bounded by eluder dimension up to logarithmic terms. When the number of episodes $K$ is sufficiently large, the leading term in our regret bound is $\tilde O \big(\sqrt{\dim_E(\mathcal{F}) \log \mathcal{N} \cdot HK}\big)$, where $\dim_E(\mathcal{F})$ is the eluder dimension of the function class $\mathcal{F}$.
>
> ---
>
> [1] Agarwal et al., Voql: Towards optimal regret in model-free rl with nonlinear function approximation.
>
> [2] Azar et al., Minimax Regret Bounds for Reinforcement Learning
>
> [3] Zhou et al., Nearly minimax optimal reinforcement learning for linear mixture markov decision processes
>
> [4] He et al., Nearly minimax optimal reinforcement learning for linear markov decision processes.
>
> [5] Foster et al., The statistical complexity of interactive decision making
>
> [6] Kong et al., Online sub-sampling for reinforcement learning with general function approximation.

---

### Official Review · Reviewer_XUgy · 2023-11-01

**Soundness:** 3 good
**Presentation:** 2 fair
**Contribution:** 2 fair
**Rating:** 5
**Confidence:** 3

**Summary:**

This paper considers low switching-cost algorithm with general function approximation, with the Bellman operator of the
underlying Markov decision process (MDP) is assumed to map any value functions
into a function class with a bounded eluder dimension. The key algorithmic design
includes a general deterministic policy-switching strategy that achieves low
switching cost, a monotonic value function structure with carefully controlled
function class complexity, and a variance-weighted regression scheme that
exploits historical trajectories with high data efficiency. MQL-UCB achieves
minimax optimal regret of $O(d \sqrt{HK})$ when K is sufficiently large and near-
optimal policy switching cost of $O(dH)$, with d being the eluder dimension of the function class, H being the planning horizon, and K being the number of episodes.

**Strengths:**

This paper proposes a novel policy-switching strategy based on the cumulative sensitivity of historical data. To the best of our knowledge, this is the first deterministic rare-switching strategy for RL with general function approximation which achieves Oe(dH) switching cost.

With the novel policy-switching scheme, this paper showcases how to reduce the complexity of value function classes while maintaining a series of monotonic value functions, strictly generalizing the LSVI-UCB++ algorithm to general function class with bounded eluder dimension.

**Weaknesses:**

1. Why in definition 2.8, it is reasonable to assume a bounded covering class for bonus? Since you assume covering for $\mathcal{F}$ in 1. of Def 2.8, and def. of $D_\mathcal{F}$ is clear, can you derive the covering number of the bonus class rather than assuming it exists?

2.  The update criteria in Line 6 of Algorithm 1 is incomplete and I cannot find it directly. Maybe that corresponds to the last equation in page 5? This is crucial as it will decide the switching cost of the algorithm.

3. How is policy updated in Algorithm 1 if the "if condition" in line 6 is not satisfied? Without it how do you rollout in Line 21?

4. The monotonic value function seems to be an interesting idea that can keep the conditional variance on track. However, the explanation is too short. May I ask how could you "ensure that the pessimistic value maintains a monotonically increasing property during updates, while the optimistic value function maintains a monotonically decreasing property"? Also, you say inspired by He et al., is the idea the same or there are some differences?

**Questions:**

Please answer the questions above.

---

> ### Author Response · Authors · 2023-11-22
> **Response to Reviewer XUgy**
>
> Thank you very much for your helpful comments!
>
> **Q1** Why in definition 2.8, it is reasonable to assume a bounded covering class for bonus? Since you assume covering for $\mathcal{F}$ in 1 of Def 2.8, and def. of $\mathcal{F}$ is clear, can you derive the covering number of the bonus class rather than assuming it exists?
>
> **A1** First of all, in general function approximation, it is common to introduce the additional assumption on the covering number of bonus function classes. For example, in [1], [2], and [3], the authors either assume that the bonus function class itself is a finite class or its covering number is bounded.
>
> Second, even when the function $\mathcal{F}$ is clearly defined and $|\mathcal{F}|$ is finite, the bonus function class $D_\mathcal{F}$ can still be very large. For example,  consider $\mathcal{F} = \{f_1, f_2\}$ where $f_1$ is 1 on every state-action pair, and $f_2 = 2$ on every state-action pair. Then for $\epsilon = 1 / 4$, the covering number of $D_\mathcal{F}$ is at least the number of state-action pairs. This is because the bonus functions resulted from taking each state-action pair as historical data are far from each other in $l_\infty$ measure. Given this fact, we still need to assume the covering number of the bonus function class is finite, even though the covering number of $\mathcal{F}$ is finite. This is exactly the reason why [4] assumes the covering number of the set of state-action pairs is bounded (Assumption 2). With this assumption, they can provide an upper bound for the covering number of the bonus function class $W$ in their Proposition 2.
>
> -----
> **Q2** The update criteria in Line 6 of Algorithm 1 is incomplete, and I cannot find it directly. Maybe that corresponds to the last equation in page 5? This is crucial as it will decide the switching cost of the algorithm.
>
> **A2** Thanks for pointing out the typo! It should be Eq. (3.1) on page 5. We have added the missing condition in Line 6 of Algorithm 1 in the revision.
>
> ---
>
> **Q3** How is policy updated in Algorithm 1 if the "if condition" in line 6 is not satisfied? Without it how do you rollout in Line 21?
>
> **A3** If the condition is not satisfied, we will simply use the same policy as in the last episode. We have made it clear in the revision.
>
> ---
>
> **Q4** The monotonic value function seems to be an interesting idea that can keep the conditional variance on track. However, the explanation is too short. May I ask how could you "ensure that the pessimistic value maintains a monotonically increasing property during updates, while the optimistic value function maintains a monotonically decreasing property"? Also, you say inspired by He et al. [5], is the idea the same or there are some differences?
>
> **A4** The monotonic property is ensured by our algorithmic design stated in Lines 10 and 11 in Algorithm 1, i.e., taking the maximum (or minimum) between the current value function estimate and the value function estimate in the previous episode when the policy-switching condition is satisfied. However, taking these steps will increase the covering number of the value function class exponentially with respect to $K$ in general, which will further make the uniform convergence-based concentration for the least-square estimation of value functions vacuous.
>
> To solve this issue, we apply the idea by [5] for linear MDPs, which uses rare policy-switching technique to reduce the number of value functions we are taking max or min. Unlike the linear function approximation setting, where the rare-switching technique has been extensively studied and based on the determinant of the covariance matrices, it is relatively underexplored for the function class with bounded generalized eluder dimension. To this end, we proposed a new rate-switching condition based on the $D^2$ quantity, i.e., Eq. (3.1) and Line 6 in Algorithm 1, making it possible to extend the idea by [5].
>
> [1] Ye et al., Corruption-robust algorithms with uncertainty weighting for nonlinear contextual bandits and markov decision processes.
>
> [2] Agarwal et al., Voql: Towards optimal regret in model-free rl with nonlinear function approximation.
>
> [3] Di et al., Pessimistic nonlinear least-squares value iteration for offline reinforcement learning.
>
> [4] Wang et al., Reinforcement learning with general value function approximation: Provably efficient approach via bounded eluder dimension.
>
> [5] He et al. Nearly minimax optimal reinforcement learning for linear markov decision processes.

---

### Author Response · Authors · 2023-11-23
**Looking forward to the feedback from reviewers**

Dear Reviewers:

We sincerely appreciate the time and effort you have dedicated to reviewing our paper. As the deadline for the author-reviewer discussion period is less than 12 hours away, we sincerely hope that you can provide feedback on our responses. We're happy to address any further questions you may have.

* For Reviewer XUgy, we have explained the necessity of finite covering number of the bonus function class, and the technical details of the monotonic value function.

* For Reviewer kAyi, we have explained the necessity for the second-moment completeness assumption and presented the regret bound in terms of eluder dimension.

* For Reviewer BQ2A, we have provided a new lower bound on the policy switching cost in RL with function approximation and arbitrary policies (both deterministic and stochastic policies) (See Appendix C), and gave more intuitive explanations about our algorithmic design in our revision, all highlighted in blue.

* For Reviewr rQVV, we have explained the technical contributions of our paper compared with previous works, and the relationship between our MDP class, the MDP class with bounded eluer dimension and other MDP classes.

 We would like to hear feedback from you and discuss with you if you have any further questions.

Thank you!

Authors

---

### Meta-Review · Area_Chair_T9w6 · 2023-12-07

**Metareview:**

The paper considers low switching-cost algorithm with general function approximation, under a certain (and rather strong) form of Bellman completeness.
The algorithmic contribution is a general policy-switching strategy that achieves low switching cost, a monotonic value function improvement with carefully controlled function complexity, and a variance-reduction mechanism.
In terms of main result, the algorithm achieves near optimal regret (although optimality of regret is only for linear MDPs) and policy switches.

Overall, the reviewers appreciate the paper, but also agree that the technical novelty and contribution is unclear. For example, the minimax optimility seem to apply to only the linear setting, and it is unclear if the additional proof techniques are significant in light of the already published work.

**Justification For Why Not Higher Score:**

The reviewers are unclear about the technical novelty of the present submission.

**Justification For Why Not Lower Score:**

N/A

---

### Decision · Program_Chairs · 2024-01-16

Reject